# Towards a General Attention Framework on Gyrovector Spaces for Matrix Manifolds

**Rui Wang**[1], **Chen Hu**[1], **Xiaoning Song**[1], **Xiao-Jun Wu**[1], **Nicu Sebe**[2], **Ziheng Chen**[2]*

[1]School of Artificial Intelligence and Computer Science, Jiangnan University
[2]Department of Information Engineering and Computer Science, University of Trento
`{cs_wr, 6233112017, x.song, wu_xiaojun}@jiangnan.edu.cn`
`niculae.sebe@unitn.it, ziheng_ch@163.com`

## Abstract

Deep Neural Networks (DNNs) operating on non-Euclidean geometries have recently demonstrated impressive performance across various machine-learning applications. Several studies have extended the attention mechanism to different manifolds. However, most existing non-Euclidean attention models are tailored to specific geometries, limiting their applicability. On the other hand, recent studies show that several matrix manifolds, such as Symmetric Positive Definite (SPD), Symmetric Positive Semi-Definite (SPSD), and Grassmannian manifolds, admit gyrovector structures, which extend vector addition and scalar product into manifolds. Leveraging these properties, we propose a Gyro Attention (GyroAtt) framework over general gyrovector spaces, applicable to various matrix geometries. Empirically, we manifest GyroAtt on three gyro structures on the SPD manifold, three on the SPSD manifold, and one on the Grassmannian manifold. Extensive experiments on four Electroencephalography (EEG) datasets demonstrate the effectiveness of our framework. The code is available at https://github.com/ChenHu-ML/GyroAtt.

## 1 Introduction

Recently, DNNs over Riemannian manifolds, known as Riemannian neural networks, have garnered increasing attention in various applications [26, 11, 61, 41, 13, 73, 53, 72, 18]. Commonly encountered manifolds include vector manifolds, such as hyperbolic [68] and spherical spaces [65], and matrix manifolds, such as SPD [5], SPSD [9, 10], and Grassmannian manifolds [1]. Among these non-Euclidean spaces, hyperbolic manifolds stand out due to the rich algebraic structure of gyrovector spaces [67–69], which enables principled and convenient extensions of Euclidean deep learning to hyperbolic manifolds [26, 61, 7]. In contrast, matrix manifolds offer a compelling trade-off between structural expressiveness and computational feasibility [20]. As a result, neural networks defined on matrix manifolds have emerged as appealing alternatives to their hyperbolic counterparts across various applications [40, 51, 52, 14, 37]. Notably, recent studies [40, 50–52] have demonstrated that several matrix manifolds, including SPD, SPSD, and Grassmannian, admit gyro structures, facilitating the extension of existing neural network components to these manifolds [53].

Inspired by the success of attention mechanisms in DNNs [70, 32, 22], researchers have explored their extensions to non-Euclidean geometries. Wherein, Gulcehre et al. [26] introduced attention to the hyperbolic spaces based on the hyperboloid and Klein models, while Pan et al. [55] extended attention to the SPD manifolds under the Log-Euclidean Metric (LEM). Wang et al. [72] further adapted it to the Grassmannian manifolds using an extrinsic approach under the projection distance. However, existing manifold-attention designs exhibit two key challenges: i) these designs are tailored for specific manifolds and metrics, limiting their generalizability; ii) although their network layers,

---

*Corresponding author.

39th Conference on Neural Information Processing Systems (NeurIPS 2025).

such as BiMap and ReEig in MAtt [55], and FrMap in GDLNet [72], preserve manifold constraints, they are fundamentally numerical and only partially respect the underlying Riemannian geometry.

As self-attention serves as the prototype of other attention variants, this work extends it to non-Euclidean settings. Leveraging the fact that several matrix manifolds admit gyro structures, we propose GyroAtt, a general framework for self-attention over gyrovector spaces. Unlike previous manifold attention approaches handcrafted for specific geometries [26, 55, 72], GyroAtt provides a unified building paradigm for attention design across different matrix manifolds. Notably, by intrinsically respecting Riemannian geometry and gyro algebra, GyroAtt generalizes fundamental attention components, such as linear transformations, attention computation, and feature aggregation, to gyrovector spaces. Specifically, we introduce *gyro homomorphisms*, extending linear transformations to gyrovector spaces. The attention scores are computed using a gyro distance-based function, while feature aggregation is performed via the Weighted Fréchet Mean (WFM) [25], the manifold analogue of the Euclidean weighted average. We instantiate GyroAtt on three gyro structures for the SPD manifold, three for the SPSD manifold, and one for the Grassmannian manifold. Extensive experiments on four EEG benchmarks show consistent improvements over geometry-specific manifold-attention baselines; *e.g.*, GyroAtt-SPSD outperforms MAtt [55] by **3.2%** on the MAMEM, while GyroAtt-SPD achieves **8.9%** and **7.8%** gains in inter-subject and inter-session settings on the BNCI2014001. Our **main contributions** are summarized below:

- **Generalizing attention to gyrovector spaces.** This is the first attention framework that unifies operations across diverse matrix manifolds through a common gyrovector space formulation, enabling flexible changes in the underlying geometry within a shared network structure.

- **Implementation on seven matrix gyrovector spaces.** We implement GyroAtt across three different matrix manifolds: *three gyro structures on the SPD manifold, one on the Grassmannian manifold, and three on the SPSD manifold.* To the best of our knowledge, we are the first to investigate attention mechanisms for the SPSD manifold.

- **Empirical validation under EEG signal classification.** We validate the effectiveness of GyroAtt on four EEG benchmarks. Apart from its good performance, the optimal geometries vary across different EEG tasks, demonstrating its efficacy and flexibility.

## 2 Preliminaries

This section reviews gyrogroups, gyrovector spaces, and their realizations on the SPD, Grassmannian, and SPSD manifolds, together with WFM. For further details, please refer to [68, 69, 56, 5, 10, 8].

**Gyrogroups and gyrovector spaces.** Gyrogroups and gyrovector spaces generalize groups and vector spaces, offering a powerful framework to analyze non-Euclidean geometries. A gyrogroup equips a set $(G, \oplus)$ with a non-associative "addition" $\oplus$ whose failure of associativity is controlled by a gyration $\mathrm{gyr}[\cdot, \cdot]$. Adding a scalar multiplication $\otimes$ that obeys vector-like axioms upgrades the structure to a *gyrovector space*. A review of gyrogroups and gyrovector spaces is given in App. D.1.

**SPD geometries.** Let $\mathcal{S}_d^{++}$ denote the space of $d \times d$ SPD matrices. When equipped with the Affine-Invariant Metric (AIM) [56], LEM [5], and Log-Cholesky Metric (LCM) [45], $\mathcal{S}_d^{++}$ induces three gyrovector spaces with corresponding binary operations [52] $\oplus_{ai}, \oplus_{le}$, and $\oplus_{lc}$, and associated gyro distances [19] $\mathrm{d}_{\mathrm{spd}}^{ai}(\cdot), \mathrm{d}_{\mathrm{spd}}^{le}(\cdot)$, and $\mathrm{d}_{\mathrm{spd}}^{lc}(\cdot)$, given in Tab. 1. Given $\mathbf{P}, \mathbf{Q} \in \mathcal{S}_d^{++}$, we denote by $\mathrm{logm}(\cdot)$ and $\mathrm{expm}(\cdot)$ the matrix logarithm and exponential. $\mathscr{L}(\mathbf{P})$ means the Cholesky decomposition of $\mathbf{P}$, with $\mathscr{L}^{-1}(\cdot)$ denoting its inverse map. Besides, $\lfloor \mathscr{L}(\mathbf{P}) \rfloor$ denotes the strictly lower triangular part, $\mathbb{D}(\mathbf{P})$ signifies the diagonal part, and then $\psi_{\mathrm{LC}}(\mathbf{P}) = \lfloor \mathscr{L}(\mathbf{P}) \rfloor + \mathrm{logm}(\mathbb{D}(\mathbf{P}))$.

**Grassmannian geometries.** The Grassmannian manifold consists of all $q$-dimensional linear subspaces within $\mathbb{R}^d$. This study centers on the Orthonormal Basis (ONB) perspective, where a subspace is represented by $\mathbf{Y} \in \mathbb{R}^{d \times q}$ satisfying $\mathbf{Y}^\top \mathbf{Y} = \mathbf{I}_q$. We denote the set of such ONB representations as $\mathbf{Y} \in \widetilde{\mathcal{G}}(q, d)$. A point in $\widetilde{\mathcal{G}}(q, d)$ represents an equivalence class of orthonormal bases: $[\mathbf{Y}] = \{\widetilde{\mathbf{Y}} \mid \widetilde{\mathbf{Y}} = \mathbf{Y}\mathbf{O}, \mathbf{O} \in \mathrm{O}(q)\}$. By abuse of notation, we use $[\mathbf{Y}]$ or $\mathbf{Y}$ interchangeably. Under the ONB perspective, the Grassmannian manifold induces a nonreductive gyrovector structure [52]. The binary operation $\widetilde{\oplus}_{gr}$ [52] and gyro distance $\mathrm{d}_{gr}(\cdot)$ [19] for $\mathbf{U}, \mathbf{V} \in \widetilde{\mathcal{G}}(q, d)$ are defined in Tab. 1.

**SPSD geometries.** Let $\mathcal{S}_{d,q}^+$ denote the set of $d \times d$ SPSD matrices of rank $q \leq d$. Each $\mathbf{P} \in \mathcal{S}_{d,q}^+$ admits a canonical decomposition $\mathbf{P} = \mathbf{U}_P \mathbf{S}_P \mathbf{U}_P^\top$, where $\mathbf{U}_P \in \widetilde{\mathcal{G}}(q, d)$ and $\mathbf{S}_P \in \mathcal{S}_q^{++}$ [9, 10].

Table 1: Summary of the gyro additions and gyro distances over different manifolds.

| Manifold | Metric | Gyro addition | Gyro distance |
|---|---|---|---|
| SPD | AIM | $\mathbf{P} \oplus_{ai} \mathbf{Q} = \mathbf{P}^{\frac{1}{2}} \mathbf{Q} \mathbf{P}^{\frac{1}{2}}$ | $\left\| \log m \left( \mathbf{Q}^{-\frac{1}{2}} \mathbf{P} \mathbf{Q}^{-\frac{1}{2}} \right) \right\|_{\mathbf{F}}$ |
| | LEM | $\mathbf{P} \oplus_{le} \mathbf{Q} = \exp m(\log m(\mathbf{P}) + \log m(\mathbf{Q}))$ | $\left\| \log m(\mathbf{P}) - \log m(\mathbf{Q}) \right\|_{\mathbf{F}}$ |
| | LCM | $\mathbf{P} \oplus_{lc} \mathbf{Q} = \mathscr{L}^{-1} \left( \lfloor \mathscr{L}(\mathbf{P}) \rfloor + \lfloor \mathscr{L}(\mathbf{Q}) \rfloor + \mathbb{D}(\mathscr{L}(\mathbf{P}))\mathbb{D}(\mathscr{L}(\mathbf{Q})) \right)$ | $\left\| \psi_{\mathrm{LC}}(\mathbf{P}) - \psi_{\mathrm{LC}}(\mathbf{Q}) \right\|_{\mathbf{F}}$ |
| Grassmannian | ONB perspective | $\mathbf{U} \widetilde{\oplus}_{gr} \mathbf{V} = \exp m([\mathrm{Log}^{gr}_{\mathbf{I}_{d,q}}(\mathbf{U}\mathbf{U}^\top), \mathbf{I}_{d,q}])\mathbf{V}$ | $\begin{array}{c} \| \arccos(\Sigma) \| \\ \mathbf{U}^\top \mathbf{V} \overset{\mathrm{SVD}}{:=} \mathbf{O}\Sigma\mathbf{R}^\top \end{array}$ |
| SPSD | $(g_{\mathrm{gr}}, \lambda g_{\mathrm{spd}})$ | $(\mathbf{U}_P, \mathbf{S}_P) \oplus_{psd,g} (\mathbf{U}_Q, \mathbf{S}_Q) = (\mathbf{U}_P \widetilde{\oplus}_{gr} \mathbf{U}_Q, \mathbf{S}_P \oplus_g \mathbf{S}_Q)$ | $\mathrm{d}_{\mathrm{gr}}(\mathbf{U}_P, \mathbf{U}_Q) + \lambda \mathrm{d}^g_{\mathrm{spd}}(\mathbf{S}_P, \mathbf{S}_Q)$ |

Following Nguyen et al. [53], we represent $\mathbf{P}$ in the product space $\mathcal{G}(q,d) \times \mathcal{S}_q^{++}$, as detailed in App. D.3. By equipping $\mathcal{S}_q^{++}$ with a Riemannian metric $g \in \{ai, le, lc\}$, the space $\mathcal{S}_{d,q}^+$ naturally inherits a corresponding gyrovector structure [53]. The induced binary operation $\oplus_{psd,g}$ [53] and the related gyro distance $\mathrm{d}_{psd,g}(\cdot)$ [19] are defined in Tab. 1, where $\lambda > 0$.

**WFMs.** The WFM [25] of points $\{\mathbf{P}_{i...N}\}$ is the point $\mathbf{S} \in \mathcal{M}$ that minimizes the weighted sum of squared distances to all points $\{\mathbf{P}_{i...N}\}$. Given weights $\{w_{1...N}\}$ satisfying the convexity constraint, *i.e.*, $\forall i, w_i > 0$ and $\sum_i w_i = 1$, the WFM is expressed as:

$$\mathrm{WFM}(\{w_i\}, \{\mathbf{P}_i\}) = \underset{\mathbf{S} \in \mathcal{M}}{\arg\min} \sum_{i=1}^{N} w_i \, \mathrm{d}^2 (\mathbf{P}_i, \mathbf{S}), \tag{1}$$

where $\mathrm{d}(\mathbf{P}_i, \mathbf{S})$ is the distance between $\mathbf{S}$ and $\mathbf{P}_i$. On Riemannian manifolds, WFMs uniquely exists when samples are locally distributed [2], which is detailed in App. D.2.1. In this paper, we always assume WFMs are well-defined.

# 3 Proposed method

Inspired by the success of attention mechanism [70, 32], recent studies have extended attention models to non-Euclidean settings, including hyperbolic [26], SPD [55], and Grassmannian [72] manifolds. However, these approaches typically rely on geometry-specific operations, such as BiMap layer [33] for SPD manifolds or FrMap layer [34] for Grassmannian manifolds, which limits their applicability and generalization across manifolds. In contrast, our proposed GyroAtt framework leverages gyro structures to provide a unified and principled formulation of attention across matrix manifolds. To motivate our design, we first revisit prior manifold-based attention mechanisms and summarize their core components in Tab. 2, before introducing our generalized gyro-based formulation.

## 3.1 Revisiting attention mechanisms on different geometries

Despite differences in underlying geometries, self-attention mechanisms generally follow a common three-stage pipeline: 1) feature transformations to compute query ($\mathbf{q}_i$), key ($\mathbf{k}_i$), and value ($\mathbf{v}_i$); 2) similarity computations, often based on distances or inner products; 3) weighted aggregations of the values ($\mathbf{r}_i$). Tab. 2 summarizes these components across geometry-aware attention models.

**Euclidean.** Standard Transformers [70] generate $\mathbf{q}_i$, $\mathbf{k}_i$, and $\mathbf{v}_i$ through linear projections $\mathrm{Linear}(\cdot)$, compute attention weights using scaled dot-product $\mathrm{Softmax}(\langle \mathbf{q}_i, \mathbf{k}_j \rangle / \sqrt{d_k})$, and obtain outputs via $\mathbf{r}_i = \sum_{j=1}^{N} \mathcal{A}_{ij} \mathbf{v}_j$, where $N$ signifies the number of value tokens.

**Hyperbolic.** HAN [26] extends attention to the hyperbolic space using the Klein ($\mathbb{K}^d$) and hyperboloid ($\mathbb{H}^d$) models. Euclidean features are first projected onto the manifold via $\pi_{\mathbb{R} \to \mathbb{K}}$ and $\pi_{\mathbb{R} \to \mathbb{H}}$. Attention scores are computed by $-\beta \, \mathrm{d}(\mathbf{q}_i, \mathbf{k}_j) - c$, where $\beta > 0$ and $c$ are learnable parameters. Value aggregations are performed using the Einstein midpoint [66].

**SPD.** MAtt [55] formulates attention on the SPD manifolds under LEM. Queries and keys are obtained via the BiMap function [33], attention scores are measured using the LEM-based geodesic distance, and LEM-based WFM is employed for value aggregation.

**Grassmannian.** GDLNet [72] adapts attention to the Grassmannian manifolds by employing FrMap and ReOrth layers for data transformation and activation [34]. Attention scores are computed based on the projection distance, while the extrinsic WFM [62] is employed for value aggregation.

In summary, the above manifold attention approaches are inherently tied to particular manifolds or metrics, limiting their direct generalization across different geometries.

Table 2: Summary of attention based on different geometries, where $f_s(\cdot)$ denotes the softmax.

| Method | Geometries | Transformations | $\mathrm{d}(\mathbf{q}_i, \mathbf{k}_i)$ | Attention $\mathcal{A}_{ij}$ | Aggregation $\mathbf{r}_i(\mathbf{R}_i)$ |
|---|---|---|---|---|---|
| Transformer [70] | Euclidean | $\mathrm{Linear}(\mathbf{x}_i)$ | $\|\mathbf{q}_i - \mathbf{k}_j\|_{\mathrm{F}}$ | $f_s(\langle \mathbf{q}_i, \mathbf{k}_j \rangle / \sqrt{d_k})$ | Arithmetic mean $\sum_j^N \mathcal{A}_{ij}\mathbf{v}_j$ |
| HAN [26] | Hyperbolic | $\pi_{\mathbb{R}\to\mathbb{H}}(\mathrm{Linear}(\mathbf{x}_i))$ $\pi_{\mathbb{R}\to\mathbb{K}}(\mathrm{Linear}(\mathbf{x}_i))$ | $\mathrm{arccosh}(-\langle \mathbf{q}_i, \mathbf{k}_i \rangle_M)$ | $f_s(-\beta\,\mathrm{d}(\mathbf{q}_i, \mathbf{k}_j) - c)$ | Einstein midpoint $\sum_j^N \left[\frac{\mathcal{A}_{ij}\gamma(\mathbf{v}_j)}{\sum_l^N \mathcal{A}_{il}\gamma(\mathbf{v}_l)}\right]\mathbf{v}_j$ |
| MAtt [55] | SPD under LEM | $\mathbf{W}\mathbf{X}_i\mathbf{W}^\top$ | $\|\mathrm{logm}(\mathbf{Q}_i) - \mathrm{logm}(\mathbf{K}_i)\|_{\mathrm{F}}$ | $f_s\left((1 + \log(1 + \mathrm{d}(\mathbf{Q}_i, \mathbf{K}_j)))^{-1}\right)$ | LEM-based WFM $\mathrm{expm}\left(\sum_j^N \mathcal{A}_{ij}\mathrm{logm}(\mathbf{V}_j)\right)$ |
| GDLNet [72] | Grassmannian under ONB | $\mathrm{ReOrth}(\mathbf{W}\mathbf{X}_i)$ | $\|\mathbf{Q}_i\mathbf{Q}_i^\top - \mathbf{K}_j\mathbf{K}_j^\top\|_{\mathrm{F}}$ | $f_s\left((1 + \log(1 + \mathrm{d}(\mathbf{Q}_i, \mathbf{K}_j)))^{-1}\right)$ | Extrinsic WFM $\Phi^{-1}\left(\sum_j^N \mathcal{A}_{ij}\Phi(\mathbf{V}_j)\right)$ |
| GyroAtt (Ours) | Gyrovector spaces (SPD, SPSD, Grassmann) | Homomorphism Eq. (4) | Gyro distance | $f_s\left((1 + \log(1 + \mathrm{d}(\mathbf{Q}_i, \mathbf{K}_j)))^{-1}\right)$ | WFM |

## 3.2 Attention mechanisms over gyrovector spaces

In this part, we extend the basic attention operations illustrated in Tab. 2 to gyrovector spaces: 1) transformation through gyro homomorphisms, which preserves the gyrovector structure; 2) distance-based similarity computation; 3) value aggregation via gyro distance-based WFM.

**Definition 3.1 (Gyro Homomorphisms).** Let $(\mathcal{M}, \oplus_\mathcal{M}, \otimes_\mathcal{M})$ and $(\mathcal{N}, \oplus_\mathcal{N}, \otimes_\mathcal{N})$ be two (nonreductive) gyrovector spaces, $\forall \mathbf{A}, \mathbf{B} \in \mathcal{M}$, and $\forall t \in \mathbb{R}$, the map $\mathrm{hom}(\cdot) : (\mathcal{M}, \oplus_\mathcal{M}, \otimes_\mathcal{M}) \to (\mathcal{N}, \oplus_\mathcal{N}, \otimes_\mathcal{N})$ is a (nonreductive) gyrovector space homomorphism if it satisfies:

$$\mathrm{hom}(\mathbf{A} \oplus_\mathcal{M} \mathbf{B}) = \mathrm{hom}(\mathbf{A}) \oplus_\mathcal{N} \mathrm{hom}(\mathbf{B}), \quad \mathrm{hom}(t \otimes_\mathcal{M} \mathbf{A}) = t \otimes_\mathcal{N} \mathrm{hom}(\mathbf{A}). \tag{2}$$

If we only consider (nonreductive) gyrogroups, $(\mathcal{M}, \oplus_\mathcal{M})$ and $(\mathcal{N}, \oplus_\mathcal{N})$, a map $\mathrm{hom}(\cdot) : (\mathcal{M}, \oplus_\mathcal{M}) \to (\mathcal{N}, \oplus_\mathcal{N})$ satisfying Eq. (35) is called a (nonreductive) gyrogroup homomorphism, which has been introduced by Suksumran and Wiboonton [63]. By abuse of notations, we call the above homomorphisms collectively gyro homomorphisms. The concept of gyro homomorphisms naturally generalizes linear maps from vector spaces to gyrovector spaces. Recall that a linear map $\mathrm{Linear}(\cdot) : \mathbb{R}^n \to \mathbb{R}^m$ is a homomorphism of vector spaces, meaning it preserves both vector addition and scalar multiplication: for any $\mathbf{z}_1, \mathbf{z}_2 \in \mathbb{R}^n$ and $t \in \mathbb{R}$, it satisfies:

$$\mathrm{Linear}(\mathbf{z}_1 + \mathbf{z}_2) = \mathrm{Linear}(\mathbf{z}_1) + \mathrm{Linear}(\mathbf{z}_2), \quad \mathrm{Linear}(t\mathbf{z}_1) = t\,\mathrm{Linear}(\mathbf{z}_1). \tag{3}$$

Therefore, we use $\mathrm{hom}(\cdot)$ for feature transformation. While the above offers an algebraic definition, a natural question arises: *how can such mappings be identified on manifolds?*

**Theorem 3.2 (Sufficient condition for gyro homomorphisms).** [↓] *Let $\mathcal{M}$ be a Riemannian homogeneous space with isometry group $G$. If $(\mathcal{M}, \oplus, \otimes)$ forms a gyrovector space and an isometry $f \in G$ fixes the identity element, i.e., $f(e) = e$, then $f$ is a gyro homomorphism.*

This theorem provides a way to identify gyro homomorphisms from homogeneous space isometries. Then, we calculate the correlation between $\mathbf{Q}_i$ and $\mathbf{K}_j$ using their gyro distance, and map $\mathrm{d}(\mathbf{Q}_i, \mathbf{K}_j)$ to a valid attention score, as defined in Eq. (5). For feature aggregation, we resort to WFM based on the gyro distance. Specifically, given a set of input data points $\{\bar{\mathbf{X}}_{i...N} \in \mathcal{M}\}$, the key operations of GyroAtt are listed below:

$$\mathbf{Q}_i = \mathrm{hom}(\mathbf{X}_i), \quad \mathbf{K}_i = \mathrm{hom}(\mathbf{X}_i), \quad \mathbf{V}_i = \mathrm{hom}(\mathbf{X}_i), \qquad \text{(Transformation)} \tag{4}$$

$$\mathcal{A}_{ij} = \mathrm{Softmax}\left((1 + \log(1 + \mathrm{d}(\mathbf{Q}_i, \mathbf{K}_j)))^{-1}\right), \qquad \text{(Attention)} \tag{5}$$

$$\mathbf{R}_i = \mathrm{WFM}\left(\mathcal{A}_i, \mathbf{V}_{i...N}\right). \qquad \text{(Aggregation)} \tag{6}$$

Here, $\mathcal{A}_i$ denote the $i$-th row of $\mathcal{A}$, and $\mathbf{R}_i$ is the resulting data under the manifold-valued weighted average (*i.e.*, WFM) between $\mathcal{A}_i$ and $\mathbf{V}_{i...N}$. While WFM offers a principled approach for aggregating manifold-valued data, its output $\mathbf{R}_i$ is inherently restricted to the geodesic convex hull of the input points. This constraint may limit the expressiveness of the resulting representations. To mitigate this limitation, we introduce a bias term and a nonlinear transformation following the aggregation step:

$$\phi(\mathbf{R}_i) = \sigma(\mathbf{B} \oplus \mathbf{R}_i), \tag{7}$$

where $\mathbf{B}$ is a learnable bias parameter, and $\sigma$ represents a power-based activation function.

So far, all ingredients are in place to build attention over gyrovector spaces, as shown in Alg. 4.

# 4 Gyro attention mechanisms on matrix manifolds

This section presents our GyroAtt framework (Alg. 4) across various matrix gyrovector spaces, including three SPD gyro spaces, one Grassmannian gyro space, and three SPSD gyro spaces.

## 4.1 Gyro attention mechanism over SPD manifolds

As summarized in Tab. 1, three SPD gyrovector spaces are induced by the AIM, LEM, and LCM, respectively. The corresponding gyro distance (used for attention calculation) and gyro addition (used for biasing) have been well studied in prior works [5, 56, 45]. Therefore, our focus is on defining the gyro homomorphisms, WFMs, and activation for these geometries. We begin by deriving the explicit forms of the gyro homomorphisms corresponding to each SPD geometry.

**Theorem 4.1 (SPD Homomorphisms).** [↓] *Given* $\mathbf{P} \in \mathcal{S}_d^{++}$, *let* $\hom_g : \mathcal{S}_d^{++} \to \mathcal{S}_d^{++}$ *be defined as*

$$\hom_g(\cdot) : \mathbf{P} \mapsto \begin{cases} \mathbf{O}\mathbf{P}\mathbf{O}^\top, & g = ai, \ \mathbf{O} \in \mathrm{O}(d), \\ \mathrm{expm}(\mathbf{M}\,\mathrm{logm}(\mathbf{P})\mathbf{M}^\top), & g = le, \ \mathbf{M} \in \mathbb{R}^{d \times d}, \\ \mathscr{L}^{-1}\big(\lfloor L(\mathbf{P}) \rfloor + \exp\big(\mathbb{D}\big(L(\mathbf{P})\big)\big)\big), & g = lc, \ \mathbf{M} \in \mathbb{R}^{d \times d}, \end{cases} \tag{8}$$

*where* $L(\mathbf{P}) = \mathbf{M}\big(\lfloor \mathscr{L}(\mathbf{P}) \rfloor + \lfloor \mathscr{L}(\mathbf{P}) \rfloor^\top + \mathbb{D}\big(\mathscr{L}(\mathbf{P})\big)\big)\mathbf{M}^\top$. *For each metric* $g \in \{ai, le, lc\}$, $\hom_g(\cdot)$ *is a gyro homomorphism with respect to* $(\mathcal{S}_d^{++}, \oplus_g, \otimes_g)$. *Moreover, if* $\mathbf{M}$ *is orthonormal, the LEM-based homomorphism is identical to the AIM one.*

**Transformation.** As shown in Thm. 4.1, the forms of $\hom_{le}(\cdot)$ and $\hom_{lc}(\cdot)$ follow from the flat structures of their geometries. When a manifold is flat, its gyro homomorphism can be directly derived. The SPD manifold under LEM is flat in the logarithmic domain, where linear mappings act as valid homomorphisms. Orthogonal constraints can improve network generalization by imposing implicit regularization [30]. Therefore, we impose orthogonality on $\mathbf{M}$ in both $\hom_{lc}(\cdot)$ and $\hom_{le}(\cdot)$.

**WFMs.** The WFMs under LEM and LCM admit closed-form solutions, whereas the one under AIM requires iterative computation via the Karcher flow algorithm [38]. These are reviewed in App. D.2.2.

**Activation.** As demonstrated by Chen et al. [17, Fig. 1] and Chen et al. [15, Sec. 5.1], the matrix power can deform the latent SPD geometries. As a consequence, we use matrix power as the activation function to activate the underlying Riemannian geometry.

## 4.2 Gyro attention mechanisms over Grassmannian manifolds

We implement GyroAtt on the Grassmannian nonreductive gyrovector spaces under the ONB perspective. The gyro distance and gyro addition are shown in Tab. 1. Similar to the SPD gyro spaces, we use gyro homomorphism for transformation and WFM for aggregation. As shown by Nguyen and Yang [52, Sec. 2.3.2], the Grassmannian gyro addition can be viewed as a non-linear activation. Therefore, we do not use additional activation before the Grassmannian gyro biasing. In the following, we discuss gyro homomorphism and WFM over the Grassmannian manifold.

**Theorem 4.2 (Grassmannian Homomorphisms).** [↓] *Let* $\mathbf{U} \in (\widetilde{\mathcal{G}}(q, d), \widetilde{\oplus}_{gr}, \widetilde{\otimes}_{gr})$, *and* $\mathbf{O} = \begin{bmatrix} \mathbf{O}_q & 0 \\ 0 & \mathbf{O}_{d-q} \end{bmatrix} \in \mathbb{R}^{d,d}$, *where* $\mathbf{O}_q \in \mathbb{R}^{q \times q}$ *and* $\mathbf{O}_{d-q} \in \mathbb{R}^{(d-q) \times (d-q)}$ *are orthonormal. The map* $\hom_{gr}(\cdot) : (\widetilde{\mathcal{G}}(q, d), \widetilde{\oplus}_{gr}, \widetilde{\otimes}_{gr}) \to (\widetilde{\mathcal{G}}(q, d), \widetilde{\oplus}_{gr}, \widetilde{\otimes}_{gr})$ *is a gyro homomorphism defined by*

$$\hom_{gr}(\mathbf{U}) = \mathbf{O}\mathbf{U}. \tag{9}$$

Eq. (9) plays the role of Grassmannian feature transformation. For the weighted aggregation, since the WFM on the Grassmannian manifold does not admit a closed-form solution, we employ the Karcher flow algorithm [1, 38]. More details are provided in App. D.2.3.

## 4.3 Gyro attention mechanisms over SPSD manifolds

As introduced in Sec. 2, any $\mathbf{P} \in \mathcal{S}_{d,q}^+$ can be canonically represented as a pair $(\mathbf{U}_P, \mathbf{S}_P) \in \widetilde{\mathcal{G}}(q, d) \times \mathcal{S}_q^{++}$ within the structured product space. According to Tab. 1, both the distance and gyro addition operations in this space are defined component-wisely over the Grassmannian and SPD subspaces. Therefore, implementing GyroAtt in the SPSD gyrovector space reduces to specifying the gyro homomorphisms, WFMs, and activation for the corresponding product structure.

Table 3: Key operators of GyroAtt over different matrix gyrovector spaces.

| Manifold | SPD | | | Grassmannian | SPSD |
|---|---|---|---|---|---|
| Metric | AIM | LEM | LCM | ONB perspective | $(g_{gr}, \lambda g_{spd})$ |
| Homomorphism | $\mathbf{OPO}^\top$ | $\mathrm{expm}(\mathbf{M}\,\mathrm{logm}(\mathbf{P})\mathbf{M}^\top)$ | $\mathscr{L}^{-1}\big(\lfloor L(\mathbf{P})\rfloor + \exp\big(\mathbb{D}(L(\mathbf{P}))\big)\big)$ | $\mathbf{OU}$ | $(\mathrm{hom}_{gr}(\mathbf{U}_P), \mathrm{hom}_g(\mathbf{S}_P))$ |
| WFM | Karcher flow Alg. 1 | Closed-form Eq. (15) | Closed-form Eq. (16) | Karcher flow Alg. 2 | $(\mathrm{WFM}_{spd}, \mathrm{WFM}_{gr})$ |
| Bias and Non-linearity | $(\mathbf{B}_{spd} \oplus_g \mathbf{R}_i)^p$ | | | $\mathbf{B}_{gr}\widetilde{\oplus}_{gr}\mathbf{R}_i$ | $(\mathbf{B}_{gr}\widetilde{\oplus}_{gr}\mathbf{U}_{R_i}, (\mathbf{B}_{spd} \oplus_{spd} \mathbf{S}_{R_i})^p)$ |

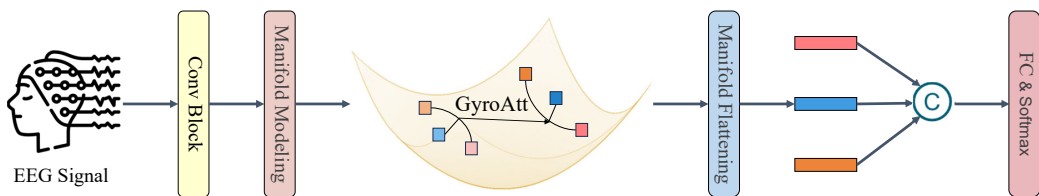

Figure 1: The GyroAtt network consists of three modules: A feature extraction module that applies convolution and manifold modeling to transform EEG signals into manifold-valued features; a gyro attention module that captures long-range dependencies among features; and a classification module that implements decision making with the flattened manifold-valued data.

**Theorem 4.3** (**SPSD Homomorphisms**). [↓] *Let* $g \in \{ai, le, lc\}$, *and* $(\mathbf{U}_P, \mathbf{S}_P) \in (\widetilde{\mathcal{G}}(q,d) \times \mathcal{S}_q^{++}, \oplus_{psd,g}, \otimes_{psd,g})$. *The map* $\mathrm{hom}_{psd,g}(\cdot) : (\widetilde{\mathcal{G}}(q,d) \times \mathcal{S}_q^{++}, \oplus_{psd,g}, \otimes_{psd,g}) \rightarrow (\widetilde{\mathcal{G}}(q,d) \times \mathcal{S}_q^{++}, \oplus_{psd,g}, \otimes_{psd,g})$ *is a gyro homomorphism defined by*

$$\mathrm{hom}_{psd,g}(\mathbf{U}_P, \mathbf{S}_P) = (\mathrm{hom}_{gr}(\mathbf{U}_P), \mathrm{hom}_g(\mathbf{S}_P)). \tag{10}$$

For aggregation, we use the WFM induced by the product geometry, as detailed in App. D.2.4. The bias and non-linear activation are also formulated in the product space:

$$\phi_{psd}(\mathbf{U}_{R_i}, \mathbf{S}_{R_i}) = (\mathbf{B}_{gr}\widetilde{\oplus}_{gr}\mathbf{U}_{R_i}, (\mathbf{B}_{spd} \oplus_{spd} \mathbf{S}_{R_i})^p), \tag{11}$$

where $\mathbf{B}_{gr} \in \widetilde{\mathcal{G}}(q,d)$ and $\mathbf{B}_{spd} \in \mathcal{S}_d^{++}$.

In summary, our GyroAtt framework comprises several basic operations. First, the mapping $\mathrm{hom}(\cdot)$ generates $\mathbf{Q}_i$, $\mathbf{K}_i$, and $\mathbf{V}_i$. Then, the attention score is computed using gyro distances between queries and keys, followed by the WFM-based aggregation on the values $\mathbf{V}_i$. Finally, the bias and non-linear activation enhance model expressiveness. GyroAtt offers a unified framework that supports multiple manifolds and metrics, demonstrating its superior generality and flexibility. Tab. 3 summarizes all the key ingredients for computing GyroAtt on the SPD, Grassmannian, and SPSD manifolds.

## 5 Experiments

Following prior works [55, 41], we evaluate the proposed GyroAtt on EEG decoding across four benchmarking datasets: BNCI2014001 [24], BNCI2015001 [64], MAMEM-SSVEP-II [54], and BCI-ERN [47] in this paper. For BNCI2014001 and BNCI2015001, we conduct both inter-session and inter-subject evaluations. In the inter-session evaluation, models are trained exclusively on data from the target subject. Additionally, the balanced accuracy, calculated as the average recall across all classes, is used as the primary metric [41]. For MAMEM-SSVEP-II, the overall accuracy is used, while for BCI-ERN, we report the Area Under the Curve (AUC) to address class imbalance. In the experiments, the first four sessions of each subject are used for training, with one for validation, and the fifth for testing. Further details regarding datasets and preprocessings are provided in App. E.1.

**Implementation details.** As shown in Fig. 1, the GyroAtt network consists of three components: a feature extraction module, a gyro attention module, and a classification head. In the feature extraction module, two convolutional blocks are first applied to the EEG signals to extract low-redundancy features, followed by pyramid-like temporal segmentation into $s$ non-overlapping subparts, each producing a covariance matrix. For GyroAtt-SPD, these covariance matrices $\mathbf{X}_i$ serve directly as inputs to the subsequent layers. In GyroAtt-SPSD and GyroAtt-Gr, each covariance matrix is transformed into its canonical form $(\mathbf{U}_X^i, \mathbf{S}_X^i)$ using Alg. 3, mapping them into the structure space $\widetilde{\mathcal{G}}(q,d) \times \mathcal{S}_q^{+i}$. Here, $\mathbf{U}_X^i$ is used as the input for GyroAtt-Gr, while both $\mathbf{U}_X^i$ and $\mathbf{S}_X^i$ are used

Table 4: The results of different methods on BNCI2014001 and BNCI2015001. Riemannian manifold attention models are highlighted in yellow. The top three results are marked in **red**, **blue**, and **cyan**.

| Manifold | Method | BNCI2014001 | | BNCI2015001 | |
|---|---|---|---|---|---|
| | | Inter-session | Inter-subject | Inter-session | Inter-subject |
| Euclidean | FBCSP+SVM [4] | $60.6 \pm 4.9$ | $32.3 \pm 7.3$ | $81.5 \pm 4.4$ | $58.6 \pm 13.4$ |
| | EEGNet [44] | $41.8 \pm 5.8$ | $43.3 \pm 17.0$ | $72.4 \pm 8.4$ | $59.2 \pm 9.5$ |
| | ShConvNet [59] | $51.3 \pm 2.3$ | $42.2 \pm 16.2$ | $74.1 \pm 4.2$ | $58.7 \pm 5.8$ |
| | FBCSP+DSS+LDA [29] | $71.3 \pm 1.8$ | $48.3 \pm 14.3$ | $84.6 \pm 4.8$ | $67.7 \pm 14.3$ |
| SPD | TSM+SVM [6] | $61.8 \pm 4.1$ | $34.7 \pm 8.6$ | $75.7 \pm 5.1$ | $56.0 \pm 6.0$ |
| | FB+TSM+LR [42] | $69.8 \pm 4.8$ | $36.5 \pm 8.2$ | $80.9 \pm 6.0$ | $60.6 \pm 10.9$ |
| | URPA+MDM [57] | $59.5 \pm 2.7$ | $46.8 \pm 14.6$ | $79.2 \pm 4.6$ | $70.3 \pm 16.1$ |
| | SPDOT+TSM+SVM [74] | $66.8 \pm 3.8$ | $38.6 \pm 8.6$ | $77.5 \pm 2.9$ | $63.3 \pm 8.1$ |
| | TSMNet [41] | $69.0 \pm 3.6$ | $51.6 \pm 16.5$ | $85.8 \pm 4.3$ | $77.0 \pm 13.7$ |
| | Graph-CSPNet [36] | $71.9 \pm 13.3$ | $45.2 \pm 9.3$ | $79.8 \pm 14.6$ | $64.2 \pm 13.4$ |
| SPD Grassmann | MAtt [55] | $66.5 \pm 8.9$ | $45.3 \pm 11.3$ | $80.8 \pm 14.8$ | $63.1 \pm 10.1$ |
| | GDLNet [72] | $58.1 \pm 8.9$ | $46.3 \pm 5.1$ | $76.9 \pm 13.6$ | $63.3 \pm 14.2$ |
| SPD | GyroAtt-SPD-AIM | $75.4 \pm 7.1$ | $53.1 \pm 14.8$ | $86.2 \pm 4.5$ | $77.9 \pm 13.0$ |
| | GyroAtt-SPD-LEM | $75.3 \pm 6.5$ | $52.3 \pm 14.1$ | $85.7 \pm 5.5$ | $76.6 \pm 13.7$ |
| | GyroAtt-SPD-LCM | $74.2 \pm 7.8$ | $52.4 \pm 15.6$ | $84.7 \pm 6.6$ | $75.5 \pm 13.8$ |
| Grassmann | GyroAtt-Gr | $72.5 \pm 7.3$ | $52.1 \pm 14.2$ | $85.0 \pm 7.7$ | $75.3 \pm 13.7$ |
| SPSD | GyroAtt-SPSD-AIM | $72.9 \pm 7.1$ | $52.4 \pm 15.6$ | $84.7 \pm 6.6$ | $75.5 \pm 13.8$ |
| | GyroAtt-SPSD-LEM | $72.8 \pm 6.9$ | $50.5 \pm 13.2$ | $85.3 \pm 5.3$ | $76.0 \pm 14.1$ |
| | GyroAtt-SPSD-LCM | $72.9 \pm 6.7$ | $51.7 \pm 13.1$ | $85.1 \pm 4.8$ | $74.9 \pm 12.6$ |

for GyroAtt-SPSD. We employ the corresponding GyroAtt block, as shown in Alg. 4, to capture long-range dependencies between different feature regions on the manifolds. In the classification module, we first flatten the manifold-valued representations by projecting them onto a tangent (or Euclidean) space, followed by vectorization. For GyroAtt-SPD, we follow previous works [71, 16] to apply matrix power normalization to the output matrix $\mathbf{P}$ from the GyroAtt block, defined as $\psi_\theta(\mathbf{P}) = \frac{1}{\theta}\mathbf{P}^\theta$ with $\theta > 0$ and $\mathbf{P} \in \mathcal{S}_d^{++}$. The scaling factor $\frac{1}{\theta}$ ensures gradient stability during optimization. For GyroAtt-Gr, we project each element $\mathbf{Y}_i \in \mathcal{G}(q, d)$ into a Euclidean space using the operator $\Phi(\mathbf{Y}_i) = \mathbf{Y}_i\mathbf{Y}_i^\top$. For GyroAtt-SPSD, both $\mathbf{U}_X^i$ and $\mathbf{S}_X^i$ are processed accordingly within the classification module. Across all three models, the resulting matrices are vectorized, concatenated, and passed through a fully connected layer followed by a Softmax function for classification. For parameter optimization and detailed implementations, please refer to Apps. E.2.1 and E.2.2.

**Main results.** We evaluated the proposed GyroAtt framework on four EEG classification datasets, with 10-fold cross-validation results summarized in Tabs. 4 and 5. Our models—GyroAtt-SPD/Gr/SPSD are compared against state-of-the-art baselines. The most effective manifold choice in GyroAtt varies by dataset. To be specific, GyroAtt-SPSD-LCM achieves best accuracy on the MAMEM and BCI-ERN datasets, surpassing GDLNet by **3.2%** and **0.9%**, respectively. On the BNCI2014001 and BNCI2015001 datasets, GyroAtt-SPD-AIM delivers the highest scores, outperforming TSMNet by **6.4%**, **1.5%**, **0.4%**, and **0.9%**, respectively across different evaluation settings. Notably, GyroAtt-Gr consistently ranks second across all datasets and outperforms GDLNet in every case. These experimental observations highlight the generality and effectiveness of our GyroAtt. The superior performance of GyroAtt can be attributed to the underlying geometry-

Table 5: Results on the MAMEM-SSVEP-II (MAMEM) and BCI-ERN datasets. Manifold attention models are highlighted in yellow. The top three results are marked in **red**, **blue**, and **cyan**.

| Manifold | Method | MAMEM | BCI-ERN |
|---|---|---|---|
| Euclidean | EEGNet [44] | $53.7 \pm 7.2$ | $74.3 \pm 2.5$ |
| | ShallowCNet [59] | $56.9 \pm 6.7$ | $71.9 \pm 2.6$ |
| | EEG-TCNet [35] | $55.5 \pm 7.7$ | $77.1 \pm 2.5$ |
| | FBCNet [46] | $53.1 \pm 5.7$ | $60.5 \pm 3.1$ |
| | TCNet-Fusion [48] | $45.0 \pm 6.6$ | $70.5 \pm 2.9$ |
| | MBEEGSE [3] | $56.5 \pm 7.3$ | $75.5 \pm 2.3$ |
| SPD Grassmann | MAtt [55] | $65.2 \pm 3.1$ | $75.7 \pm 2.2$ |
| | GDLNet [72] | $65.5 \pm 2.9$ | $78.2 \pm 2.5$ |
| SPD | GyroAtt-SPD-AIM | $66.3 \pm 2.2$ | $75.8 \pm 3.3$ |
| | GyroAtt-SPD-LEM | $66.2 \pm 2.5$ | $76.1 \pm 4.2$ |
| | GyroAtt-SPD-LCM | $65.1 \pm 2.5$ | $75.4 \pm 3.7$ |
| Grassmann | GyroAtt-Gr | $67.1 \pm 1.6$ | $78.4 \pm 1.4$ |
| SPSD | GyroAtt-SPSD-AIM | $66.5 \pm 2.3$ | $78.2 \pm 1.9$ |
| | GyroAtt-SPSD-LEM | $66.5 \pm 2.9$ | $79.1 \pm 1.7$ |
| | GyroAtt-SPSD-LCM | $68.7 \pm 1.5$ | $78.4 \pm 1.6$ |

Table 6: The impact of matrix power activation on the accuracy of GyroAtt-SPD/SPSD.

| Geometry | Activation | BNCI2014001 | | BNCI2015001 | | MAMEM |
|---|---|---|---|---|---|---|
| | | Inter-session | Inter-subject | Inter-session | Inter-subject | |
| SPD | w/o | $74.8 \pm 6.7$ | $51.2 \pm 15.7$ | $85.5 \pm 5.0$ | $75.4 \pm 12.9$ | $64.3 \pm 2.4$ |
| | w/ | $\mathbf{75.4 \pm 7.1}$ | $\mathbf{53.1 \pm 14.8}$ | $\mathbf{86.2 \pm 4.5}$ | $\mathbf{77.9 \pm 13.0}$ | $\mathbf{66.3 \pm 2.2}$ |
| SPSD | w/o | $72.2 \pm 7.2$ | $49.2 \pm 13.7$ | $83.9 \pm 5.1$ | $74.2 \pm 14.4$ | $66.5 \pm 2.4$ |
| | w/ | $\mathbf{72.9 \pm 7.1}$ | $\mathbf{52.4 \pm 15.6}$ | $\mathbf{85.3 \pm 5.3}$ | $\mathbf{76.0 \pm 14.1}$ | $\mathbf{68.7 \pm 1.5}$ |

Table 8: Replacing $\mathrm{hom}(\cdot)$ and power activation with BiMap and ReEig in GyroAtt-SPD-LEM

| Transformation | Activation | BNCI2014001 | | BNCI2015001 | |
|---|---|---|---|---|---|
| | | Inter-session | Inter-subject | Inter-session | Inter-subject |
| BiMap | Power | $74.0 \pm 6.5$ | $52.3 \pm 15.0$ | $85.2 \pm 7.2$ | $77.2 \pm 13.2$ |
| Homomorphisms | ReEig | $75.1 \pm 6.3$ | $52.6 \pm 14.2$ | $85.9 \pm 5.3$ | $76.4 \pm 12.8$ |
| BiMap | ReEig | $73.6 \pm 6.8$ | $52.2 \pm 15.2$ | $85.4 \pm 7.8$ | $76.8 \pm 13.0$ |
| Homomorphisms | Power | $75.4 \pm 7.1$ | $53.1 \pm 14.8$ | $86.2 \pm 4.5$ | $77.9 \pm 13.0$ |

aware attention mechanism, which effectively captures long-range dependencies and spatiotemporal fluctuations inherent in EEG signals.

**Ablations on the matrix power-based nonlinear activation** $\sigma(\cdot)$. Tab. 6 illustrates the impact of the nonlinear activation (as defined in Tab. 3) on the accuracy of GyroAtt-SPD/SPSD. Generally, removing nonlinear activation results in performance degradation, *e.g.*, GyroAtt experiences a 2.0% and 2.4% accuracy drop on the MAMEM dataset. These experimental findings emphasize the importance of matrix power activation in enhancing model expressiveness by introducing nonlinearity into the metric space of the underlying feature manifold.

**Gyro distance *vs.* gyro inner product.** The Euclidean inner product is a natural similarity metric in flat space, but a global inner product is generally unavailable on curved manifolds. Hence, most manifold-attention models rely on distance-based similarity [26, 55, 72]. Gyrovector spaces are an exception because a *gyro inner product* can be defined in the tangent space

Table 7: Accuracy comparison under different measures.

| Methods | Similarity | BNCI2014001 Inter-session | BNCI2015001 Inter-session | MAMEM |
|---|---|---|---|---|
| GyroAtt-SPD | Gyro inner product | $74.7 \pm 6.8$ | $85.6 \pm 5.4$ | $63.9 \pm 3.2$ |
| | Gyro distance | $\mathbf{75.4 \pm 7.1}$ | $\mathbf{86.2 \pm 4.5}$ | $\mathbf{66.3 \pm 2.2}$ |
| GyroAtt-Gr | Gyro inner product | $72.4 \pm 7.3$ | $83.4 \pm 5.9$ | $65.7 \pm 3.1$ |
| | Gyro distance | $\mathbf{72.5 \pm 7.3}$ | $\mathbf{85.0 \pm 7.7}$ | $\mathbf{67.1 \pm 1.6}$ |
| GyroAtt-SPSD | Gyro inner product | $71.6 \pm 6.3$ | $83.3 \pm 5.4$ | $65.0 \pm 2.6$ |
| | Gyro distance | $\mathbf{72.9 \pm 6.2}$ | $\mathbf{85.3 \pm 5.3}$ | $\mathbf{68.7 \pm 1.5}$ |

at the identity [52][Defs. 2.9, 2.15]: $\langle \mathbf{P}, \mathbf{Q} \rangle^{\mathrm{gyr}} = \langle \mathrm{Log}_{\mathbf{I}}(\mathbf{P}), \mathrm{Log}_{\mathbf{I}}(\mathbf{Q}) \rangle_{\mathbf{I}}$. To evaluate its suitability, we replaced the distance-based similarity in Eq. (5) with gyro inner product and conducted an ablation study, as reported in Tab. 7. Across all datasets, the gyro distance consistently outperforms the inner product in accuracy. An essential reason is that the gyro inner product is defined within a single tangent space, which may not reflect the geometric structure faithfully. In contrast, gyro distance better preserves the intrinsic manifold geometry. The detailed implementations are found in App. E.6.

**GyroAtt-SPD-LEM *vs.* MAtt.** Although both GyroAtt-SPD-LEM and MAtt [55] implement attention mechanisms under the SPD manifold with LEM, GyroAtt-SPD-LEM differs from MAtt in two key components: linear transformation $\mathrm{hom}(\cdot)$, and the nonlinear activation $\sigma(\cdot)$. While BiMap and ReEig used in MAtt preserve SPD property, they are loosely connected to the underlying Riemannian geometry. Hence, we replace $\mathrm{hom}(\cdot)$ and $\sigma(\cdot)$ with Bimap and ReEig and made an ablation study, as reported in Tab. 8. Either replacement leads to a performance drop, while combining both further degrades accuracy. From a theoretical standpoint, gyro homomorphisms generalize linear maps and naturally reduce to them in the Euclidean setting. This indicates that our transformation layer is a principled extension of Euclidean attention. In contrast, the ReEig function merely rescales eigenvalues to ensure positive definiteness, lacking geometric interpretability. As shown in [17][Fig. 1], matrix power can deform any given Riemannian metric, effectively acting as a geometry-aware activation. These findings confirm that integrating gyro-based transformations and nonlinearities is not only theoretically justified but also empirically beneficial.

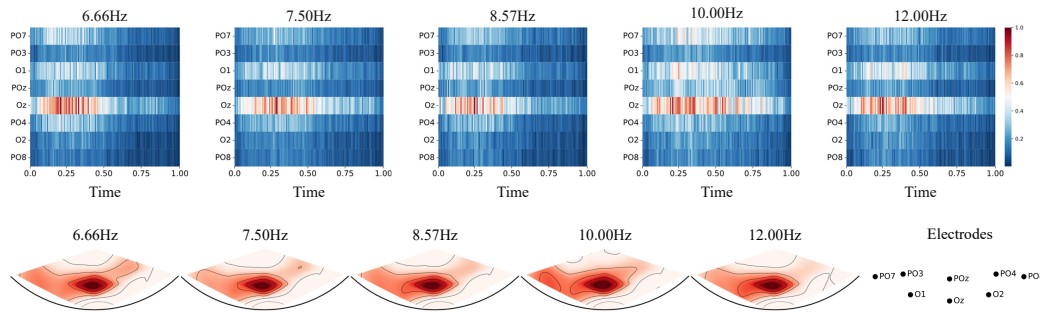

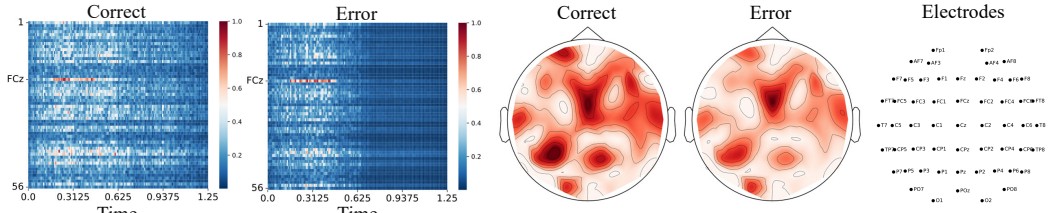

Figure 2: Spatial topomaps and gradient heatmaps of GyroAtt-SPSD on the MAMEM (subject S11) dataset. Strong responses appear around the Oz electrode, especially within 0–0.5s. In heatmaps, the x-axis denotes time and the y-axis denotes EEG channels.

Figure 3: Spatial topomaps and gradient heatmaps of GyroAtt-SPSD, showing 'correct' vs. 'error' trials. Strong activations consistently emerge around the FCz electrode within 0.1–0.4s.

**Training efficiency.** GyroAtt demonstrates superior or comparable efficiency to existing manifold attention networks such as MAtt and GDLNet. In particular, GyroAtt-SPD under LCM achieves efficiency on par with MAtt, benefiting from the computational advantages of the Cholesky decomposition. More detailed discussions and comparisons are provided in App. E.8.

Table 9: Running time (s/epoch) comparison.

| Method | BNCI2014001 | | BNCI2015001 | |
|---|---|---|---|---|
| | Inter-session | Inter-subject | Inter-session | Inter-subject |
| MAtt | 4.86 | 89.12 | 2.74 | 56.78 |
| GDLNet | 4.55 | 88.59 | 1.71 | 47.66 |
| GyroAtt-SPD-LCM | 4.11 | 87.44 | 2.42 | 49.86 |

**EEG interpretability analysis.** For the MAMEM dataset, as shown in Fig. 2, GyroAtt shows strong gradient responses around the Oz electrode across five stimulus frequencies, especially within 0–0.5s time window. This pattern aligns well with the established findings on the relationship between SSVEP signals and the Oz region [31, 28]. This is likely due to the Oz central location in the primary visual cortex, which leads to stronger evoked potentials and signal-to-noise ratio. On the BCI-ERN dataset, as illustrated in Fig. 3, gradient responses for both 'correct' and 'error' trials centered around the FCz electrode. This observation aligns with substantial empirical evidence that the anterior cingulate cortex, a central medial prefrontal cortex region connected to limbic and frontal areas, underlies ERN generation. Notably, consistent gradient responses around FCz were observed for both feedback types within the 0.1–0.4s interval, reinforcing Event-Related Potential (ERP) waveform differences between correct and error trials as reported in [27].

## 6 Conclusion

In this paper, we propose GyroAtt, a principled framework that generalizes the Euclidean attention mechanism to gyrovector spaces. Specifically, we adopt gyro homomorphism, gyro distance-based attention, and WFM as counterparts to the transformation, attention, and aggregation operations in Euclidean attention. Notably, we identify the concrete non-trivial expressions of gyro homomorphisms on different matrix gyro spaces. The principled construction of GyroAtt enables a direct assessment of the impact of geometry on a given task while keeping the network architecture constant. Extensive experiments and ablation studies on four EEG benchmarking datasets certify the effectiveness and flexibility of our proposed framework.

## Acknowledgments and Disclosure of Funding

This work was supported in part by the National Natural Science Foundation of China (62306127, 62020106012, 62332008), the Natural Science Foundation of Jiangsu Province (BK20231040, BK20221535), the Fundamental Research Funds for the Central Universities (JUSRP124015), the Postgraduate Research & Practice Innovation Program of Jiangsu Province (SJCX25_1319), the Key Project of Wuxi Municipal Health Commission (Z202318), the EU Horizon projects ELIAS (No. 101120237), the ELLIOT (No. 101214398), and the National Key R&D Program of China (2023YFF1105102, 2023YFF1105105).

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

## Appendix Contents

# A   Limitations

While several widely used manifolds, such as hyperbolic, SPD, Grassmannian, and SPSD manifolds, admit gyrovector structures, this property does not hold universally across all manifolds. As a result, our method is not directly applicable to manifolds lacking such a structure. Extending GyroAtt to accommodate more general non-gyrovector manifolds constitutes a promising direction for future research.

# B   Notations

To enhance readability, we summarize the main mathematical symbols and operators used throughout this paper in Tab. 10.

Table 10: Summary of notations.

| Notations | Explanation |
|---|---|
| $(G, \oplus)$ | A gyrogroup $G$ with a binary operation $\oplus$ |
| $\mathcal{S}_d^{++}$ | Space of $d \times d$ SPD matrices |
| $\mathcal{S}^d$ | Space of $d \times d$ symmetric matrices |
| $\mathcal{S}_{d,q}^{+}$ | Space of $d \times d$ SPSD matrices with rank $q \leq d$ |
| $\mathcal{G}(q, d)$ | Grassmannian in the projector perspective |
| $\widetilde{\mathcal{G}}(q, d)$ | Grassmannian in the ONB perspective |
| $\oplus_{ai}, \ominus_{ai}, \otimes_{ai}$ | Binary, inverse, and scalar multiplication operations in $\mathcal{S}_d^{++}$ under AIM |
| $\oplus_{le}, \ominus_{le}, \otimes_{le}$ | Binary, inverse, and scalar multiplication operations in $\mathcal{S}_d^{++}$ under LEM |
| $\oplus_{lc}, \ominus_{lc}, \otimes_{lc}$ | Binary, inverse, and scalar multiplication operations in $\mathcal{S}_d^{++}$ under LCM |
| $\widetilde{\oplus}_{gr}, \widetilde{\ominus}_{gr}, \widetilde{\otimes}_{gr}$ | Binary, inverse, and scalar multiplication operations in $\widetilde{\mathcal{G}}(q, d)$ |
| $\oplus_{gr}, \ominus_{gr}, \otimes_{gr}$ | Binary, inverse, and scalar multiplication operations in $\mathcal{G}(q, d)$ |
| $\oplus_{psd,g}, \ominus_{psd,g}, \otimes_{psd,g}$ | Binary, inverse, and scalar multiplication operations in $\widetilde{\mathcal{G}}(q, d) \times \mathcal{S}_d^{++}$ under metrics $g$ |
| $\langle \mathbf{P}, \mathbf{Q} \rangle^g$ | Inner product in $\mathcal{S}_d^{++}$ under metrics $g$ |
| $\langle \mathbf{U}, \mathbf{V} \rangle^{gr}$ | Inner product in $\widetilde{\mathcal{G}}(q, d)$ |
| $\langle (\mathbf{U}_P, \mathbf{S}_P), (\mathbf{U}_Q, \mathbf{S}_Q) \rangle^{psd,g}$ | Inner product in $\widetilde{\mathcal{G}}(q, d) \times \mathcal{S}_d^{++}$ under metrics $g$ |
| $\lVert \ominus_g \mathbf{P} \oplus_g \mathbf{Q} \rVert_g^{spd}$ | the gyrodistance in $\mathcal{S}_d^{++}$ under metrics $g$ |
| $\lVert \widetilde{\ominus}_{gr} \mathbf{U} \widetilde{\oplus}_{gr} \mathbf{V} \rVert^{gr}$ | the gyrodistance in $\widetilde{\mathcal{G}}(q, d)$ |
| $\lVert (\widetilde{\ominus}_{gr} \mathbf{U}_P \widetilde{\oplus}_{gr} \mathbf{U}_Q, \ominus_g \mathbf{S}_P \oplus_g \mathbf{S}_Q) \rVert_{psd}^g$ | the gyrodistance in $\widetilde{\mathcal{G}}(q, d) \times \mathcal{S}_d^{++}$ under metrics $g$ |
| $[\cdot, \cdot]$ | the matrix commutator |
| $\mathrm{expm}(\cdot), \mathrm{logm}(\cdot)$ | Matrix exponentiation and logarithm |
| $\mathscr{L}(\cdot), \mathscr{L}^{-1}(\cdot)$ | Cholesky decomposition and its inverse |
| $\mathbb{D}(\cdot)$ | A diagonal matrix with diagonal elements from a square matrix |
| $\lfloor \cdot \rfloor$ | The strictly lower triangular part of a square matrix |
| $\mathrm{Log}_{\mathbf{P}}^{gr}(\mathbf{Q})$ | Logarithmic map of $\mathbf{Q}$ at $\mathbf{P}$ in $\mathcal{G}(q, d)$ |
| $\mathcal{M}, \mathcal{N}$ | Matrix manifold |
| WFM | the weighted Fréchet mean |
| $\mathrm{hom}_{ai}(\cdot), \mathrm{hom}_{le}(\cdot), \mathrm{hom}_{lc}(\cdot)$ | the maps in $\mathcal{S}_d^{++}$ under AIM, LEM, and LCM satisfying gyro homomorphism |
| $\mathrm{hom}_{gr}(\cdot)$ | the maps in $\widetilde{\mathcal{G}}(q, d)$ satisfying gyro homomorphism |
| $\mathrm{hom}_{psd,g}(\cdot)$ | the maps in $\widetilde{\mathcal{G}}(q, d) \times \mathcal{S}_d^{++}$ under metrics $g$ satisfying gyro homomorphism |
| $\lVert \cdot \rVert_{\mathbf{F}}$ | The norm induced by the standard Frobenius inner product |
| $O(d)$ | The special orthogonal group |
| $\mathrm{Exp}_{\mathbf{P}}^{ai}(\mathbf{A})$ | Exponential map of $\mathbf{A}$ at $\mathbf{P}$ in $\mathcal{S}_d^{++}$ under AIM |
| $\mathrm{Log}_{\mathbf{P}}^{ai}(\mathbf{Q})$ | Logarithmic map of $\mathbf{Q}$ at $\mathbf{P}$ in $\mathcal{S}_d^{++}$ under AIM |
| $\mathrm{Exp}_{\mathbf{P}}^{gr}(\mathbf{W})$ | Exponential map of $\mathbf{W}$ at $\mathbf{P}$ in $\mathcal{G}(q, d)$ |
| $\widetilde{\mathrm{Exp}}_{\mathbf{X}}^{gr}(\mathbf{H})$ | Exponential map of $\mathbf{H}$ at $\mathbf{X}$ in $\widetilde{\mathcal{G}}(q, d)$ |
| $\widetilde{\mathrm{Log}}_{\mathbf{P}}^{gr}(\mathbf{Q})$ | Logarithmic map of $\mathbf{Q}$ at $\mathbf{P}$ in $\widetilde{\mathcal{G}}(q, d)$ |

# C   Abbreviations

For completeness, the abbreviations appearing in this paper are listed below for easy reference.

## List of Abbreviations

| | | |
|---|---|---|
| DNN | Deep Neural Network | 1 |
| EEG | Electroencephalography | 1, 2, 6–9 |
| ERP | Event-Related Potential | 9 |
| GyroAtt | Gyro Attention | 1–7, 9 |
| LCM | Log-Cholesky Metric | 2, 3, 5, 6, 9 |
| LEM | Log-Euclidean Metric | 1–6 |
| ONB | Orthonormal Basis | 2–6 |
| SPD | Symmetric Positive Definite | 1–9 |
| SPSD | Symmetric Positive Semi-Definite | 1–9 |
| WFM | Weighted Fréchet Mean | 2–6 |

# D  Preliminaries

## D.1  Gyrogroups and gyrovector spaces

Gyrogroups and gyrovector spaces generalize groups and vector spaces, offering a powerful framework to analyze non-Euclidean geometries. Below, we formally present their definitions.

**Definition D.1** (**Gyrogroups [69]**). A gyrogroup is a generalization of groups. Let $G$ be a nonempty set with a binary operation $\oplus$ and an identity element $\mathbf{E} \in G$. A pair $(G, \oplus)$ is a gyrogroup if it satisfies the following axioms:

(G1) There exists an identity element $\mathbf{E} \in G$ such that for all $\mathbf{A} \in G$, $\mathbf{E} \oplus \mathbf{A} = \mathbf{A}$.

(G2) For each $\mathbf{A} \in G$, there exists a left inverse $\ominus \mathbf{A} \in G$ satisfying $\ominus \mathbf{A} \oplus \mathbf{A} = \mathbf{E}$.

(G3) For all $\mathbf{A}, \mathbf{B}, \mathbf{C} \in G$, there exists an automorphism $\mathrm{gyr}[\mathbf{A}, \mathbf{B}](\cdot) : G \to G$, satisfying

$$\mathbf{A} \oplus (\mathbf{B} \oplus \mathbf{C}) = (\mathbf{A} \oplus \mathbf{B}) \oplus \mathrm{gyr}[\mathbf{A}, \mathbf{B}](\mathbf{C}). \tag{12}$$

Here, the map $\mathrm{gyr}[\mathbf{A}, \mathbf{B}](\cdot)$ is called the gyroautomorphism, or the gyration of $G$ generated by $\mathbf{A}, \mathbf{B}$.

(G4) For all $\mathbf{A}, \mathbf{B} \in G$, The map $\mathrm{gyr}[\mathbf{A}, \mathbf{B}]$ generated by each $\mathbf{A}, \mathbf{B}$ satisfies the left loop property: $\mathrm{gyr}[\mathbf{A}, \mathbf{B}] = \mathrm{gyr}[\mathbf{A} \oplus \mathbf{B}, \mathbf{B}]$.

**Definition D.2** (**Gyrocommutative Gyrogroups [69]**). A gyrogroup $(G, \oplus)$ is gyrocommutative if it satisfies the gyrocommutative law: $\mathbf{A} \oplus \mathbf{B} = \mathrm{gyr}[\mathbf{A}, \mathbf{B}](\mathbf{B} \oplus \mathbf{A})$ for all $\mathbf{A}, \mathbf{B} \in G$.

The following definition of gyrovector spaces is derived from Nguyen [51, Def. 2.3], which is slightly different from in Ungar [69, Def. 3.2].

**Definition D.3** (**Gyrovector Spaces [51]**). A gyrocommutative gyrogroup $(G, \oplus)$ equipped with a scalar multiplication $\otimes : \mathbb{R} \times G \to G$ is a gyrovector space if the following axioms are satisfied:

(V1) $1 \otimes \mathbf{A} = \mathbf{A}, 0 \otimes \mathbf{A} = t \otimes \mathbf{E} = \mathbf{E}$, and $(-1) \otimes \mathbf{A} = \ominus \mathbf{A}$.

(V2) $(s + t) \otimes \mathbf{A} = s \otimes \mathbf{A} \oplus t \otimes \mathbf{A}$.

(V3) $(st) \otimes \mathbf{A} = s \otimes (t \otimes \mathbf{A})$.

(V4) $\mathrm{gyr}[\mathbf{A}, \mathbf{B}](t \otimes \mathbf{C}) = t \otimes \mathrm{gyr}[\mathbf{A}, \mathbf{B}]\mathbf{C}$.

(V5) $\mathrm{gyr}[s \otimes \mathbf{A}, t \otimes \mathbf{A}] = \mathrm{Id}$, where $\mathrm{Id}$ is the identity map.

## D.2  Weighted Fréchet Mean

### D.2.1  Existence and uniqueness of the Weighted Fréchet Mean

The WFM is a central tool in manifold-based learning for aggregating features that reside on non-Euclidean spaces. Formally, given a set of points $\{x_i\}$ on a Riemannian manifold $\mathcal{M}$ with associated non-negative weights $\{w_i\}$ summing to one, the WFM is defined as the minimizer of the weighted sum of squared distances.

Table 12: Uniqueness and Solution of WFM.

| Geometry | SPD-AIM | SPD-LEM | SPD-LCM | Grassmann | SPSD |
|---|---|---|---|---|---|
| Uniqueness of WFM | Global | Global | Global | Within $r < \frac{\pi}{4\sqrt{2}}$ | The Grassmann component within $r < \frac{\pi}{4\sqrt{2}}$ |
| Solution of WFM | Karcher Flow | Closed Form | Closed Form | Karcher Flow | $\text{WFM}_{\text{spd}}$, $\text{WFM}_{\text{gr}}$ |
| Reference | [56] | [5] | [45] | [2] | [10] |

**Uniqueness.** The uniqueness of WFM is generally guaranteed when all sample points $\{x_i\}$ lie within a geodesically convex hull of $\mathcal{M}$ [2]. In our paper, we focus on several commonly used manifolds, including SPD manifolds with various metrics (AIM, LEM, LCM), the Grassmann manifold, and the SPSD manifold. As summarized in Tab. 12, the WFM is globally unique for SPD manifolds under AIM, LEM, and LCM. For the Grassmann manifold, uniqueness holds when the radius $r$ of the geodesic ball satisfies $r < \frac{\pi}{4\sqrt{2}}$ [2]. The same condition applies to the Grassmann component of the SPSD manifold under its canonical form. Following prior work [12, Sec. 2], we assume that the WFM is well-defined in our settings.

**Computation.** In terms of solution methods, the WFM can be computed via iterative optimization (e.g., Karcher flow) or, in some cases, closed-form expressions. For instance, SPD-LEM and SPD-LCM permit closed-form solutions [5, 45], while SPD-AIM and Grassmann manifolds typically require gradient-based optimization [56, 2, 39]. For SPSD, we follow the product formulation in [10], where the WFM is computed by combining the SPD and Grassmann components.

### D.2.2 Weighted Fréchet Mean on SPD manifolds

---

**Algorithm 1:** Karcher Flow Algorithm on the SPD Manifold under AIM

---

**Input** : A set of SPD matrices $\mathbf{X}_{1\ldots N} \in \mathcal{S}_d^{++}$
A set of weights $w_{1\ldots N} > 0$ with $\sum_i w_i = 1$
Number of iterations $K$
**Output** : The WFM $\mathbf{G}_k \in \mathcal{S}_d^{++}$
Initialize $\mathbf{G}_0 = \mathbf{I}$
**for** $k \leftarrow 1$ **to** $K$ **do**
$\quad \mathbf{G}_k \leftarrow \text{Exp}_{\mathbf{G}_{k-1}}^{ai} \left( \sum_{i=1}^N w_i \, \text{Log}_{\mathbf{G}_{k-1}}^{ai}(\mathbf{X}_i) \right)$
**end**

---

**Affine-Invariant Metric.** We begin by introducing the exponential and logarithmic maps under the affine-invariant metric (AIM), followed by the Karcher flow algorithm.

On the manifold $\mathcal{S}_d^{++}$ endowed with AIM, the exponential map at a point $\mathbf{P} \in \mathcal{S}_d^{++}$ is given by [1]:

$$\text{Exp}_{\mathbf{P}}^{ai}(\mathbf{A}) = \mathbf{P}^{\frac{1}{2}} \text{expm}\left( \mathbf{P}^{-\frac{1}{2}} \mathbf{A} \mathbf{P}^{-\frac{1}{2}} \right) \mathbf{P}^{\frac{1}{2}}, \tag{13}$$

where $\mathbf{A} \in T_{\mathbf{P}} \mathcal{S}_d^{++}$ is a tangent vector at $\mathbf{P}$. The logarithmic map, which is the inverse of the exponential map, is defined as

$$\text{Log}_{\mathbf{P}}^{ai}(\mathbf{Q}) = \mathbf{P}^{\frac{1}{2}} \text{logm}\left( \mathbf{P}^{-\frac{1}{2}} \mathbf{Q} \mathbf{P}^{-\frac{1}{2}} \right) \mathbf{P}^{\frac{1}{2}}, \tag{14}$$

for any $\mathbf{Q} \in \mathcal{S}_d^{++}$.

As shown in Alg. 1, the Karcher flow algorithm computes the weighted Fréchet mean (WFM) on the SPD manifold through an iterative process. In each iteration, the data points are projected onto the tangent space at the current estimate $\mathbf{G}_{k-1}$ using the logarithmic map (Eq. (14)), a weighted average is calculated in this tangent space, and the result is mapped back to the manifold using the exponential map (Eq. (13)). This algorithm is guaranteed to converge on manifolds with non-positive curvatures, such as $\mathcal{S}_d^{++}$ [38]. We initialize $\mathbf{G}_0$ as the identity matrix $\mathbf{I}$ and set the number of iterations $K = 1$.

**Log-Euclidean Metric.** Under the log-Euclidean metric (LEM), the WFM has a closed-form expression provided by Chen et al. [15]:

$$\mathbf{G} = \text{expm}\left( \sum_{i=1}^N w_i \, \text{logm}(\mathbf{X}_i) \right), \tag{15}$$

where $\mathbf{X}_{1\ldots N} \in \mathcal{S}_d^{++}$, $w_{1\ldots N} > 0$, and $\sum_i w_i = 1$.

**Log-Cholesky Metric.** Similarly, for the log-Cholesky metric (LCM), the WFM also admits a closed-form solution as shown by Chen et al. [15]:

$$\mathbf{G} = \mathscr{L}^{-1}\left(\sum_{i=1}^{N} w_i \lfloor \mathscr{L}(\mathbf{X}_i) \rfloor + \prod_{i=1}^{N} \mathbb{D}(\mathscr{L}(\mathbf{X}_i))^{w_i}\right), \tag{16}$$

where $\mathbf{X}_{1\ldots N} \in \mathcal{S}_d^{++}$, $w_{1\ldots N} > 0$, and $\sum_i w_i = 1$.

### D.2.3 Weighted Fréchet Mean on Grassmannian manifolds

We now present the exponential and logarithmic maps, as well as the parallel translation under the ONB perspective, followed by the project perspective.

For the Grassmannian manifold $\widetilde{\mathcal{G}}(q, d)$ in the ONB perspective, the exponential map at $\mathbf{X} \in \widetilde{\mathcal{G}}(q, d)$ is defined as

$$\widetilde{\mathrm{Exp}}_{\mathbf{X}}^{gr}(\mathbf{H}) = \mathbf{X}\mathbf{V}\cos\mathbf{\Sigma} + \mathbf{U}\sin\mathbf{\Sigma}, \tag{17}$$

where $\mathbf{H}$ is a tangent vector at $\mathbf{X}$, and $\mathbf{U}\mathbf{\Sigma}\mathbf{V}^\top$ is the thin singular value decomposition (SVD) of $\mathbf{H}$:

$$\mathbf{U}\mathbf{\Sigma}\mathbf{V}^\top = \mathrm{thinSVD}(\mathbf{H}). \tag{18}$$

The logarithmic map, which is the inverse of the exponential map, is given by

$$\widetilde{\mathrm{Log}}_{\mathbf{X}}^{gr}(\mathbf{Y}) = \mathbf{U}\tan^{-1}\mathbf{\Sigma}\mathbf{V}^\top, \tag{19}$$

where $\mathbf{X}, \mathbf{Y} \in \widetilde{\mathcal{G}}(q, d)$, and

$$\mathbf{U}\mathbf{\Sigma}\mathbf{V}^\top = \mathrm{thinSVD}\left((\mathbf{I} - \mathbf{X}\mathbf{X}^\top)\mathbf{Y}(\mathbf{X}^\top\mathbf{Y})^{-1}\right). \tag{20}$$

As stated in Edelman et al. [23, Theorem 2.4], let $\mathbf{H}$ and $\Delta$ be tangent vectors at point $\mathbf{Y}$ on the Grassmann manifold. The parallel transport of $\Delta$ along the geodesic in the direction $\dot{\mathbf{Y}}(0) = \mathbf{H}$ is given by

$$\tau\Delta(t) = \left((\mathbf{Y}\mathbf{V} \quad \mathbf{U})\begin{pmatrix} -\sin(\mathbf{\Sigma}t) \\ \cos(\mathbf{\Sigma}t) \end{pmatrix}\mathbf{U}^\top + (\mathbf{I} - \mathbf{U}\mathbf{U}^\top)\right)\Delta. \tag{21}$$

Shifting to the projector perspective for the Grassmannian manifold $\mathcal{G}(q, d)$, let $\mathbf{P} \in \mathcal{G}(q, d)$ and $\Delta \in T_{\mathbf{P}}\mathcal{G}(q, d)$. The exponential map is defined as [8]

$$\mathrm{Exp}_{\mathbf{P}}^{gr}(\Delta) = \mathrm{expm}([\Delta, \mathbf{P}])\mathbf{P}\,\mathrm{expm}(-[\Delta, \mathbf{P}]). \tag{22}$$

As shown by Sakai [58], two points are in each other's cut locus if there exists more than one shortest geodesic connecting them. When the exponential map $\mathrm{Exp}_{\mathbf{P}}^{gr}$ is restricted to the injectivity domain $\mathrm{ID}_{\mathbf{P}}$, for any $\mathbf{F} \in \mathcal{G}(q, d) \setminus \mathrm{Cut}_{\mathbf{P}}$, there exists a unique tangent vector $\Delta \in \mathrm{ID}_{\mathbf{P}} \subset T_{\mathbf{P}}\mathcal{G}(q, d)$ such that $\mathrm{Exp}_{\mathbf{P}}^{gr}(\Delta) = \mathbf{F}$. For such a point $\mathbf{F}$, the logarithmic map is given by

$$\mathrm{Log}_{\mathbf{P}}^{gr}(\mathbf{Q}) = [\Omega, \mathbf{P}], \tag{23}$$

where $\mathbf{P}, \mathbf{Q} \in \mathrm{Gr}_{n,p}$, and $\Omega$ is calculated as

$$\Omega = \frac{1}{2}\log\left((\mathbf{I}_n - 2\mathbf{Q})(\mathbf{I}_n - 2\mathbf{P})\right). \tag{24}$$

As shown in Alg. 1, the Karcher flow algorithm computes the WFM on the Grassmannian manifold through an iterative process. We initialize $\mathbf{G}_0$ as the identity matrix $\mathbf{X}_i$ and set the number of iterations $K = 1$.

### D.2.4 Weighted Fréchet Mean on SPSD manifolds

As demonstrated by Bonnabel and Sepulchre [9], the WFM for a batch of points $\mathbf{X}_{1,\ldots N} \in \mathcal{S}_{d,q}^+$ can be expressed as $(\mathrm{WFM}_{\mathrm{gr}}(\mathbf{U}_X^i), \mathrm{WFM}_{\mathrm{spd}}^g(\mathbf{S}_X^i))$. Here, $\mathrm{WFM}_{\mathrm{gr}}$ denotes the WFM on the Grassmannian manifold, while $\mathrm{WFM}_{\mathrm{spd}}^g(\cdot)$ represents the WFM on the SPD manifold under metric $g$. The matrices $\mathbf{U}_X^i$ and $\mathbf{S}_X^i$ correspond to the canonical representation of $\mathbf{X}_i$.

**Algorithm 2:** Karcher Flow Algorithm on the Grassmannian Manifold under ONB Perspective

---

**Input:** A set of Grassmannian points $\mathbf{X}_{1\ldots N} \in \widetilde{\mathcal{G}}(q, d)$
      A set of weights $w_{1\ldots N} > 0$ with $\sum_i w_i = 1$
      Number of iterations $K$
**Output:** The WFM $\mathbf{G} \in \widetilde{\mathcal{G}}(q, d)$
Initialize $\mathbf{G}_0 = \mathbf{X}_1$
**for** $k \leftarrow 1$ **to** $K$ **do**
    $\mathbf{G}_k \leftarrow \widetilde{\mathrm{Exp}}^{gr}_{\mathbf{G}_{k-1}} \left( \sum_{i=1}^{N} w_i \widetilde{\mathrm{Log}}^{gr}_{\mathbf{G}_{k-1}}(\mathbf{X}_i) \right)$
**end**

---

**Algorithm 3:** Computation of Canonical Representation in SPSD manifold

---

**Input:** A batch of SPSD matrices $\mathbf{X}_{1\ldots N} \in \mathrm{S}^+_{n,q}$
      A constant $\gamma \in [0, 1]$
**Output:** A batch of Canonical Representation $(\mathbf{U}^i_X, \mathbf{S}^i_X)_{i=1,\ldots,N}$ of SPSD manifold
$\mathbf{U}^m \leftarrow \widetilde{\mathbf{I}}_{n,q}$;
$(\mathbf{U}_i, \mathbf{\Sigma}_i, \mathbf{V}_i)_{i=1,\ldots,N} \leftarrow \mathrm{SVD}((\mathbf{X}_i)_{i=1,\ldots,N})$
$(\mathbf{U}_i)_{i=1,\ldots,N} \leftarrow (\mathbf{U}_i[:, :q])_{i=1,\ldots,N}$;
**if** *training* **then**
    $\mathbf{U} \leftarrow \mathrm{GrMean}((\mathbf{U}_i)_{i=1,\ldots,N})$
    $\mathbf{U}^m \leftarrow \mathrm{GrGeodesic}(\mathbf{U}^m, \mathbf{U}, \gamma)$
**end**
**for** $i \leftarrow 1$ **to** $N$ **do**
    $(\mathbf{U}_i)^\top \mathbf{U}^m = \mathbf{Y}_i (\cos \mathbf{\Sigma}_i) \mathbf{V}_i^\top$
    $(\mathbf{U}^i_X, \mathbf{S}^i_X) \leftarrow (\mathbf{U}_i \mathbf{Y}_i, \mathbf{V}_i \mathbf{Y}_i^\top \mathbf{U}_i^\top \mathbf{\Sigma}_i \mathbf{U}_i \mathbf{Y}_i \mathbf{V}_i^\top)$
**end**

---

### D.3 Canonical representation in SPSD manifolds

Nguyen et al. [53] introduced a canonical representation of $\mathbf{P}$ in the structure space $\widetilde{\mathcal{G}}(q, d) \times \mathcal{S}^{++}_q$. As shown in Alg. 3, we follow this approach to derive the canonical representation of each point in $\mathcal{S}^+_{d,q}$. Canonical Representation of SPSD matrices is obtained in three steps. This first is to impose a decomposition on $\mathbf{X}_i$, *i.e.*, $\mathbf{X}_i \simeq \mathbf{U}_i \mathbf{\Sigma}_i \mathbf{U}_i^\top$, where $\mathbf{U}_i \in \widetilde{\mathcal{G}}(q, d)$ and $\mathbf{\Sigma}_i \in \mathcal{S}^{++}_q$. Then we use the mean of $\mathbf{U}_i)_{i=1,\ldots,N}$ as the common subspace, and rotated $(\mathbf{U}^i, \mathbf{\Sigma}^i)$ to the identified common subspace, denoted as $(\mathbf{U}^i_X, \mathbf{S}^i_X)$. Here, $\mathrm{GrMean}((\mathbf{U}_i)_{i=1,\ldots,N})$ computes the Fréchet mean of its arguments, as described in Alg. 2, with weights set to $w_{1,\ldots,N} = \frac{1}{N}$. $\mathrm{GrGeodesic}(\mathbf{U}^m, \mathbf{U}, \gamma)$ computes a point on a geodesic (Eq. (21)) from $\mathbf{U}^m$ to $\mathbf{U}$ at step $\gamma$ ($\gamma = 0.1$ in our experiments).

## E Implementation details and additional experiments

### E.1 Datasets and EEG preprocessing

**MAMEM-SSVEP-II.** This dataset includes EEG recordings from 11 subjects performing SSVEP tasks. Participants focused on one of five visual stimuli flickering at different frequencies for five seconds. Each subject completed five sessions, with five trials per stimulation frequency in each session. EEG signals were captured with 256 channels at a sampling rate of 250 Hz.

**BCI-ERN.** This dataset involves 26 subjects in a P300-based spelling task to measure ERN. EEG data were recorded from 56 electrodes following the extended 10-20 system at a sampling rate of 600 Hz. Each subject underwent five sessions: the first four with 60 trials each and the fifth with 100 trials. We used data from 16 subjects available in the initial competition release.

**BNCI2014001.** This dataset comprises EEG recordings from 9 subjects performing four motor imagery tasks: imagining movements of the left hand, right hand, both feet, and tongue. Each subject participated in two sessions on different days, each containing six runs. Each run included 48 trials—12 per class—totaling 288 trials per session.

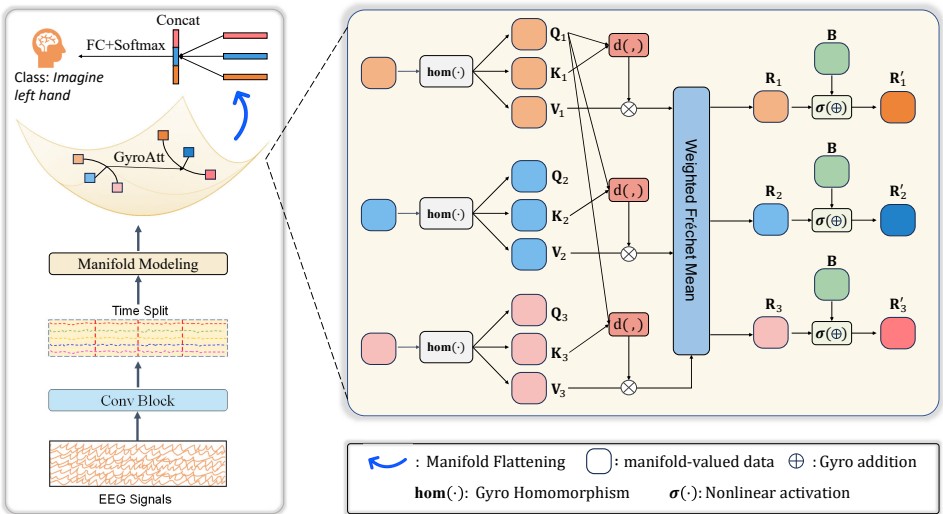

Figure 4: The GyroAtt network architecture comprises three components: a feature extraction module that converts EEG signals into manifold-valued data, a Gyro Attention module that explicitly captures long-range dependencies among features, and a classification module that flattens manifold data for a fully connected layer and softmax function.

**BNCI2015001.** EEG signals were recorded from electrodes centered around positions C3, Cz, and C4, according to the International 10-20 System. Data were collected using a g.GAMMAsys active electrode system with a g.USBamp amplifier, sampled at 512 Hz with a bandpass filter between 0.5 and 100 Hz and a notch filter at 50 Hz.

**EEG Preprocessing.** For the MAMEM-SSVEP-II dataset, we adhered to the preprocessing protocol of Pan et al. [55]. The steps included: (1) band-pass filtering between 1–50 Hz; (2) selecting eight channels (PO7, PO3, PO, PO4, PO8, O1, Oz, and O2) located in the occipital area corresponding to the visual cortex; and (3) segmenting each trial into four 1-second segments from 1s to 5s after cue onset. This resulted in 500 trials of 1-second, 8-channel SSVEP signals per subject, with each input EEG segment comprising 125 time points.

For the BCI-ERN dataset, we followed the preprocessing procedure outlined by Pan et al. [55]. The steps involved: (1) downsampling the signals from 600 Hz to 128 Hz; (2) applying a band-pass filter between 1–40 Hz. After preprocessing, each trial consisted of 56 channels with 160 time points.

For the BNCI2014001 and BNCI2015001 datasets, we followed the preprocessing steps described by Kobler et al. [41]. Using the Python packages moabb and mne, we resampled the EEG signals to 250/256 Hz, applied temporal filters to extract oscillatory activity in the 4–36 Hz range, and extracted short segments ($\leq 3$ seconds) associated with class labels.

Table 13: GyroAtt-SPSD architectures across four datasets. The $q$ is the rank of the SPSD matrices.

| Block | MAMEM-SSVEP-II | BCI-ERN | BNCI2014001 | BNCI2015001 | Operation |
|---|---|---|---|---|---|
| Input data | $1 \times 8 \times 125$ | $1 \times 56 \times 160$ | $1 \times 22 \times 750$ | $1 \times 13 \times 768$ | |
| TempConv | $125 \times 1 \times 125$ | $22 \times 1 \times 160$ | $4 \times 22 \times 750$ | $5 \times 13 \times 768$ | Convolution |
| SpatConv | $21 \times 1 \times 126$ | $57 \times 1 \times 161$ | $43 \times 1 \times 750$ | $44 \times 1 \times 768$ | Convolution |
| Split & CovPool | $2 \times 21 \times 21$ | $3 \times 19 \times 19$ | $6 \times 43 \times 43$ | $3 \times 44 \times 44$ | Split + Covariance |
| SPDDSMBN | w/o | w/o | $6 \times 43 \times 43$ | $3 \times 44 \times 44$ | Domain Alignment Kobler et al. [41] |
| SPSDCom | $2 \times (21 \times 9, 9 \times 9)$ | $3 \times (19 \times q, q \times q)$ | $6 \times (43 \times 18, 18 \times 18)$ | $3 \times (44 \times 18, 18 \times 18)$ | Alg. 3 |
| GyroAtt-SPSD | $2 \times (21 \times 9, 9 \times 9)$ | $3 \times (19 \times q, q \times q)$ | $6 \times (43 \times 18, 18 \times 18)$ | $3 \times (44 \times 18, 18 \times 18)$ | Alg. 4 |
| R2E | $2 \times (21 \times 21, 9 \times 9)$ | $3 \times (19 \times 19, q \times q)$ | $6 \times (43 \times 43, 18 \times 18)$ | $3 \times (44 \times 44, 18 \times 18)$ | $(\Phi(\cdot), \psi(\cdot))$ |
| Flat | $(882, 162)$ | $(1083, q^2)$ | $(11094, 1944)$ | $(5547, 972)$ | Vectorization |
| Classifier | 5 | 2 | 4 | 2 | FC + Softmax |

Table 14: GyroAtt-SPD architectures across four datasets.

| Block | MAMEM-SSVEP-II | BCI-ERN | BNCI2014001 | BNCI2015001 | Operation |
|---|---|---|---|---|---|
| Input data | $1 \times 8 \times 125$ | $1 \times 56 \times 160$ | $1 \times 22 \times 750$ | $1 \times 13 \times 768$ | |
| TempConv | $125 \times 1 \times 125$ | $22 \times 1 \times 160$ | $4 \times 22 \times 750$ | $5 \times 13 \times 768$ | Convolution |
| SpatConv | $21 \times 1 \times 126$ | $57 \times 1 \times 161$ | $43 \times 1 \times 750$ | $44 \times 1 \times 768$ | Convolution |
| Split & CovPool | $2 \times 21 \times 21$ | $3 \times 19 \times 19$ | $6 \times 43 \times 43$ | $3 \times 44 \times 44$ | Split + Covariance |
| SPDDSMBN | w/o | w/o | $6 \times 43 \times 43$ | $3 \times 44 \times 44$ | Domain Alignment |
| GyroAtt-SPD | $2 \times 21 \times 21$ | $3 \times 19 \times 19$ | $6 \times 43 \times 43$ | $3 \times 44 \times 44$ | Alg. 4 |
| R2E | $2 \times 21 \times 21$ | $3 \times 19 \times 19$ | $6 \times 43 \times 43$ | $3 \times 44 \times 44$ | $\psi(\cdot)$ |
| Flat | 882 | 1083 | 11094 | 5547 | Vectorization |
| Classifier | 5 | 2 | 4 | 2 | FC + Softmax |

Table 15: GyroAtt-Gr Architectures across four datasets. The $q$ is the dimension of the linear subspaces.

| Block | MAMEM-SSVEP-II | BCI-ERN | BNCI2014001 | BNCI2015001 | Operation |
|---|---|---|---|---|---|
| Input data | $1 \times 8 \times 125$ | $1 \times 56 \times 160$ | $1 \times 22 \times 750$ | $1 \times 13 \times 768$ | |
| TempConv | $125 \times 1 \times 125$ | $22 \times 1 \times 160$ | $4 \times 22 \times 750$ | $5 \times 13 \times 768$ | Convolution |
| SpatConv | $21 \times 1 \times 126$ | $57 \times 1 \times 161$ | $43 \times 1 \times 750$ | $43 \times 1 \times 768$ | Convolution |
| Split & CovPool | $2 \times 21 \times 21$ | $3 \times 19 \times 19$ | $6 \times 43 \times 43$ | $3 \times 44 \times 44$ | Split + Covariance |
| SPDDSMBN | w/o | w/o | $6 \times 43 \times 43$ | $3 \times 44 \times 44$ | Domain Alignment |
| GrCom | $2 \times 21 \times 9$ | $3 \times 19 \times q$ | $6 \times 43 \times 18$ | $3 \times 44 \times 18$ | Alg. 3 |
| GyroAtt-Gr | $2 \times 21 \times 9$ | $3 \times 19 \times q$ | $6 \times 43 \times 18$ | $3 \times 44 \times 18$ | Alg. 4 |
| R2E | $2 \times 21 \times 21$ | $3 \times 19 \times 19$ | $6 \times 43 \times 43$ | $3 \times 44 \times 44$ | $\Phi(\cdot)$ |
| Flat | 882 | 1083 | 11094 | 5547 | Vectorization |
| Classifier | 5 | 2 | 4 | 2 | FC + Softmax |

## E.2 Additional implementation details

### E.2.1 Network architectures.

Fig. 4 and Tab. 13 provide a summary of the specific network architectures of GyroAtt-SPSD across the four datasets, and we summarize the forward pass of GyroAtt in Alg. 4 . The network structures for GyroAtt-Gr (Tab. 15) and GyroAtt-SPD (Tab. 14) are identical to that of GyroAtt-SPSD. We just introduced GyroAtt-SPSD as an example.

For the MAMEM-SSVEP-II and BCI-ERN datasets, the initial convolutional block consists of a convolutional layer, followed by batch normalization and an ELU activation function. The subsequent convolutional block performs depthwise spatial convolution. A pointwise convolution, batch normalization, and another ELU activation follow this. In the MAMEM-SSVEP-II dataset, features are split into two non-overlapping segments, followed by covariance pooling. For the BCI-ERN dataset, the second convolutional block is repeated in two additional blocks. The outputs from these blocks are concatenated along the channel dimension. The data is then split along the channel dimension, and covariance pooling is applied, resulting in three covariance matrices.

For BNCI2014001 and BNCI2015001 datasets, the initial convolutional layer employs 4 or 5 filters with a kernel size of $(1, 25)$, performing temporal convolution while maintaining the same size through padding. The second convolutional layer applies spatial convolution with a kernel size of $(22, 1)$ to integrate information from different channels. The output sequences undergo temporal pyramid partitioning, dividing each sequence into $i$ equal segments at the $i$-th level (with levels set to 3 and 2, respectively). To address distribution shifts across subjects and runs, we incorporate subject- and run-specific batch normalization layers [41].

The attention module designed in the gyrovector spaces is constituted by five operation layers, which are the Gyro homomorphism layer ($f_{\text{hom}}$) used to generate $\mathbf{Q}_i$, $\mathbf{K}_i$, and $\mathbf{V}_i$ for each input data, the similarity measurement layer ($f_{\text{sim}}$) for computing the correlation between $\mathbf{Q}_i$ and $\mathbf{K}_j$, the $\text{Softmax}$ layer ($f_{\text{smx}}$) used to normalize the obtained attention matrix along the row direction, the weighted Fréchet Mean layer ($f_{\text{wFM}}$) for the implementation of weighted aggregation, and the power-based nonlinear activation layer ($f_{\text{pac}}$) used to improve the representational capacity of GyroAtt module by introducing nonlinearity to the underlying metric space.

For classification, our GyroAtt-SPD model employs matrix power normalization following Wang et al. [71] and Chen et al. [16]. Specifically, we apply the transformation $\psi_\theta(\mathbf{P}) = \frac{1}{\theta}\mathbf{P}^\theta$ to the $i$-th

---
**Algorithm 4:** Gyro Attention over gyrovector spaces

---
**Input:** A set of manifold-valued features $\{\mathbf{X}_{1\dots N}\}$
**Output:** A set of manifold-valued features $\{\mathbf{R}'_{1\dots N}\}$

---
**for** $i \leftarrow 1$ **to** $N$ **do**
  Queries: $\mathbf{Q}_i = \text{hom}(\mathbf{X}_i)$,
  Keys: $\mathbf{K}_i = \text{hom}(\mathbf{X}_i)$,
  Values: $\mathbf{V}_i = \text{hom}(\mathbf{X}_i)$
**end**
**for** $i \leftarrow 1$ **to** $N$ **do**
  **for** $j \leftarrow 1$ **to** $N$ **do**
    Similarity: $\mathcal{S}_{ij} = (1 + \log(1 + \text{d}(\mathbf{Q}_i, \mathbf{K}_j)))^{-1}$
  **end**
  Attention calculation: $\mathcal{A}_{ij} = \text{Softmax}(\mathcal{S}_{ij})$
  Aggregation: $\mathbf{R}_i = \text{WFM}(\{\mathcal{A}_{ij}\}_{j=1}^N, \{\mathbf{V}_j\}_{j=1}^N)$
  Bias and nonlinearity: $\mathbf{R}'_i = \sigma(\mathbf{R}_i \oplus \mathbf{B})$
**end**

---

output matrix $\mathbf{P} \in \mathcal{S}_d^{++}$, where $\theta > 0$. The coefficient $\frac{1}{\theta}$ stabilizes the gradient flow during training and facilitates convergence. In GyroAtt-Gr, we transform elements $\mathbf{Y}_i \in \mathcal{G}(q,d)$ by applying a projection operator $\Phi(\mathbf{Y}_i) = \mathbf{Y}_i\mathbf{Y}_i^\top$ to map them into the corresponding flat space. In contrast, for GyroAtt-SPSD, we project $(\mathbf{U}_X^i, \mathbf{S}_X^i) \in \widetilde{\mathcal{G}}(q,d) \times \mathcal{S}_q^{++}$ onto their respective manifolds. In all three GyroAtt, the transformed matrices are vectorized, concatenated, and fed into a fully connected layer followed by a $\text{Softmax}$ function. All experiments were conducted on an i9-14900 CPU with 64GB RAM and two NVIDIA RTX4080 Super GPUs.

### E.2.2 Optimization

We address the optimization of parameters that are SPD matrices by modeling them within the space of symmetric matrices and applying the exponential map to the identity matrix.

For any parameter $\mathbf{P} \in \widetilde{\mathcal{G}}(d,q)$, we parameterize it using a matrix $\mathbf{B} \in \mathbb{R}^{q,d-q}$ such that

$$\begin{bmatrix} 0 & \mathbf{B} \\ -\mathbf{B}^\top & 0 \end{bmatrix} = [\text{Log}_{\mathbf{I}_{n,p}}^{gr}(\mathbf{P}\mathbf{P}^\top), \mathbf{I}_{n,p}]. \tag{25}$$

With this parameterization, the parameter $\mathbf{P}$ can be computed as

$$\mathbf{P} = \exp\left(\begin{bmatrix} 0 & \mathbf{B} \\ -\mathbf{B}^\top & 0 \end{bmatrix}\right) \widetilde{\mathbf{I}}_{n,p}.$$

To optimize parameters $\mathbf{O} \in SO(n)$, we start by generating parameter $\mathbf{A} \in \mathbb{R}^{n \times n}$, then compute its skew-symmetric matrix $\mathbf{S} = \mathbf{A} - \mathbf{A}^\top$. With this parameterization, the parameter $\mathbf{P}$ can be computed as

$$\mathbf{O} = (\mathbf{I} - \mathbf{S})(\mathbf{I} + \mathbf{S})^{-1}, \tag{26}$$

This approach enables us to optimize all parameters within Euclidean spaces, eliminating the need to employ optimization techniques specific to Riemannian manifolds.

### E.3 Statistical Significance Analysis

To assess the robustness of the reported improvements, we conducted paired $t$-tests comparing GyroAtt with the baseline MAtt across all evaluation folds. Tab. 16 reports the average performance gains and corresponding significance levels, where * denotes $p$-value $< 0.05$ and *n.s.* indicates "not significant".

Although some confidence intervals overlap, the observed improvements are consistent across benchmarks. In particular, the $+14.8\%$ gain under the inter-subject condition—known to be the most challenging EEG setting—demonstrates the practical effectiveness and robustness of GyroAtt.

Table 16: Paired *t*-test results comparing GyroAtt with MAtt. Asterisks (*) denote statistical significance ($p < 0.05$).

| | BNCI2014001 | | BNCI2015001 | | BCI-CHA | MAMEM |
|---|---|---|---|---|---|---|
| | Inter-session | Inter-subject | Inter-session | Inter-subject | | |
| Gains (%) | 7.8 | 8.9 | 5.4 | 14.8 | 3.4 | 3.2 |
| Significance | * | n.s. | n.s. | * | * | * |

## E.4 Graph node classification and radar recognition tasks

While our primary evaluation focuses on EEG decoding, we further assess the generalization capability of GyroAtt on two additional domains: graph node classification and drone recognition.

Table 17: Accuracy (%) on graph node classification benchmarks.

| Method | Pubmed | Cora |
|---|---|---|
| H-GAT | $77.5 \pm 1.6$ | $78.1 \pm 1.1$ |
| SPD-GAT | $77.8 \pm 0.6$ | $79.4 \pm 0.6$ |
| SPD-GAT-MAtt | $77.5 \pm 0.6$ | $80.1 \pm 1.1$ |
| SPD-GAT-GyroAtt | $\mathbf{78.5 \pm 0.4}$ | $\mathbf{81.6 \pm 1.0}$ |

**Graph node classification.** We conduct experiments on two standard graph node classification datasets: Pubmed [49] and Cora [60]. We adopt SPD-GAT [75] as the backbone and substitute its attention module with GyroAtt-SPD (denoted SPD-GAT-GyroAtt) and MAtt (SPD-GAT-MAtt). We also include H-GAT [75] for comparison. As shown in Tab. 17, GyroAtt-SPD consistently outperforms all baselines on both datasets.

Table 18: Accuracy (%) on Radar dataset for drone recognition.

| Method | Radar |
|---|---|
| MAtt | $96.8 \pm 1.1$ |
| GDLNet | $94.7 \pm 0.9$ |
| GyroAtt-SPD | $\mathbf{98.5 \pm 0.8}$ |
| GyroAtt-Gr | $96.2 \pm 1.2$ |
| GyroAtt-SPSD | $96.9 \pm 0.9$ |

**Drone recognition.** We evaluate GyroAtt on the Radar dataset [11], following the GDLNet [72] architecture. The dataset comprises 3,000 synthetic radar signals, evenly distributed across three classes. Tab. 18 shows that GyroAtt-SPD attains the highest accuracy, further validating the adaptability of our method across different modalities.

## E.5 Additional ablations studies

### E.5.1 Ablations on the Riemannian metrics and matrix power-based nonlinear activation

Tab. 19 illustrates the impact of the different metrics and power parameter $p$ (as defined in Tab. 3) on the performance of GyroAtt based on two Riemannian matrix manifolds. The candidate values of metrics are AIM, LEM, and LCM, with $p$ values set to $\{0.25, 0.50, 0.75\}$. As shown in this table, for SPD-based architectures, GyroAtt under the SPD-AIM geometry with $p = 0.5$ achieves the highest accuracy on both the BNCI2014001 and BNCI2015001 datasets, while the SPD-LCM geometry with $p = 0.75$ records the second-highest inter-session accuracy (86.0%) on the BNCI2015001 dataset. For SPSD-based settings, GyroAtt under the SPSD-LCM geometry with $p = 0.75$ reaches the highest accuracy (68.7%) on the MAMEM-SSVEP-II dataset. Furthermore, it is evident that GyroAtt is generally robust to variations in $p$ across all experimental scenarios. These findings emphasize the importance of selecting the metric space of the underlying feature manifold and demonstrate that the proposed matrix power activation enhances model performance by introducing nonlinearity into the metric space.

Table 19: Ablations of GyroAtt on Riemannian metrics and matrix power activation $p$. The best result under each geometry is highlighted in **bold**.

| Geometry | $p$ | BNCI2014001 | | BNCI2015001 | | MAMEM-SSVEP-II |
| --- | --- | --- | --- | --- | --- | --- |
| | | Inter-session | Inter-subject | Inter-session | Inter-subject | |
| SPD-AIM | w/o | $74.8 \pm 6.7$ | $51.2 \pm 15.7$ | $85.5 \pm 5.0$ | $75.4 \pm 12.9$ | $64.3 \pm 2.4$ |
| | 0.25 | $75.2 \pm 6.9$ | $51.4 \pm 14.3$ | $85.8 \pm 6.8$ | $77.1 \pm 12.8$ | $61.9 \pm 2.5$ |
| | 0.50 | $\mathbf{75.4 \pm 7.1}$ | $\mathbf{53.1 \pm 14.8}$ | $\mathbf{86.2 \pm 4.5}$ | $\mathbf{77.9 \pm 13.0}$ | $66.1 \pm 2.6$ |
| | 0.75 | $75.0 \pm 8.1$ | $51.0 \pm 13.8$ | $85.9 \pm 6.6$ | $77.4 \pm 12.6$ | $\mathbf{66.3 \pm 2.2}$ |
| SPD-LEM | w/o | $74.9 \pm 7.3$ | $51.7 \pm 15.8$ | $85.2 \pm 5.2$ | $75.3 \pm 12.3$ | $65.6 \pm 2.3$ |
| | 0.25 | $74.7 \pm 6.7$ | $\mathbf{52.3 \pm 14.1}$ | $85.6 \pm 6.7$ | $75.6 \pm 13.0$ | $63.7 \pm 2.5$ |
| | 0.50 | $\mathbf{75.3 \pm 6.5}$ | $51.4 \pm 14.1$ | $\mathbf{85.7 \pm 5.5}$ | $\mathbf{76.6 \pm 13.7}$ | $65.3 \pm 2.7$ |
| | 0.75 | $75.1 \pm 7.3$ | $52.3 \pm 15.0$ | $85.4 \pm 7.0$ | $75.5 \pm 12.8$ | $\mathbf{66.2 \pm 2.5}$ |
| SPD-LCM | w/o | $73.2 \pm 6.7$ | $51.9 \pm 14.8$ | $85.3 \pm 7.2$ | $76.2 \pm 13.3$ | $64.0 \pm 2.8$ |
| | 0.25 | $73.4 \pm 7.5$ | $52.4 \pm 13.4$ | $85.6 \pm 7.5$ | $75.4 \pm 14.0$ | $64.3 \pm 2.5$ |
| | 0.50 | $74.0 \pm 8.2$ | $\mathbf{52.7 \pm 13.6}$ | $85.9 \pm 6.7$ | $\mathbf{77.4 \pm 13.2}$ | $64.1 \pm 3.2$ |
| | 0.75 | $\mathbf{74.2 \pm 7.8}$ | $51.7 \pm 14.6$ | $\mathbf{86.0 \pm 6.8}$ | $76.3 \pm 13.2$ | $\mathbf{65.1 \pm 2.5}$ |
| SPSD-AIM | w/o | $72.2 \pm 7.2$ | $49.2 \pm 13.7$ | $84.1 \pm 7.2$ | $73.6 \pm 14.3$ | $65.8 \pm 2.6$ |
| | 0.25 | $72.4 \pm 6.8$ | $50.7 \pm 14.8$ | $84.0 \pm 6.8$ | $\mathbf{75.5 \pm 13.8}$ | $66.4 \pm 3.0$ |
| | 0.50 | $\mathbf{72.9 \pm 7.1}$ | $\mathbf{52.4 \pm 15.6}$ | $\mathbf{84.7 \pm 6.6}$ | $74.2 \pm 14.2$ | $\mathbf{66.5 \pm 2.9}$ |
| | 0.75 | $72.5 \pm 6.7$ | $51.0 \pm 15.3$ | $84.0 \pm 4.9$ | $74.5 \pm 13.6$ | $65.7 \pm 2.7$ |
| SPSD-LEM | w/o | $72.1 \pm 6.7$ | $49.8 \pm 12.9$ | $83.9 \pm 5.1$ | $74.2 \pm 14.4$ | $66.2 \pm 1.9$ |
| | 0.25 | $\mathbf{72.8 \pm 6.9}$ | $49.9 \pm 14.0$ | $\mathbf{85.3 \pm 5.3}$ | $\mathbf{76.0 \pm 14.1}$ | $\mathbf{66.5 \pm 2.3}$ |
| | 0.50 | $72.5 \pm 6.6$ | $50.5 \pm 14.2$ | $84.5 \pm 5.8$ | $75.4 \pm 14.5$ | $66.4 \pm 2.5$ |
| | 0.75 | $72.7 \pm 7.6$ | $\mathbf{50.5 \pm 13.2}$ | $85.2 \pm 4.8$ | $75.0 \pm 14.2$ | $66.2 \pm 1.7$ |
| SPSD-LCM | w/o | $72.3 \pm 7.3$ | $49.5 \pm 12.0$ | $84.8 \pm 6.1$ | $75.4 \pm 13.2$ | $66.5 \pm 2.4$ |
| | 0.25 | $72.2 \pm 7.5$ | $50.6 \pm 13.9$ | $\mathbf{85.1 \pm 4.8}$ | $\mathbf{74.9 \pm 12.6}$ | $67.7 \pm 2.3$ |
| | 0.50 | $\mathbf{72.9 \pm 6.7}$ | $48.4 \pm 13.3$ | $84.9 \pm 6.1$ | $74.5 \pm 13.6$ | $66.2 \pm 3.6$ |
| | 0.75 | $72.8 \pm 6.3$ | $\mathbf{51.7 \pm 13.1}$ | $85.1 \pm 5.8$ | $73.9 \pm 15.4$ | $\mathbf{68.7 \pm 1.5}$ |

Table 20: Ablations of GyroAtt on matrix power normalization $\theta$ used in classification and Riemannian metrics. The best result under each geometry is highlighted in **bold**.

| Geometry | $\theta$ | BNCI2014001 | | BNCI2015001 | | MAMEM-SSVEP-II |
| --- | --- | --- | --- | --- | --- | --- |
| | | Inter-session | Inter-subject | Inter-session | Inter-subject | |
| SPD-AIM | 0.25 | $74.9 \pm 6.9$ | $51.2 \pm 13.6$ | $86.1 \pm 7,3$ | $76.2 \pm 12.8$ | $61.9 \pm 2.5$ |
| | 0.50 | $\mathbf{75.4 \pm 7.1}$ | $\mathbf{53.1 \pm 14.8}$ | $\mathbf{86.2 \pm 4.5}$ | $\mathbf{77.9 \pm 13.0}$ | $66.2 \pm 2.8$ |
| | 0.75 | $75.0 \pm 8.1$ | $51.7 \pm 14.5$ | $86.0 \pm 6.5$ | $77.1 \pm 14.3$ | $\mathbf{66.3 \pm 2.2}$ |
| SPD-LEM | 0.25 | $75.2 \pm 6.7$ | $\mathbf{52.7 \pm 12.9}$ | $85.1 \pm 5.7$ | $\mathbf{76.9 \pm 14.5}$ | $60.7 \pm 2.4$ |
| | 0.50 | $\mathbf{75.3 \pm 6.5}$ | $51.4 \pm 14.1$ | $85.7 \pm 5.5$ | $76.6 \pm 13.7$ | $66.1 \pm 2.8$ |
| | 0.75 | $75.1 \pm 7.3$ | $52.3 \pm 13.3$ | $\mathbf{85.8 \pm 6.3}$ | $76.4 \pm 13.1$ | $\mathbf{66.2 \pm 2.5}$ |
| SPD-LCM | 0.25 | $\mathbf{74.2 \pm 7.5}$ | $52.1 \pm 14.5$ | $85.6 \pm 5.9$ | $77.3 \pm 13.4$ | $64.5 \pm 2.9$ |
| | 0.50 | $74.0 \pm 8.2$ | $\mathbf{52.7 \pm 13.6}$ | $85.9 \pm 6.7$ | $\mathbf{77.4 \pm 13.2}$ | $64.3 \pm 2.8$ |
| | 0.75 | $74.1 \pm 7.8$ | $52.0 \pm 14.7$ | $\mathbf{86.0 \pm 5.3}$ | $75.8 \pm 13.8$ | $\mathbf{65.1 \pm 2.5}$ |
| SPSD-AIM | 0.25 | $72.7 \pm 7.0$ | $51.2 \pm 15.8$ | $84.0 \pm 6.8$ | $\mathbf{75.5 \pm 13.8}$ | $66.3 \pm 2.9$ |
| | 0.50 | $\mathbf{72.9 \pm 6.2}$ | $\mathbf{52.4 \pm 15.6}$ | $\mathbf{84.5 \pm 6.6}$ | $74.2 \pm 15.2$ | $\mathbf{66.3 \pm 2.4}$ |
| | 0.75 | $72.7 \pm 6.7$ | $50.0 \pm 15.2$ | $84.4 \pm 4.9$ | $75.3 \pm 13.5$ | $65.7 \pm 2.7$ |
| SPSD-LEM | 0.25 | $\mathbf{72.8 \pm 7.1}$ | $\mathbf{50.7 \pm 13.9}$ | $\mathbf{85.3 \pm 5.3}$ | $\mathbf{76.0 \pm 14.1}$ | $\mathbf{66.6 \pm 2.6}$ |
| | 0.50 | $72.5 \pm 6.6$ | $50.6 \pm 14.2$ | $84.5 \pm 5.8$ | $75.1 \pm 12.9$ | $66.5 \pm 1.9$ |
| | 0.75 | $72.7 \pm 7.4$ | $49.5 \pm 12.9$ | $84.3 \pm 4.8$ | $74.7 \pm 14.3$ | $66.2 \pm 1.7$ |
| SPSD-LCM | 0.25 | $72.1 \pm 7.4$ | $49.9 \pm 13.1$ | $\mathbf{85.1 \pm 4.8}$ | $74.9 \pm 12.6$ | $67.6 \pm 2.1$ |
| | 0.50 | $\mathbf{72.9 \pm 6.7}$ | $48.4 \pm 13.3$ | $84.1 \pm 5.6$ | $74.4 \pm 13.7$ | $68.1 \pm 1.6$ |
| | 0.75 | $71.6 \pm 6.1$ | $\mathbf{50.1 \pm 12.8}$ | $84.1 \pm 5.7$ | $\mathbf{75.0 \pm 12.9}$ | $\mathbf{68.7 \pm 1.5}$ |

### E.5.2 Ablations on the matrix power normalization

We conduct ablation experiments to assess the impact of the power normalization parameter $\theta$ on the performance of the proposed GyroAtt, as summarized in Tab. 20. For each gyro structure, we let the parameter $\theta$ vary within the set $\{0.25, 0.50, 0.75\}$. Among the SPD-based configurations, our GyroAttNet under SPD-AIM geometry achieves the highest inter-session accuracy on the BNCI2014001 dataset and the best inter-subject accuracy on the BNCI2015001 dataset at $p = 0.5$. For the SPSD-based settings, SPSD-LEM geometry consistently performs well across multiple metrics, especially for the inter-session scenario in BNCI2015001, where it achieves a top accuracy of $85.3\%$. It also

can be noted that smaller or larger values of $p$ (*e.g.*, 0.25 or 0.75) tend to yield lower accuracy in most cases. In contrast, a moderate value of $p = 0.5$ appears to be more suitable for both SPD and SPSD geometries, as it could maintain a good normalization power. Besides, GyroAtt tends to be less sensitive to changes in $\theta$ across all experimental scenarios. In short, these results confirm the effectiveness of the introduced matrix power normalization in classification.

### E.5.3 Ablation on the number of GyroAtt blocks

We perform an ablation study to assess the impact of stacking multiple GyroAtt blocks. Tab. 21 reports results on BNCI2014001 and BNCI2015001 under inter-session settings, with the number of GyroAtt blocks varying from 1 to 3.

For both SPD and SPSD variants, using two blocks slightly improves performance over one, but adding a third block provides no further gain and even causes mild degradation. These results suggest that a single GyroAtt block is sufficient to model EEG signals effectively, likely due to the low dimensionality and limited temporal complexity of the data. Extended results are included in the supplementary material.

Table 21: Ablation on the number of GyroAtt blocks. Performance (%) on BNCI2014001 and BNCI2015001 under inter-session settings.

| Method | Number of Blocks | BNCI2014001 | BNCI2015001 |
|---|---|---|---|
| GyroAtt-SPD | 1 | $75.4 \pm 7.1$ | $\mathbf{86.2 \pm 6.5}$ |
| GyroAtt-SPD | 2 | $\mathbf{76.5 \pm 6.7}$ | $86.1 \pm 6.2$ |
| GyroAtt-SPD | 3 | $76.0 \pm 6.3$ | $86.0 \pm 6.1$ |
| GyroAtt-SPSD | 1 | $72.9 \pm 6.7$ | $84.7 \pm 6.6$ |
| GyroAtt-SPSD | 2 | $\mathbf{73.1 \pm 6.2}$ | $\mathbf{84.9 \pm 5.2}$ |
| GyroAtt-SPSD | 3 | $72.9 \pm 5.1$ | $84.8 \pm 5.3$ |

### E.6 Implementation details of gyro inner product.

In Euclidean space, attention mechanisms commonly use the inner product as a similarity measure. Nguyen and Yang [52], Nguyen et al. [53] extends this concept by defining the inner product on SPD, SPSD, and Grassmannian manifolds. The specific formulations are detailed as follows:

For $\mathbf{P}, \mathbf{Q} \in \mathcal{S}_d^{++}$, the SPD inner product is given by [52]:

$$\langle \mathbf{P}, \mathbf{Q} \rangle^g = \langle \mathrm{Log}_{\mathbf{I}_d}^g(\mathbf{P}), \mathrm{Log}_{\mathbf{I}_d}^g(\mathbf{Q}) \rangle_{\mathbf{I}_d}^g, \tag{27}$$

For $\mathbf{U}, \mathbf{V} \in \widetilde{\mathcal{G}}(q, d)$, the inner product is given by:

$$\langle \mathbf{U}, \mathbf{V} \rangle^{gr} = \langle \widetilde{\mathrm{Log}}_{\widetilde{\mathbf{I}}_{d,q}}^{gr}(\mathbf{U}), \widetilde{\mathrm{Log}}_{\widetilde{\mathbf{I}}_{d,q}}^{gr}(\mathbf{V}) \rangle_{\widetilde{\mathbf{I}}_{d,q}}, \tag{28}$$

For $(\mathbf{U}_P, \mathbf{S}_P), (\mathbf{U}_Q, \mathbf{S}_Q) \in \widetilde{\mathcal{G}}(q, d) \times \mathcal{S}_q^{++}$, the inner product is defined as:

$$\langle (\mathbf{U}_P, \mathbf{S}_P), (\mathbf{U}_Q, \mathbf{S}_Q) \rangle^{psd,g} = \lambda \langle \mathbf{U}_P \mathbf{U}_P^\top, \mathbf{U}_Q \mathbf{U}_Q^\top \rangle_{\widetilde{\mathbf{I}}_{d,q}}^{gr} + \langle \mathbf{S}_P, \mathbf{S}_Q \rangle_{\mathbf{I}_q}^g, \tag{29}$$

We replaced the distance-based similarity computation in Eq. (5) with the inner product defined in follow and conducted ablation experiments on the MAMEM, BNCI2014001, and BNCI2015001 datasets under inter-session settings.

### E.7 Implementation details of replacing geometric components in GyroAtt

We conducted an ablation study to evaluate the contributions of the Gyro Homomorphism and nonlinear activation in GyroAtt. Specifically, we replaced these components in GyroAtt-SPD and GyroAtt-SPSD with equivalent layers from SPDNet and GrNet, such as Bimap, Frmap, and ReEig, to assess their impact on performance.

**Implementation of component replacement on GyroAtt.** We replaced components in GyroAtt with their equivalents from MAtt and GDLNet to assess their contributions. Specifically, in GyroAtt-SPD, we replaced the Gyro Homomorphism $\hom(\cdot)$ with the Bimap layer and the matrix power-based nonlinear activation $\sigma(\cdot)$ with the ReEig layer. In GyroAtt-SPSD, we replaced $\hom(\cdot)$ with the Frmap layer and $\sigma(\cdot)$ with the ReEig layer. That is, we substituted $\hom_{gr}(\mathbf{U}_P)$ in $\hom_{psd,g}(\cdot)$ with the Frmap layer and replaced $(\mathbf{S}_{R_i})^p$ in $(\mathbf{U}_{R_i}, (\mathbf{S}_{R_i})^p)$ with the ReEig layer.

The BiMap (bilinear transformation) layer is defined as:

$$\mathbf{X}^{(l)} = \mathbf{W}^{(l)}\mathbf{X}^{(l-1)}\mathbf{W}^{(l)^\top}, \tag{30}$$

where $\mathbf{X}^{(l)} \in \mathcal{S}_{d2}^{++}, \mathbf{X}^{(l-1)} \in \mathcal{S}_{d1}^{++}, \mathbf{W}^{(l)} \in \mathbb{R}^{d_2 \times d_1}$ with $d_1 > d_2$ is a semi-orthogonal matrix. For the parameter $\mathbf{W}^{(l)}$, we use the geoopt [43] package to optimize. The FrMap layer is defined as:

$$\mathbf{X}^{(l)} = \mathbf{W}^{(l)^\top}\mathbf{X}^{(l-1)}, \tag{31}$$

where $\mathbf{X}^{(l)} \in \mathcal{G}(d_2, q), \mathbf{X}^{(l-1)} \in \mathcal{G}(d_1, q)$, and $\mathbf{W}^{(l)} \in \mathbb{R}^{d_2 \times d_1}$ is a semi-orthogonal matrix with $d_1 > d_2$. We optimized $\mathbf{W}^{(l)}$ using Geoopt.

The ReEig (rectified eigenvalues activation) layer is defined as:

$$\mathbf{X}^l = \mathbf{U}^{(l)}\max(\mathbf{\Sigma}^{(l)}, \epsilon\mathbf{I}_d)\mathbf{U}^{(l)^\top}, \tag{32}$$

with $\mathbf{X}^{l-1} = \mathbf{U}^{(l)}\mathbf{\Sigma}^{(l)}\mathbf{U}^{(l)^\top}$, where $\mathbf{\Sigma}^{(l)}$ contains the eigenvalues of $\mathbf{X}^{l-1}$, and $\epsilon\mathbf{I}_d$ is used to ensure numerical stability and set by 1e-4. Here, we set the dimensions of the Bimap layer to $21 \times 18$, $43 \times 20$, and $44 \times 20$ and the frmap layer to $21 \times 18$, $43 \times 30$, and $44 \times 30$ for the MAMEM-SSVEP-II, BNCI2014001, and BNCI2015001 datasets, respectively.

As shown in Tab. 8 and Tab. 22, replacing $\hom(\cdot)$ with the Bimap layer or $\sigma(\cdot)$ with the ReEig layer leads to significant performance degradation across the datasets. Similarly, for GyroAtt-SPSD, replacing $\hom(\cdot)$ with Frmap or $\sigma(\cdot)$ with ReEig degrades performance. This occurs because $\hom(\cdot)$ and $\sigma(\cdot)$ respect the gyro algebraic structure and underlying Riemannian geometry. The $\hom(\cdot)$ function, as a Gyro homomorphism, preserves the Gyro algebraic structure of $\oplus$ and $\otimes$, serving as a natural generalization of linear transformations in Euclidean spaces. In contrast, Bimap lacks these properties. Similarly, $\sigma(\cdot)$ introduces nonlinearity to SPD matrices and, more importantly, acts as an activation and deformation mechanism for the Riemannian metric, as discussed in Chen et al. [17]. On the other hand, to some extent, ReEig is primarily a numerical activation method, ensuring only $\mathcal{S}_d^{++} \to \mathcal{S}_d^{++}$ without addressing these deeper structural and geometric considerations.

Table 22: Ablations of GyroAtt-SPSD, Replacing Gyro Homomorphisms and Power Activations with SPDNet or GrNet methods (The Frmap and ReEig layers).

| Transformation | Activation | BNCI2014001 | | BNCI2015001 | |
| --- | --- | --- | --- | --- | --- |
| | | Inter-session | Inter-subject | Inter-session | Inter-subject |
| Frmap | Power | $68.9 \pm 6.9$ | $51.2 \pm 12.9$ | $82.3 \pm 6.2$ | $65.8 \pm 13.1$ |
| Homomorphisms | ReEig | $72.3 \pm 6.9$ | $49.6 \pm 13.3$ | $84.9 \pm 6.2$ | $74.1 \pm 12.3$ |
| Frmap | ReEig | $68.8 \pm 7.2$ | $50.7 \pm 13.8$ | $81.6 \pm 6.1$ | $72.9 \pm 13.3$ |
| Homomorphisms | Power | $72.9 \pm 7.1$ | $52.4 \pm 15.6$ | $85.3 \pm 5.3$ | $76.0 \pm 14.1$ |

### E.8 Computational complexity analysis and comparison

The computational complexity of GyroAtt, MAtt, and GDLNet primarily depends on operations such as WFM, gyro homomorphisms, and similarity measurements. The primary computational overhead arises from the types and quantities of matrix functions used in manifold-valued computations.

### E.8.1 Number of matrix functions in manifold attention mechanisms

To analyze computational complexity, we examine the matrix functions employed in the attention mechanisms of MAtt, GDLNet, and our GyroAtt framework. These methods primarily use Singular Value Decomposition (SVD) and Cholesky decomposition, each with a computational complexity of $\mathcal{O}(d^3)$ for matrices of dimension $d$. In GyroAtt-Gr and GyroAtt-SPSD, SVD is applied to matrices sized $d \times q$, resulting in a complexity of $\mathcal{O}(dq^2)$. Tab. 23 outlines the number of required matrix functions for each method, where $C$ is the number of inputs to the attention module (e.g., queries or keys), and $q$ denotes the subspace dimension.

Table 23: Complexity comparison of different manifold attention models, where Chol denotes the Cholesky decomposition.

| Methods | Metrics | SVD ($d \times d$) $\mathcal{O}(d^3)$ | Chol ($d \times d$) $\mathcal{O}(d^3)$ | SVD ($d \times q$) $\mathcal{O}(dq^2)$ | SVD ($q \times q$) $\mathcal{O}(q^3)$ | Chol ($q \times q$) $\mathcal{O}(q^3)$ |
|---|---|---|---|---|---|---|
| MAtt | LEM | $3C$ | 0 | 0 | 0 | 0 |
| GDLNet | ONB | $C$ | 0 | 0 | 0 | 0 |
| GyroAtt-Gr | ONB | 0 | 0 | $C^2 + C$ | $2C$ | 0 |
| GyroAtt-SPD | AIM | $2C^2 + 3C$ | 0 | 0 | 0 | 0 |
| GyroAtt-SPD | LEM | $4C$ | 0 | 0 | 0 | 0 |
| GyroAtt-SPD | LCM | $C$ | $3C$ | 0 | 0 | 0 |
| GyroAtt-SPSD | AIM | 0 | 0 | $C^2 + C$ | $3C^2 + 4C$ | 0 |
| GyroAtt-SPSD | LEM | 0 | 0 | $C^2 + C$ | $C^2 + 6C$ | 0 |
| GyroAtt-SPSD | LCM | 0 | 0 | $C^2 + C$ | $C^2 + 2C$ | $3C$ |

Table 24: Training time (seconds/epoch) comparison of different manifold attention networks on the BNCI2014001 and BNCI2015001 datasets.

| Methods | Metrics | BNCI2014001 Inter-session | BNCI2014001 Inter-subject | BNCI2015001 Inter-session | BNCI2015001 Inter-subject |
|---|---|---|---|---|---|
| MAtt | LEM | 4.86 | 89.12 | 2.74 | 56.78 |
| GDLNet | ONB | 4.55 | 88.59 | 1.71 | 47.66 |
| GyroAtt-Gr | ONB | 5.28 | 100.20 | 1.98 | 54.07 |
| GyroAtt-SPD | AIM | 10.36 | 149.14 | 4.44 | 81.73 |
| GyroAtt-SPD | LEM | 7.89 | 138.59 | 4.28 | 80.66 |
| GyroAtt-SPD | LCM | 4.11 | 87.44 | 2.42 | 49.86 |
| GyroAtt-SPSD | AIM | 6.78 | 123.44 | 2.37 | 65.30 |
| GyroAtt-SPSD | LEM | 6.54 | 116.49 | 2.34 | 64.23 |
| GyroAtt-SPSD | LCM | 5.71 | 103.57 | 2.16 | 58.50 |

### E.8.2 Experimental comparison of time efficiency

For practical comparison, we measured the average training time per epoch (seconds) for our GyroAtt variants and baseline models MAtt and GDLNet on the BNCI2014001 and BNCI2015001 datasets, as shown in Tab. 24. The results indicate that GyroAtt-Gr runs slower than GDLNet, mainly because of more SVD operations in WFM, increasing computational complexity. As predicted by Tab. 23 and confirmed in Tab. 24, GyroAtt-SPSD is less computationally efficient than GDLNet and MAtt across all metrics. In contrast, GyroAtt-SPD with the LCM metric slightly outperforms MAtt in runtime. Additionally, Tab. 24 shows that any GyroAtt variant using the LCM metric runs faster than those using AIM and LEM metrics, primarily because the Cholesky decomposition (used in LCM) is generally more efficient than SVD due to smaller constant factors. Notably, AIM-based GyroAtt variants are slower than their LEM-based counterparts because AIM requires more eigenvalue computations than LEM.

Our computational complexity analysis and empirical runtime comparisons highlight key observations:

- **GyroAtt-Gr**: Runs slower than GDLNet due to more SVD operations.

- **GyroAtt-SPSD**: Less efficient than GDLNet and MAtt across all metrics, involving extra SVD computations on $d \times q$ and $q \times q$ matrices.

- **GyroAtt-SPD with LCM**: Achieves comparable or better runtime than MAtt, benefiting from the efficiency of the Cholesky decomposition.

- **Metric Impact**: The LCM metric leads to lower training times due to the efficiency of Cholesky decomposition over SVD. AIM-based GyroAtt variants are slower than LEM-based ones because AIM requires more eigenvalue operations.

Although some GyroAtt variants introduce additional computational overhead compared to baseline methods, especially under certain metrics, the performance improvements justify the increased

complexity. Selecting efficient metrics like LCM can mitigate computational costs, making GyroAtt practical for real-world applications.

### E.9  Model compactness and parameter efficiency

Table 25: Learnable parameters (in millions) of different methods across four EEG datasets.

| Method | MAMEM-SSVEP-II | BCI-ERN | BNCI2014001 | BNCI2015001 |
|---|---|---|---|---|
| MAtt | 0.07M | 0.03M | 0.03M | 0.01M |
| GDLNet | 0.51M | 0.05M | 0.03M | 0.01M |
| GyroAtt-SPD | 0.11M | 0.08M | 0.04M | 0.01M |
| GyroAtt-Gr | 0.11M | 0.07M | 0.04M | 0.01M |
| GyroAtt-SPSD | 0.12M | 0.09M | 0.05M | 0.01M |

As shown in Tab. 25, all GyroAtt variants maintain a comparable parameter count to MAtt, and are substantially more compact than GDLNet on MAMEM-SSVEP-II. Despite this, GyroAtt achieves stronger performance across all datasets.

### E.10  Long-range dependencies

A central advantage of self-attention is its ability to capture long-range dependencies via position-independent pairwise interactions, where each token attends to all others in a single step.

Our GyroAtt module inherits this property by extending self-attention to Riemannian manifolds. Specifically, it computes pairwise attention over manifold-valued features using geodesic distances, followed by aggregation via the weighted Fréchet mean. This allows any two positions to interact directly, independent of sequence length.

### E.11  Visualizing learned representations via Rie-SNE

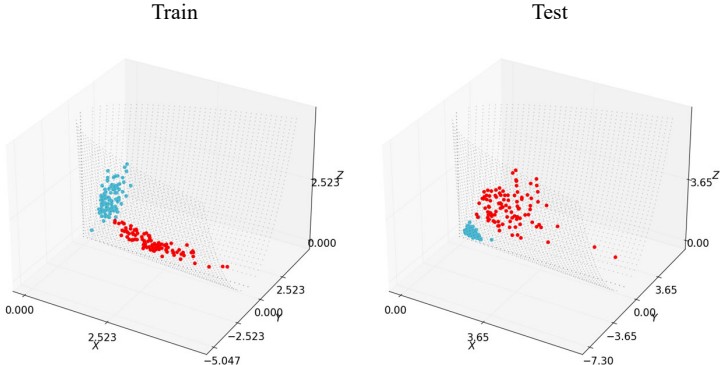

Figure 5:  Rie-SNE visualization of SPD representations extracted by GyroAtt-SPD-LEM on BNCI2015001. We plot the manifold-valued features from the training (left) and test (right) sets. Each point corresponds to SPD matrices used for final classification, with colors denoting different classes. Clear separation between classes and structural consistency across splits highlight the model's capacity to learn discriminative geometric features.

We visualize the SPD matrices output from GyroAtt-SPD-LEM prior to classification using Rie-SNE [21], separately for the training and test sets on BNCI2015001. As shown in Fig. 5, the learned representations exhibit compact intra-class clustering and clear inter-class separation across both splits.

## F  Proofs of the Theorems in the Main Paper

### F.1  Proof of the Thm. 3.2

*Proof of Thm. 3.2* .  This theorem can be induced from Nguyen and Yang [52][Lem. 2.1-2.2]. Let the Riemannian metric tensor be $g$. We have

$$f : (\mathcal{M}, f^*g) \to (\mathcal{M}, g), \text{ with } f(e) = e,$$

where $f^*g$ is the pullback metric under the diffeomorphism $f$. As $f$ is an isometry, we have $f^*g = g$. Applying Lem. 2.1-2.2 in Nguyen and Yang [52], one can immediately prove $f(\mathbf{P} \oplus \mathbf{Q}) = f(\mathbf{P}) \oplus f(\mathbf{Q})$ and $f(t \otimes \mathbf{Q}) = t \otimes f(\mathbf{Q})$ for any $t \in \mathbb{R}$ and $\mathbf{P}, \mathbf{Q} \in M$. $\qquad \square$

### F.2   Proof of the Thm. 4.1

*Proof of Thm. 4.1* .  For each $g \in \{ai, le, lc\}$, let $\left(\mathcal{S}_d^{++}, \oplus_g, \otimes_g\right)$ be the gyrovector space specified in Tab. 1. We must show that the map

$$\hom_g : \mathcal{S}_d^{++} \to \mathcal{S}_d^{++}, \mathbf{P} \mapsto \begin{cases} \mathbf{OPO}^\top, & g = ai, \ \mathbf{O} \in \mathrm{O}(d), \\ \mathrm{expm}\left(\mathbf{M}\log\mathrm{m}(\mathbf{P})\mathbf{M}^\top\right), & g = le, \ \mathbf{M} \in \mathbb{R}^{d \times d}, \\ \mathscr{L}^{-1}\left(\lfloor L(\mathbf{P})\rfloor + \exp\left(\mathbb{D}\left(L(\mathbf{P})\right)\right)\right), & g = lc, \ \mathbf{M} \in \mathbb{R}^{d \times d}, \end{cases} \quad (33)$$

with

$$L(\mathbf{P}) = \mathbf{M}\left(\lfloor \mathscr{L}(\mathbf{P})\rfloor + \lfloor \mathscr{L}(\mathbf{P})\rfloor^\top + \mathbb{D}\left(\mathscr{L}(\mathbf{P})\right)\right)\mathbf{M}^\top, \tag{34}$$

is a gyro homomorphism, *i.e.,*

$$\hom_g(\mathbf{P}) \oplus_g \hom_g(\mathbf{Q}) = \hom_g\left(\mathbf{P} \oplus_g \mathbf{Q}\right), \tag{35}$$

$$t \otimes_g \hom_g(\mathbf{P}) = \hom_g\left(t \otimes_g \mathbf{P}\right) \quad \forall \, \mathbf{P}, \mathbf{Q} \in \mathcal{S}_d^{++}, \ t > 0. \tag{36}$$

**AIM case** $(g = ai)$**.**

The $\oplus_{ai}$ and $\otimes_{ai}$ are defined by:

$$\mathbf{P} \oplus_{ai} \mathbf{Q} = \mathbf{P}^{\frac{1}{2}}\mathbf{Q}\mathbf{P}^{\frac{1}{2}}. \tag{37}$$

$$t \otimes_{ai} \mathbf{P} = \mathbf{P}^t \tag{38}$$

Let the isometry $\hom_{ai}(\mathbf{P}) = \mathbf{OPO}^\top$ with $\mathbf{O} \in O(d)$ (orthogonal group). Since the identity element is $e = \mathbf{I}$ and $\hom_{ai}(e) = \mathbf{OIO}^\top = \mathbf{I} = e$, the sufficient condition in Theorem 3.2 applies: an isometry fixing $e$ is a gyro homomorphism.

For completeness, we verify both operations explicitly.

We begin by showing that $\hom_{ai}(\cdot)$ satisfies Eq. (35). let $\hom_{ai}(\mathbf{P}) = \mathbf{OPO}^\top$, with any $\mathbf{P}, \mathbf{Q} \in \mathcal{S}_d^{++}$, then we have

$$\begin{aligned} \hom_{ai}(\mathbf{P}) \oplus_{ai} \hom_{ai}(\mathbf{Q}) &\stackrel{(1)}{=} \left(\mathbf{OPO}^\top\right)^{\frac{1}{2}} \mathbf{OQO}^\top \left(\mathbf{OPO}^\top\right)^{\frac{1}{2}} \\ &\stackrel{(2)}{=} \mathbf{OP}^{\frac{1}{2}}\mathbf{O}^\top\mathbf{OQO}^\top\mathbf{OP}^{\frac{1}{2}}\mathbf{O}^\top \\ &= \mathbf{OP}^{\frac{1}{2}}\mathbf{QP}^{\frac{1}{2}}\mathbf{O}^\top \\ &= \hom_{ai}(\mathbf{P} \oplus_{ai} \mathbf{Q}). \end{aligned} \tag{39}$$

The derivation of Eq. (39) follows.

(1) follow from Eqs. (8) and (37).

(2) follows from the fact that $\mathbf{P}$ is an SPD matrix and $\mathbf{O}$ is an orthogonal matrix.

Now, we proof that $\hom_{ai}(\cdot)$ satisfies Eq. (36). For the $\otimes_{ai}$, we have

$$\begin{aligned} t \otimes_{ai} \hom_{ai}(\mathbf{P}) &\stackrel{(1)}{=} \left(\mathbf{OPO}^\top\right)^t \\ &\stackrel{(2)}{=} \mathbf{OP}^t\mathbf{O}^\top \\ &= \hom_{ai}(t \otimes_{ai} \mathbf{P}). \end{aligned} \tag{40}$$

The derivation of Eq. (40) follows.

(1) follow from Eqs. (8) and (38).

(2) follows from the fact that $\mathbf{P}$ is an SPD matrix and $\mathbf{O}$ is an orthogonal matrix.

**LEM case** $(g = le)$**.**

The $\oplus_{le}$ and $\otimes_{le}$ are defined by:

$$\mathbf{P} \oplus_{le} \mathbf{Q} = \mathrm{expm}(\mathrm{logm}(\mathbf{P}) + \mathrm{logm}(\mathbf{Q})), \tag{41}$$

$$t \otimes_{le} \mathbf{P} = \mathbf{P}^t \tag{42}$$

We begin by showing that $\mathrm{hom}_{le}(\cdot)$ satisfies Eq. (35). For the $\oplus_{le}$, with any $\mathbf{P}, \mathbf{Q} \in \mathcal{S}_d^{++}$, we have

$$
\begin{aligned}
\mathrm{hom}_{le}(\mathbf{P}) \oplus_{le} \mathrm{hom}_{le}(\mathbf{Q}) &\overset{(1)}{=} \mathrm{expm}\left(\mathbf{M}\,\mathrm{logm}\,(\mathbf{P})\,\mathbf{M}^\top + \mathbf{M}\,\mathrm{logm}\,(\mathbf{Q})\,\mathbf{M}^\top\right) \\
&= \mathrm{expm}\left(\mathbf{M}\left(\mathrm{logm}\,(\mathbf{P}) + \mathrm{logm}\,(\mathbf{Q})\right)\mathbf{M}^\top\right) \\
&= \mathrm{hom}_{le}(\mathbf{P} \oplus_{le} \mathbf{Q}).
\end{aligned}
\tag{43}
$$

The derivation of Eq. (43) follows.

(1) follow from Eqs. (8) and (41).

For $\otimes_{le}$, we have

$$
\begin{aligned}
t \otimes_{le} \mathrm{hom}_{le}(\mathbf{P}) &\overset{(1)}{=} \left(\mathrm{expm}\left(\mathbf{M}\,\mathrm{logm}\,(\mathbf{P})\,\mathbf{M}^\top\right)\right)^t \\
&\overset{(2)}{=} \mathrm{expm}\left(t\mathbf{M}\,\mathrm{logm}\,(\mathbf{P})\,\mathbf{M}^\top\right) \\
&= \mathrm{hom}_{le}(t \otimes_{le} \mathbf{P}).
\end{aligned}
\tag{44}
$$

**Reduction to AIM for orthogonal $\mathbf{M} = \mathbf{O} \in \mathrm{O}(d)$.**

For the $\oplus_{le}$, with any $\mathbf{P}, \mathbf{Q} \in \mathcal{S}_d^{++}, \mathbf{O} \in \mathrm{O}(d)$ we have

$$
\begin{aligned}
\mathrm{hom}_{le}(\mathbf{P}) \oplus_{le} \mathrm{hom}_{le}(\mathbf{Q}) &\overset{(1)}{=} \mathrm{expm}\left(\mathbf{O}\left(\mathrm{logm}\,(\mathbf{P}) + \mathrm{logm}\,(\mathbf{Q})\right)\mathbf{O}^\top\right) \\
&\overset{(2)}{=} \mathbf{O}\,\mathrm{expm}\left((\mathrm{logm}\,(\mathbf{P}) + \mathrm{logm}\,(\mathbf{Q}))\right)\mathbf{O}^\top \\
&= \mathrm{hom}_{le}(\mathbf{P} \oplus_{ai} \mathbf{Q}).
\end{aligned}
\tag{45}
$$

The derivation of Eq. (45) follows.

(1) follow from Eqs. (41) and (43).

(2) follows from the fact that $\mathbf{P}$ is an SPD matrix and $\mathbf{O}$ is an orthogonal matrix.

For the $\otimes_{le}$, we have

$$
\begin{aligned}
t \otimes_{le} \mathrm{hom}_{le}(\mathbf{P}) &\overset{(1)}{=} \mathrm{expm}\left(t\mathbf{O}\,\mathrm{logm}\,(\mathbf{P})\,\mathbf{O}^\top\right) \\
&\overset{(2)}{=} \mathbf{O}\,\mathrm{expm}\left(t\,\mathrm{logm}\,(\mathbf{P})\right)\mathbf{O}^\top \\
&= \mathrm{hom}_{le}(t \otimes_{le} \mathbf{P}).
\end{aligned}
\tag{46}
$$

The derivation of Eq. (46) follows.

(1) follow from Eqs. (42) and (44).

(2) follows from the fact that $\mathbf{P}$ is an SPD matrix and $\mathbf{O}$ is an orthogonal matrix.

**LCM case $(g = lc)$.**

The $\oplus_{lc}$ and $\otimes_{lc}$ are defined by:

$$t \otimes_{lc} \mathbf{P} = \mathscr{L}^{-1}\left(t\lfloor\mathscr{L}(\mathbf{P})\rfloor + \mathbb{D}(\mathscr{L}(\mathbf{P}))^t\right), \tag{47}$$

$$\mathbf{P} \oplus_{lc} \mathbf{Q} = \mathscr{L}^{-1}\left(\lfloor\mathscr{L}(\mathbf{P})\rfloor + \lfloor\mathscr{L}(\mathbf{Q})\rfloor + \mathbb{D}(\mathscr{L}(\mathbf{P}))\mathbb{D}(\mathscr{L}(\mathbf{Q}))\right). \tag{48}$$

We begin by showing that $\mathrm{hom}_{lc}(\cdot)$ satisfies Eq. (35). With any $\mathbf{P}, \mathbf{Q} \in \mathcal{S}_d^{++}$, for $\oplus_{lc}$, we can rewrite $\oplus_{lc}$ and $\mathrm{hom}_{lc}$ as

$$\mathbf{P} \oplus_{lc} \mathbf{Q} = \mathscr{L}^{-1}\left(\exp\mathbb{D}\left(\log\mathbb{D}\left(\mathscr{L}(\mathbf{P})\right) + \log\mathbb{D}\left(\mathscr{L}(\mathbf{Q})\right)\right)\right), \tag{49}$$

$$\mathrm{hom}_{lc}(\mathbf{P}) = \mathscr{L}^{-1}\left(\exp\mathbb{D}\left(L(\mathbf{P})\right)\right), \tag{50}$$

where $L(\mathbf{P}) = \mathbf{M}\left(\lfloor\mathscr{L}(\mathbf{P})\rfloor + \lfloor\mathscr{L}(\mathbf{P})\rfloor^\top + \mathbb{D}(\mathscr{L}(\mathbf{P}))\right)\mathbf{M}^\top$, $\log\mathbb{D}(\mathbf{F})$ and $\exp\mathbb{D}(\mathbf{F})$ are given by

$$\log\mathbb{D}(\mathbf{F}) = \lfloor\mathbf{F}\rfloor + \mathrm{logm}(\mathbb{D}(\mathbf{F})), \tag{51}$$

$$\exp\mathbb{D}(\mathbf{F}) = \lfloor\mathbf{F}\rfloor + \mathrm{expm}(\mathbb{D}(\mathbf{F})), \tag{52}$$

Then we have

$$
\begin{aligned}
\hom_{lc}(\mathbf{P}) \oplus_{lc} \hom_{lc}(\mathbf{Q}) &\overset{(1)}{=} \mathscr{L}^{-1}\left(\exp \mathbb{D}\left(L(\mathbf{P}) + L(\mathbf{Q})\right)\right) \\
&\overset{(2)}{=} \mathscr{L}^{-1}\left(\exp \mathbb{D}\left(L(\mathbf{P} + \mathbf{Q})\right)\right) \\
&= \hom_{lc}(\mathbf{P} \oplus_{lc} \mathbf{Q})
\end{aligned}
\tag{53}
$$

The derivation of Eq. (53) follows.

(1) follow from Eqs. (8) and (48).

(2) follow from the properties of $L(\cdot)$.

Hence $\hom_g$ is a gyro homomorphism for each $g \in \{ai, le, lc\}$, completing the proof.

$\square$

### F.3  Proof of the Thm. 4.2

*Proof of Thm. 4.2 .* The $\widetilde{\oplus}_{gr}$ and $\widetilde{\otimes}_{gr}$ are defined by:

$$
\mathbf{U}\widetilde{\oplus}_{gr}\mathbf{V} = \operatorname{expm}([\operatorname{Log}_{\mathbf{I}_{d,q}}^{gr}(\mathbf{U}\mathbf{U}^\top), \mathbf{I}_{d,q}])\mathbf{V},
\tag{54}
$$

$$
t\widetilde{\otimes}_{gr}\mathbf{U} = \operatorname{expm}\left(\left[t\operatorname{Log}_{\mathbf{I}_{n,q}}^{gr}, \mathbf{I}_{d,q}\right]\right)\mathbf{I}_{d,q}
\tag{55}
$$

Let the isometry $\hom_{gr}(\mathbf{U}) = \mathbf{O}\mathbf{U}$ with $\mathbf{O} \in \mathrm{O}(d)$. The reference element $\mathbf{I}_{d,q}$ denotes the canonical subspace spanned by the first $q$ basis vectors. Since $\mathbf{O}\mathbf{I}_{d,q}$ and $\mathbf{I}_{d,q}$ represent the same subspace on $\mathcal{G}(q,d)$, $\hom_{gr}$ fixes the identity element in the quotient sense and satisfies the condition of Thm. 3.2.

For completeness, we verify that it preserves both gyroaddition and gyromultiplication below.

we begin by showing that $\hom_{gr}(\cdot)$ satisfies Eq. (35). For any $\mathbf{U}, \mathbf{V} \in \mathcal{G}(q,d)$, we have

$$
\begin{aligned}
\hom_{gr}(\mathbf{U})\widetilde{\oplus}_{gr}\hom_{gr}(\mathbf{V}) &\overset{(1)}{=} \operatorname{expm}([\operatorname{Log}_{\mathbf{I}_{n,q}}^{gr}(\mathbf{O}\mathbf{U}\mathbf{U}^\top\mathbf{O}^\top), \mathbf{I}_{n,q}])\mathbf{O}\mathbf{V} \\
&\overset{(2)}{=} \operatorname{expm}([\mathbf{O}\operatorname{Log}_{\mathbf{I}_{n,q}}^{gr}(\mathbf{U}\mathbf{U}^\top)\mathbf{O}^\top, \mathbf{O}\mathbf{I}_{n,q}\mathbf{O}^\top])\mathbf{O}\mathbf{V} \\
&= \operatorname{expm}(\mathbf{O}[\operatorname{Log}_{\mathbf{I}_{n,q}}^{gr}(\mathbf{U}\mathbf{U}^\top), \mathbf{I}_{n,q}]\mathbf{O}^\top)\mathbf{O}\mathbf{V} \\
&\overset{(3)}{=} \mathbf{O}\operatorname{expm}([\operatorname{Log}_{\mathbf{I}_{n,q}}^{gr}(\mathbf{U}\mathbf{U}^\top), \mathbf{I}_{n,q}])\mathbf{O}^\top\mathbf{O}\mathbf{V} \\
&= \mathbf{O}\operatorname{expm}([\operatorname{Log}_{\mathbf{I}_{n,q}}^{gr}(\mathbf{U}\mathbf{U}^\top), \mathbf{I}_{n,q}])\mathbf{V} \\
&= \hom_{gr}(\mathbf{U}\widetilde{\oplus}_{gr}\mathbf{V}).
\end{aligned}
\tag{56}
$$

The derivation of Eq. (56) follows.

(1) follow from Eqs. (9) and (54).

(2) follows from the fact that $\operatorname{Log}_{\mathbf{O}\mathbf{I}_{n,q}\mathbf{O}^\top}^{gr}(\mathbf{O}\mathbf{U}\mathbf{U}^\top\mathbf{O}^\top) = \mathbf{O}\operatorname{Log}_{\mathbf{I}_{n,q}}^{gr}(\mathbf{U}\mathbf{U}^\top)\mathbf{O}^\top$, and for $\mathbf{O} = \begin{bmatrix} \mathbf{O}_q & 0 \\ 0 & \mathbf{O}_{d-q} \end{bmatrix}$, $\mathbf{O}\mathbf{I}_{n,q}\mathbf{O}^\top = \mathbf{I}_{n,q}$.

(3) follows from the fact that $\mathbf{O}$ is an orthogonal matrix.

Now, we proof that $\hom_{gr}(\cdot)$ satisfies Eq. (36). The differential homomorphism $\Phi : \widetilde{\mathcal{G}}(q,d) \to \mathcal{G}(q,d), \mathbf{U} \to \mathbf{U}\mathbf{U}^\top$ exists between $\widetilde{\mathcal{G}}(q,d)$ and $\mathcal{G}(q,d)$, and $\widetilde{\otimes}_{gr}$ is derived from $\otimes_{gr}$ via this differential homomorphism. Thus, to prove that $\widetilde{\otimes}_{gr}$ satisfies Eq. (36), it suffices to show that $\otimes_{gr}$ satisfies Eq. (36). The $\otimes_{gr}$ is defined by:

$$
t \otimes_{gr} \mathbf{U} = \operatorname{expm}\left(\left[t\bar{\mathbf{U}}, \mathbf{I}_{d,q}\right]\right)\mathbf{I}_{d,q}\operatorname{expm}\left(\left[-\bar{t}\mathbf{U}, \mathbf{I}_{d,q}\right]\right)
\tag{57}
$$

For $\otimes_{gr}$, we have

$$
\begin{aligned}
t \otimes_{gr} \hom_{gr}(\mathbf{U}\mathbf{U}^\top) &= (t\widetilde{\otimes}_{gr} \hom_{gr}(\mathbf{U}))(t\widetilde{\otimes}_{gr} \hom_{gr}(\mathbf{U}))^\top \\
&\overset{(1)}{=} \exp\!m(t[\mathrm{Log}^{gr}_{\mathbf{I}_{n,q}}(\mathbf{O}\mathbf{U}\mathbf{U}^\top\mathbf{O}^\top), \mathbf{I}_{n,q}])\mathbf{I}_{n,q}\exp\!m(t[\mathrm{Log}^{gr}_{\mathbf{I}_{n,q}}(\mathbf{O}\mathbf{U}\mathbf{U}^\top\mathbf{O}^\top), \mathbf{I}_{n,q}]) \\
&\overset{(2)}{=} \mathbf{O}\exp\!m([\mathrm{Log}^{gr}_{\mathbf{I}_{n,q}}(\mathbf{U}\mathbf{U}^\top), \mathbf{I}_{n,q}])\mathbf{O}^\top\mathbf{I}_{n,q}\mathbf{O}\exp\!m([\mathrm{Log}^{gr}_{\mathbf{I}_{n,q}}(\mathbf{U}\mathbf{U}^\top), \mathbf{I}_{n,q}])\mathbf{O}^\top \\
&= \mathbf{O}\exp\!m([\mathrm{Log}^{gr}_{\mathbf{I}_{n,q}}(\mathbf{U}\mathbf{U}^\top), \mathbf{I}_{n,q}])\mathbf{I}_{n,q}\exp\!m([\mathrm{Log}^{gr}_{\mathbf{I}_{n,q}}(\mathbf{U}\mathbf{U}^\top), \mathbf{I}_{n,q}])\mathbf{O}^\top \\
&= \hom_{gr}(t \otimes_{gr} \mathbf{U}\mathbf{U}^\top).
\end{aligned}
\tag{58}
$$

Since $\otimes_{gr}$ satisfies Eq. (36), we can proof $\widetilde{\otimes}_{gr}$ satisfies Eq. (36). $\qquad\square$

### F.4   Proof of the Thm. 4.3

*Proof of Thm. 4.3* . The $\widetilde{\oplus}_{psd,g}$ and $\otimes_{psd,g}$ are defined by:

$$
(\mathbf{U}_P, \mathbf{S}_P) \oplus_{psd,g} (\mathbf{U}_Q, \mathbf{S}_Q) = (\mathbf{U}_P\widetilde{\oplus}_{gr}\mathbf{U}_Q, \mathbf{S}_P \oplus_g \mathbf{S}_Q), \tag{59}
$$

$$
t \otimes_{psd,g} (\mathbf{U}_P, \mathbf{S}_P) = (t\widetilde{\otimes}_{gr}\mathbf{U}_P, t \otimes_g \mathbf{S}_P) \tag{60}
$$

we begin by showing that $\hom_{psd,g}$ satisfies Eq. (35). As shown in Eq. (10) For any $(\mathbf{U}_P, \mathbf{S}_P), (\mathbf{U}_Q, \mathbf{S}_Q) \in \widetilde{\mathcal{G}}(q,d) \times \mathcal{S}_q^{++}$, we have:

$$
\begin{aligned}
\hom_{psd,g}((\mathbf{U}_P, \mathbf{S}_P) \oplus_{psd,g} (\mathbf{U}_Q, \mathbf{S}_Q)) &\overset{(1)}{=} \hom_{psd,g}(\mathbf{U}_P\widetilde{\oplus}_{gr}\mathbf{U}_Q, \mathbf{S}_P \oplus_g \mathbf{S}_Q) \\
&= (\hom_{gr}(\mathbf{U}_P\widetilde{\oplus}_{gr}\mathbf{U}_Q), \hom_g(\mathbf{S}_P \oplus_g \mathbf{S}_Q)) \\
&\overset{(2)}{=} (\hom_{gr}(\mathbf{U}_P)\widetilde{\oplus}_{gr}\hom_{gr}(\mathbf{U}_Q), \hom_g(\mathbf{S}_P) \oplus_g \hom_g(\mathbf{S}_Q)) \\
&\overset{(3)}{=} (\hom_{gr}(\mathbf{U}_P), \hom_g(\mathbf{S}_P)) \oplus_{psd,g} (\hom_{gr}(\mathbf{U}_Q), \hom_g(\mathbf{S}_Q)) \\
&= \hom_{psd,g}(\mathbf{U}_P, \mathbf{S}_P) \oplus_{psd,g} \hom_{psd,g}(\mathbf{U}_Q, \mathbf{S}_Q).
\end{aligned}
\tag{61}
$$

The derivation of Eq. (61) follows.

(1) follow from Eqs. (10) and (59).

(2) and (3) follow from the fact that $\hom_{gr}$ and $\hom_g$ are gyro homomorphisms.

For scalar multiplication, we have:

$$
\begin{aligned}
\hom_{psd,g}(t \otimes_{psd,g} (\mathbf{U}_P, \mathbf{S}_P)) &\overset{(1)}{=} \hom_{psd,g}(t\widetilde{\otimes}_{gr}\mathbf{U}_P, t \otimes_g \mathbf{S}_P) \\
&= (\hom_{gr}(t\widetilde{\otimes}_{gr}\mathbf{U}_P), \hom_g(t \otimes_g \mathbf{S}_P)) \\
&\overset{(2)}{=} (t\widetilde{\otimes}_{gr}\hom_{gr}(\mathbf{U}_P), t \otimes_g \hom_g(\mathbf{S}_P)) \\
&\overset{(3)}{=} t \otimes_{psd,g} (\hom_{gr}(\mathbf{U}_P), \hom_g(\mathbf{S}_P)) \\
&= t \otimes_{psd,g} \hom_{psd,g}(\mathbf{U}_P, \mathbf{S}_P).
\end{aligned}
\tag{62}
$$

The derivation of Eq. (62) follows.

(1) follow from Eqs. (10) and (60).

(2) and (3) follow from the fact that $\hom_{gr}(\cdot)$ and $\hom_g(\cdot)$ are gyro homomorphisms.

$\qquad\square$

