# OpenReview forum: "Towards a General Attention Framework on Gyrovector Spaces for Matrix Manifolds"
_NeurIPS.cc/2025/Conference — NeurIPS 2025 poster_

### Official Review · Reviewer_3rxG · 2025-07-02

**Clarity:** 2
**Significance:** 2
**Originality:** 2
**Rating:** 2
**Confidence:** 4

**Summary:**

This paper proposes a general attention mechanism tailored for matrix manifolds by leveraging operations within gyrovector spaces. It introduces a homomorphism as a fundamental component of the proposed attention framework, specifically designed for Symmetric Positive Definite (SPD), Grassmannian, and Symmetric Positive Semi-Definite (SPSD) manifolds. The effectiveness of the proposed method is empirically validated through experiments on EEG signal classification tasks.

**Questions:**

See the weaknesses section

**Ethical Concerns:**

["NO or VERY MINOR ethics concerns only"]

**Final Justification:**

### Resolved concerns
- Performance: I now agree that the performance improvement of the proposed method is not marginal compared to the baselines.

### Remaining concerns
- Meaningless framework: the proposed GyroAtt framework do not really provides insight or special results due to the difficulty on generalizing the isometry on different manifolds and the absence of theoretical and empirical evidence on homomorphism.

Due to the remaining concerns, I remain my negative score.

**Limitations:**

yes

**Paper Formatting Concerns:**

Nothing

**Quality:**

2

**Strengths And Weaknesses:**

### Strengths

- The paper is well-written and easy to follow.
- The newly proposed homomorphisms appear to be novel contributions.

### Weaknesses

- The proposed method exhibits poor performance, with overlapping confidence intervals between GyroAtt and baseline methods undermining the claimed superiority.
- Although the primary contribution is the introduction of homomorphisms, the paper lacks both theoretical justification and empirical validation demonstrating that these homomorphisms offer advantages over alternative transformations, such as multi-layer perceptrons (MLPs) with matrix manifold outputs.
- While the paper claims a unified perspective on attention mechanisms across matrix manifolds, it fails to derive meaningful insights from this unified framework. Moreover, due to the practical complexity involved in implementing homomorphisms for arbitrary matrix manifolds, the unified view has limited applicability. Consequently, the contribution effectively reduces to separately proposing attention mechanisms for three distinct manifolds rather than establishing a genuinely unified methodology.

---

> ### Author Rebuttal · Authors · 2025-07-31
>
> We thank Reviewer $\textcolor{purple}{\rm{3rxG (R4)}}$ for the careful review and the suggestive comments. Below, we address the comments in detail. 😊
>
> ## **1. GyroAtt yields consistent and statistically significant improvements over baselines.**
>
> Table A: Comparison between GyroAtt and the baseline MAtt, where * denotes $p$-vale < 0.05 and n.s. denotes "not significant".
> ||MAMEM|BCI-CHA|BNCI2014001 inter-session|BNCI2014001 inter-subject|BNCI2015001 inter-session|BNCI2015001 inter-subject|
> |-|-|-|-|-|-|-|
> gains|3.2%|3.4%|7.8%|8.9%|5.4%|14.8%|
> statistical significance|*|*|*|n.s.|n.s.|*|
>
> We emphasize that the observed performance gains are both substantial and consistent across datasets. As shown in Table A, four out of six cases show statistical significance at $p < 0.05$, based on paired t-tests across cross-validation folds. Importantly, GyroAtt consistently outperforms MAtt across all datasets, with a notable $+14.8\%$ gain in the inter-subject setting, which is particularly challenging in EEG decoding. While some confidence intervals do overlap, this is expected due to the intrinsic characteristics of EEG data, including low signal-to-noise ratios (SNR), significant inter-/intra-session domain shifts, and high variability in subject responses—all of which naturally widen confidence intervals. Despite this high-variance regime, the consistent trend and statistical significance in most cases support the robustness of GyroAtt.
>
> ## **2. Advantages of Gyro Hom over manifold MLP‑like layers**
>
> Table B: Ablation on transformation layers (mean ± std, %). Here, / means that the method cannot converge.
> |Transformation|BNCI2014001 inter-session|#Time|BNCI2014001 inter-subject|#Time|BNCI2015001 inter-session|#Time|BNCI2015001 inter-subject|#Time|
> |-|-|-|-|-|-|-|-|-|
> |BiMap|74.0±6.5|6.31|52.3±15.0|115.85|85.2±7.2|3.56|77.2±13.2|73.81|
> |SPD-FC-LEM|70.54±6.8|14.71|40.6±13.2|257.77|84.33±6.2|6.13|74.81±14.8|171.29|
> |SPD-FC-LCM|49.27±11.2|14.48|/|/|74.03±10.4|5.84|/|/|
> |Homomorphisms|75.4±7.4|7.89|53.1±14.1|138.59|86.2±4.5|4.28|77.9±13.0|80.66|
>
> ### **2.1 Comparison with SPD-FC Layers.**
>
> As reported in Table 8, we benchmark BiMap [a] as the canonical MLP‑like transformation on SPD. For a broader comparison, we also include SPD‑FC‑LE/LC [b], which implements manifold fully connected (FC) layers under Log‑Euclidean/Log‑Cholesky metrics. Taking the Log-Euclidean FC layer (SPD-FC-LE) as an example, let $v _{(i,j)}(X) = \langle \ominus _{le} P _{(i,j)} \oplus _{le} X, W _{(i,j)} \rangle ^{le}, P _{(i,j)}, W _{(i,j)} \in \text{Sym} _n ^{+,le},i \leq j,i,j = 1,\ldots,m$. Then the output of an FC layer is computed as $Y = \mathrm{expm}([y _{(i,j)}] _{i,j=1} ^m)$, where $y _{(i,j)}$ is given by
> $$
> y _{(i,j)} =
> \begin{cases}
> v _{(i,j)}(X), & \text{if } i = j \\\\
> \frac{1}{\sqrt{2}} v _{(i,j)}(X), & \text{if } i < j \\\\
> \frac{1}{\sqrt{2}} v _{(j,i)}(X), & \text{if } i > j \\\\
> \end{cases}
> $$
>
> We now show that SPD-FC-LE is essentially a Euclidean FC layer in the log domain and corresponds to a special case of Gyro Hom under LEM. Here, the LE addition reduces to Euclidean operations in the log domain. Specifically,
> $$
> \ominus_{\text{le}} P \oplus_{\text{le}} X = \mathrm{expm}(\mathrm{logm}X - \mathrm{logm} P).
> $$
> so
> $$
> v_{(i,j)}(X)= \langle\mathrm{logm} X - \mathrm{logm} P_{(i,j)}, \mathrm{logm} W_{(i,j)}\rangle = \langle\mathrm{logm} X, \mathrm{logm} W_{(i,j)}\rangle + b_{(i,j)},
> $$
> where $b_{(i,j)} = -\langle \mathrm{logm} P_{(i,j)}, \mathrm{logm} W_{(i,j)} \rangle$. This shows that SPD-FC-LE is exactly a Euclidean FC layer applied in the log domain, followed by a matrix exponential:
> $$
> F(X) = \mathrm{expm}(f(\mathrm{logm}\, X)), \quad \text{with} f \text{ is FC on } \mathrm{Sym}(n).
> $$
>
> Importantly, when $b_{(i,j)} = 0$, i.e., $P_{(i,j)} = I$, the transformation becomes a linear map. In this case, $F$ satisfies the gyro homomorphism:
> $$
> F(X \oplus_{le} Y) = F(X) \oplus_{le} F(Y), \quad F(t \otimes_{le} X) = t \otimes_{le} F(X),
> $$
> which confirms that SPD-FC-LE is a special case of Gyro Hom under LEM. Similarly, SPD-FC-LC is a special case under the LCM metric.
>
> Only replacing Gyro Hom with BiMap or SPD‑FC‑LE/LC while keeping all other architectural components unchanged, we observe that Gyro‑Hom consistently outperforms SPD‑FC‑LE/LC across four datasets (Table B).  In addition, Gyro‑Hom is significantly more efficient in computational complexity. Recall that each $v_{(i,j)}$ in SPD-FC-LE requires a pair of symmetric matrices $(P_{(i,j)}, W_{(i,j)}) \in \mathrm{Sym}(n)$, leading to $n(n+1)$ parameters per component. For $S^d\to S^d$, this amounts to $\tfrac{1}{2} n^2(n+1)^2$ parameters in total. Moreover, the computational complexity is $\mathcal{O}(n^4)$ per forward pass, in addition to the $\mathcal{O}(n^3)$ cost of matrix logarithms and exponentials. In contrast, our Gyro Hom uses only an $n\times n$ parameter with **$n^2$** parameters and $\mathcal{O}(n^3)$ flops (matrix multiplications and matrix logarithms and exponentials), matching the observed wall‑clock savings.
>
> ### **2.2 Comparison with BiMap.**
>
> To further highlight the benefits of our method beyond efficiency, we analyze its theoretical advantages over BiMap.
> - **Gyro Algebraic Structure Simplifies Manifold Layer Design**: Gyro spaces define operators such as gyro addition $\oplus$ and scalar multiplication $\otimes$, providing an algebraic framework that is both Euclidean-like and geometry-aware. This facilitates the construction of manifold layers while respecting intrinsic geometry.
> - **Respecting Gyro Algebraic Structure via Gyro Homomorphisms**: GyroAtt employs gyro homomorphisms in its transformation layers, preserving the underlying algebraic structure and Riemannian geometry. In contrast, MAtt and GDLNet rely on BiMap/FrMap, which only partially adhere to Riemannian geometry. While these layers enforce manifold constraints, they lack isometric properties, potentially distorting the intrinsic geometric structure. For example, the distance $d(S_1, S_2)$ between two SPD matrices $S_1$ and $S_2$ may not be preserved after transformation, i.e., $d(\mathrm{BiMap}(S_1), \mathrm{BiMap}(S_2)) \neq d(S_1, S_2)$. This limitation undermines their ability to fully respect and retain the original geometric properties of the data, which can negatively impact performance.
>
>
>
> ## **3. Insights and theoretical advantages**
> Table B: Comparison between the GyroAtt and Euclidean Attention.
> |Component|GyroAtt|Euclidean Attention|
> |-|-|-|
> |Transformation|G1: Gyro homomorphism|E1: Linear map (homomorphism)|
> |Attention|G2: Distance-based similarity|E2: Inner-product-based similarity|
> |Aggregation|G3: Weighted Fréchet mean|E3: Weighted arithmetic mean|
>
> - **Theoretical generality:** Gyrovector spaces naturally generalize Euclidean vector spaces. As shown in Table B, each GyroAtt component extends its Euclidean counterpart in a mathematically grounded way.
> - **Natural Extension:** When the manifold is standard Euclidean geometry, G1 and G3 in GyroAtt exactly reduce to E1 and E3 in Euclidean Attention.
> - **Scalability across geometries:** The unified formulation allowed us to instantiate GyroAtt over seven variants (three SPD, three SPSD, one Grassmann) under the consistent network archetecture, highlighting the applicability of the framework across diverse settings.
>
>
> ## **4. Principled identification of gyro homomorphisms**
> Thanks for the constructive comments. There are indeed a principled way to identify the gyro homomorphism. As gyrovector space over the manifold is normally defined on the Riemannian homogeneous space [Sec. 3, c], let us focus on the homogeneous space.
>
> **Theorem A (Sufficient Condition)** Let $\mathcal{M}$ be a Riemannian homogeneous space whose group of isometries is $G$. Besides, $(\mathcal{M},\oplus,\otimes)$ forms a gyrovector space, where $\oplus$ and $\otimes$ are defined as [Eqs. (1-3), d]. Let $e$ be the gyro identity element. Every isometry $f \in G$ that fixes the identity $f(e) = e$ is a gyro homomorphism.
>
> Intuitively, we can identify a homomorphism from the isometries, which are known for a given homogeneous space. This theorem applies to all known matrix gyrovector spaces (e.g., SPD, SPSD, and Grassmannian). For example, one the SPD manifold under the AIM, $GL(n)$ is the isometry group, each of which characterize an isometry, $f _A : \mathcal{S} _{++} ^n \in S \mapsto A S A ^\top \in \mathcal{S} _ {++} ^n$, with $A \in GL(n)$. Applying $f _{A}(I)=I$ implies that $A$ should be an orthogonal matrix. This is exactly the result in Thm. 4.1.
>
> Besides, if a manifold has a flat structure, we can directly identify the gyro homomorphism from the flat structure. This is how we obtain the homomorphism for the SPD manifold under LEM and LCM (Thm. 4.1). Specifically, for the LEM, the SPD manifold becomes a flat space under the matrix logarithm. This allows us to construct a gyro homomorphism via linear operations in the log domain. In this case, the homomorphism reduces to the form
> $$
> \mathrm{hom}_{le}(P)=\mathrm{expm}(M\mathrm{logm}(P)M^\top),
> $$
> which matches the second case in Eq. (8) of Theorem 4.1.
>
> **Proof of Thm. A** This theorem can be induced from Lem. 2.1-2.2 in [d]. Let the Riemannian metric tensor be $g$. We have
> $$
> f: (M,f^*g) \to (M,g), \text{ with } f(e) = e,
> $$
> where $f^*g$ is the pullback metric under the diffeomorphism $f$. As $f$ is an isometry, we have $f^*g=g$. Applying Lem. 2.1-2.2 in [d], one can immediately prove
> $f(P \oplus Q) = f(P) \oplus f(Q)$ and $f(t \otimes Q) = t \otimes f(Q)$ for any $t \in \mathbb{R}$ and $P,Q \in M$.
>
>
> **References**
>
> > [a] A Riemannian network for SPD matrix learning.
> >
> > [b] Matrix Manifold Neural Networks++.
> >
> > [c] The Gyro-Structure of Some Matrix Manifolds.
> >
> > [d] Building Neural Networks on Matrix Manifolds: A Gyrovector Space Approach.

---

> > ### Comment · Reviewer_3rxG · 2025-08-03
> > **Response by 3rxG**
> >
> > Thanks for the additional clarifications. The t-test demonstrating statistically significant gains on the SPD task is certainly encouraging. Nevertheless, my principal concern remains unresolved: Table B ("Comparison between the GyroAtt and Euclidean Attention") does not reveal any tangible benefit of having G1, G2, and G3 generalize E1, E2, and E3.
> >
> > There appear to be two plausible avenues for addressing this issue. First, in the spirit of the Stereographic model [1], one could learn a parameter that adaptively selects a Riemannian manifold best suited to the data, including the Euclidean space. However, Theorem A states that the relevant homomorphisms are induced by isometries, and encoding such an isometry with a single curvature-like parameter, as the Stereographic model does, seems infeasible. Consequently, the contribution remains essentially an implementation of five distinct attention layers.
> >
> > Second, one could claim that Euclidean attention succeeds because it follows a three-step recipe, i.e., (i) mapping via a homomorphism, (ii) computing inner-product similarities, and (iii) aggregating with a weighted arithmetic mean, and that simply reflecting the recipe to gyrovector spaces should likewise be effective. Yet the manuscript offers no theoretical analysis to substantiate this claim. Empirically, Table B (“Ablation on transformation layers”) shows that substituting BiMap with the proposed homomorphism yields only negligible differences, and the influence of the homomorphism is not investigated on manifolds other than SPD.
> >
> > For these reasons, the previous score is maintained.
> >
> > [1] Constant curvature graph convolutional networks, ICML 2020

---

> ### Author Response · Authors · 2025-08-07
> **Additional Response to Reviewer 3rxG (Part 1)**
>
> We sincerely thank you for the thoughtful comments, constructive suggestions, and the time dedicated to reviewing our paper.
>
> ## 1. Homomorphism over the stereographic model
>
> We sincerely thank the reviewer for their insightful comments. The stereographic model $\mathfrak{st}_\kappa^d$ also admits gyrovector structure [a, Eqs.(2-3)]. Therefore, we can readily identify its gyro-homomorphism by our Theorem A. It turns out that the homomorphism is independent of curvature. This is not surprising, as gyroaddition and scalar gyromultiplication share the same expressions, except for the curvature parameter $\kappa$.
>
> In particular, Theorem 12 in [a] shows that every isometry $\phi: \mathfrak{st} _\kappa ^d \to \mathfrak{st} _\kappa ^d$ over the stereographic model can be written as
> $$
> \phi(x)=z \oplus _\kappa R x, \quad \text { where } z \in  \mathfrak{st} _\kappa ^d \; \mathrm{and}\; R \in \mathrm{O}(d).
> $$
> Wherein, $\mathrm{O}(d)$ is the group of $d \times d$ orthogonal matrices, and $\oplus _\kappa$ represents the gyroaddition. By applying Theorem A,
> $$
> \phi(\mathbf{0})=\mathbf{0}, \text{ with } \mathbf{0} \text{ as the zero vector, }
> $$
> we derive the following gyro-homomorphism for $\mathfrak{st} _\kappa ^d$:
> $$
> \mathrm{Hom}(x) = R x, \quad \text{where } x \in \mathfrak{st} _\kappa^d, \forall \kappa \in \mathbb{R} \text{ and } R \in \mathrm{O}(d).
> $$
> Importantly, this expression is independent of the curvature $\kappa$, applicable to Euclidean ($\kappa = 0$), hyperbolic ($\kappa < 0$), and spherical ($\kappa > 0$) settings.
>
> Please note that in GyroAtt, the curvature parameter $\kappa$ still plays a critical role in other components of the attention mechanism. Since gyro distance is curvature-dependent, $\kappa$ directly affects both the query–key similarity computation and the weighted Fréchet mean (WFM) aggregation.

---

> ### Author Response · Authors · 2025-08-07
> **Additional Response to Reviewer 3rxG (Part 2)**
>
> ## 2. More Comparisons with MLP-like Layers
>
> Table A: Summary of differrnt MLP-like layers on the SPD Manifold
> |Method|Tangent-Space Based|
> |-|-|
> |SPD-Tangent-FC|Yes|
> |SPD-FC [d, Sec. 3.2.1]|No|
> |BiMap [e, Sec. 3.1]|No|
> |Gyro Homomorphism|No|
>
> Table B : Summary of differrnt MLP-like layers on the Grassmannian Manifold
> |Method|Tangent-Space Based|
> |-|-|
> |Grassmann-Tangent-FC|Yes|
> |Gr-Scaling [c, Sec. 3.2]|Yes|
> |GDLNet [g]|No|
> |FrMap+ReOrth [b]|No|
> |Gyro Homomorphism|No|
>
> We appreciate the reviewer’s insightful suggestions. Since the SPSD manifold currently lacks established MLP-like layers, we compare our gyro homomorphism against different MLP-like layers on the SPD and Grassmann manifolds. Following the Möbius-type matrix-vector multiplication [a, Defs. 1-2],  we also construct the Tangent-FC map $F:\mathcal{M} \to \mathcal{M}$ via the tangent space at the identity:
> $$
> F(S)=
> \mathrm{Exp} _{I}\Bigl(
> \operatorname{mat}(
> f(\operatorname{vec}(\mathrm{Log} _{I}(S)))
> )\Bigr), \forall S \in \mathcal{M}
> $$
> where $\mathrm{Log} _{I}(\cdot)$ and $\mathrm{Exp} _{I}(\cdot)$ are the Riemannian Log and Exp taken at the identity, $\mathrm{vec}(\cdot)$ the vectorisation, and $\operatorname{mat}(\cdot)$ its inverse that reshapes a vector back to a matrix.
>
> We summarize the experimental results below. Please refer to the following sections for detailed comparisons.
> - SPD Manifolds: GyroAtt-SPD equipped with the GyroHom layer consistently delivers the highest accuracy across all SPD-based transformation layers, while significantly reducing runtime compared to SPD-FC and SPD-Tangent-FC.
> - Grassmannian Manifolds：GyroAtt-Gr with the GyroHom layer achieves superior accuracy over all Grassmannian transformations, and demonstrates better efficiency than Gr-Scaling and Gr-Tangent-FC.
>
> ### 2.1 SPD Manifold Comparison
>
> Table C: Ablation on transformation layers (mean ± std, %) for SPD manifolds under LEM. Here, / means that the method cannot converge.
> |Transformation|BNCI2014001 inter-session|#Time|BNCI2014001 inter-subject|#Time|BNCI2015001 inter-session|#Time|BNCI2015001 inter-subject|#Time|
> |-|-|-|-|-|-|-|-|-|
> |SPD-Tangent-FC-LEM|42.4±6.5|14.2|/|/|71.5±10.2|6.01|/|/|
> |SPD-FC-LEM|70.5±6.8|14.71|40.6±13.2|257.77|84.3±6.2|6.13|74.81±14.8|171.29|
> |GyroAtt-SPD-LEM-BiMap|74.0±6.5|6.31|52.3±15.0|115.85|85.2±7.2|3.56|77.2±13.2|73.81|
> |GyroAtt-SPD-LEM-Hom|**75.4±7.4**|7.89|**53.1±14.1**|138.59|**86.2±4.5**|4.28|**77.9±13.0**|80.66|
>
>
> Table D: Ablation on transformation layers (mean ± std, %), for SPD manifolds under LCM. Here, / means that the method cannot converge.
> |Transformation|BNCI2014001 inter-session|#Time|BNCI2014001 inter-subject|#Time|BNCI2015001 inter-session|#Time|BNCI2015001 inter-subject|#Time|
> |-|-|-|-|-|-|-|-|-|
> |SPD-Tangent-FC-LCM|43.4±12.2|14.02|/|/|72.13±11.3|5.62|/|/|
> |SPD-FC-LCM|49.2±11.2|14.48|/|/|74.03±10.4|5.84|/|/|
> |GyroAtt-SPD-LCM-BiMap|73.9±5.8|5.81|48.3±15.9|105.67|83.2±5.3|3.01|73.2±15.3|62.42|
> |GyroAtt-SPD-LCM-Hom|**74.2±7.8**|4.11|**52.4±15.6**|87.44|**84.7±6.6**|2.42|**75.5±13.8**|49.86|
>
>
> Table E: Ablation on transformation layers (mean ± std, %), for SPD manifolds under AIM. Here, / means that the method cannot converge. “N/A”: not completed due to SPD-FC’s $\mathcal{O}(d^5)$ cost per forward pass.
> |Transformation|BNCI2014001 inter-session|#Time|BNCI2014001 inter-subject|#Time|BNCI2015001 inter-session|#Time|BNCI2015001 inter-subject|#Time|
> |-|-|-|-|-|-|-|-|-|
> |SPD-Tangent-FC-AIM|42.45±6.5|14.2|/|/|71.5±10.2|6.01|/|/|
> |SPD-FC-AIM|N/A|N/A|N/A|N/A|84.91±5.9|223.27|N/A|N/A|
> |GyroAtt-SPD-AIM-BiMap|73.7±8.2|8.21|45.3±12.9|126.70|85.1±6.2|4.02|74.1±15.6|73.69|
> |GyroAtt-SPD-AIM-Hom|**75.4±7.1**|10.36|**53.1±14.8**|149.14|**86.2±4.5**|4.44|**77.9±13.0**|81.73|
>
>
> Table F: Comparison of MAtt (mean ± std, %).
> |Transformation|BNCI2014001 inter-session|BNCI2014001 inter-subject|BNCI2015001 inter-session|BNCI2015001 inter-subject|
> |-|-|-|-|-|
> |MAtt|66.5±8.9|45.3±11.3|80.8±14.8|63.1±10.1|
> |GyroAtt-SPD|**75.4±7.1**|**53.1±14.8**|**86.2±4.5**|**77.9±13.0**|

---

> ### Author Response · Authors · 2025-08-07
> **Additional Response to Reviewer 3rxG (Part 3)**
>
> We substitute the gyro-homomorphism (GyroHom) layer in SPD GyroAtt with the existing SPD MLP-like layers, such as BiMap, SPD FC, and Tangent-FC. We compare them under LEM, LCM, and AIM metrics.
>
> - **BiMap.** Across all datasets and metrics, GyroHom outperforms BiMap in accuracy. As shown in Table E, under the AIM metric, GyroHom achieves **+7.8%** improvement in the BNCI2014001 inter-subject task (**53.1% vs. 45.3%**) and **+3.8%** in the BNCI2015001 inter-subject setting (**77.9% vs. 74.1%**).
>
> - **Tangent-FC.** As shown in Tables C–E, Tangent-FC fails to converge in several settings and underperforms GyroHom by large margins where results are available (e.g., **42.4% vs. 75.4%** under AIM, inter-session). Since both LEM and AIM employ the same Log and Exp maps at the identity, Tangent-FC yields identical results under these two metrics. Therefore, we report it only once. GyroHom is significantly more stable and effective than Tangent-FC.
>
> - **SPD-FC.** While SPD-FC can achieve competitive accuracy, its computational cost is prohibitive. For each mapping $S^d \to S^d$, the SPD-FC-AIM layer requires performing $2d(d+1)$ SVDs of size $d \times d$ per forward pass, resulting in **N/A** due to excessive runtime. In contrast, GyroHom achieves better accuracy with significantly lower runtime e.g., **4.44s vs. 223.27s** under AIM, BNCI2015001 inter-session.
>
> - **MAtt.** As shown in **Table D**, GyroHom also outperforms the widely used **MAtt** baseline across all tasks. Under the **inter-subject** settings, GyroHom improves performance by a notable margin:**+7.8%** on BNCI2014001 (**53.1% vs. 45.3%**), **+14.8%** on BNCI2015001 (**77.9% vs. 63.1%**).
>
> These results confirm that GyroHom offers a superior trade-off between accuracy, efficiency, and stability compared to existing SPD manifold layers.
>
> ### 2.2 Grassmannian Manifold Comparison
>
> Table E: Ablation on transformation layers (mean ± std, %). Here, / means that the method cannot converge.
> |Transformation|BNCI2014001 inter-session|#Time|BNCI2014001 inter-subject|#Time|BNCI2015001 inter-session|#Time|BNCI2015001 inter-subject|#Time|
> |-|-|-|-|-|-|-|-|-|
> |GDLNet|58.1±8.9|4.55|46.3±5.1|88.59|76.9±13.6|1.71|63.3±14.2|47.66|
> |GyroAtt-Gr-FrMap+ReOrth|67.2±6.4|4.84|49.1±14.9|92.1|81.9±7.2|1.79|72.3±14.7|49.22|
> |GyroAtt-Gr-Tangent-FC|61.5±6.0|9.89|46.3±13.8|188.34|78.7±6.5|2.57|73.4±13.0|71.23|
> |GyroAtt-Gr-Scaling|61.76±6.1|9.68|/|/|79.2±7.1|2.43|/|/|
> |GyroAtt-Gr|**72.5±7.3**|5.28|**52.1±14.2**|100.20|**85.0±7.7**|1.98|**75.3±13.7**|54.07|
>
> We additionally evaluate GyroAtt on the Grassmannian manifold by comparing it with two MLP-like transformation layers: FrMap+ReOrth [b, Secs. 3.1-2], Tangent-FC and Gr-Scaling [c, Sec. 3.2]. Gr-Scaling is defined as:
>
> $$
> A \widetilde{\otimes} _{gr} P = \mathrm{expm} (
> \begin{bmatrix}
> 0 & A * B \\\\ - (A * B)^\top & 0
> \end{bmatrix}) \widetilde{I} _{n, p}
> $$
>
>
> where:$A \in \mathbb{R} ^{p \times (n-p)}$, $B \in \mathbb{R} ^{(n-p) \times p}$. $P$ is defined as:
> $$
> \mathbf{P} = \mathrm{expm} \left(
> \begin{bmatrix}
> 0 & B \\\\ - B^T & 0
> \end{bmatrix}
> \right) \widetilde{\mathbf{I}} _{n, p}
> $$
>
> As shown in **Table E**, GyroAtt consistently achieves higher accuracy, lower runtime, and better stability across all tasks.
> - **GDLNet.** GyroAtt achieves much higher accuracy than GDLNet across all tasks ,e.g., **+14.4%** on BNCI2014001 inter-session, **+12.0%** on BNCI2015001 inter-subject, with comparable runtime.
> - **FrMap+ReOrth.** GyroAtt consistently outperforms FrMap across all tasks, with **+5.3%**, **+3.1%**, and **+3.0%** improvements on BNCI2014001 inter-session, BNCI2015001 inter-session, and BNCI2015001 inter-subject, respectively.
> - **Tangent-FC.** GyroAtt is both faster and more accurate. On BNCI2014001 inter-subject, it achieves **+5.8%** higher accuracy (**52.1% vs. 46.3%**) with nearly **2× lower runtime** (**100.2s vs. 188.3s**), and shows slightly lower variance.
> - **Gr-Scaling.** Gr-Scaling fails to converge on inter-subject tasks and performs worse where it does run (**79.2% vs. 85.0%**). It also incurs higher cost (**9.68s vs. 5.28s**), indicating lower efficiency.
>
> GyroAtt achieves the best overall trade-off between **accuracy, speed, and stability**, outperforming all baselines on the Grassmann manifold.
>
>
> **References**
>
> > [a] Constant curvature graph convolutional networks
> >
> > [b] Building deep networks on Grassmann manifolds
> >
> > [c] Building Neural Networks on Matrix Manifolds: A Gyrovector Space Approach
> >
> > [d] Matrix Manifold Neural Networks++
> >
> > [e] A Riemannian network for SPD matrix learning
> >
> > [f] MAtt: A manifold attention network for EEG decoding

---

> ### Author Response · Authors · 2025-08-07
> **Sincere Thanks**
>
> Dear Reviewer 3rxG,
>
> We sincerely thank you for the thoughtful comments and the time dedicated. Due to the number of follow-up experiments required to address the two main points raised, our response was submitted slightly later. Based on your feedback, we have carefully addressed the key concerns as follows:
>
> 1. **Gyro-Homomorphism over the Stereographic Model:** We show that our gyro-homomorphism naturally extends to the stereographic model and is curvature-independent.
>
> 2. **Additional Comparisons with MLP-like Layers:** We added experiments comparing GyroHom with more SPD and Grassmannian MLP-like transformations.
>
> We hope these additions (please see our responses in Parts 1, 2, and 3) further clarify the theoretical scope and empirical effectiveness of our approach. If you have any further questions or suggestions, we would be more than happy to continue the discussion.
>
> Thanks again
>
> Warm regards

---

> > ### Author Response · Authors · 2025-08-09
> > **Gentle Reminder**
> >
> > Dear Reviewer 3rxG,
> >
> > As the author–reviewer discussion will end in less than 24 hours, we kindly ask if you could review our latest response and let us know whether it addresses your concerns. Your feedback is especially valuable and helpful, and we would greatly appreciate your reconsideration if the clarifications are satisfactory.
> >
> > Thank you for your time and support.

---

> > > ### Comment · Reviewer_3rxG · 2025-08-09
> > > **Re-response**
> > >
> > > Thanks for the response. However, still the following two issues are remaining.
> > >
> > > First, it is unclear about the intention of “homomorphism over the stereographic model” in the additional response. The reference of the stereographic model was meant to highlight prior work that proposed a unified framework for constant-curvature spaces, in which the **optimization process could dynamically traverse between different manifolds depending on the data**. However, in the additional response, there is no evidence of a unified optimization framework that enables movement between gyrovector spaces, e.g., SPD, SPSD, and Gr, in practice. As a result, the suggested generalization framework still fails to provide meaningful insights to the community.
> > >
> > > Second, when the standard deviation intervals of the reported results overlap with the baselines, confidence in the claimed advantage of the proposed method is significantly diminished. In such cases, the observed differences are likely to fall within statistical noise, and the results cannot be considered strong evidence of a meaningful improvement. Therefore, if the goal is to demonstrate significant improvement over the baselines, it is crucial to adopt a more rigorous evaluation, e.g., either by **reporting confidence intervals** or by **conducting appropriate statistical significance tests** as done in the earlier responses. This would provide a more reliable basis for validating the method’s superiority.

---

> > > > ### Author Response · Authors · 2025-08-09
> > > >
> > > > ## 1. Intention of “Homomorphism over the Stereographic Model”
> > > >
> > > > The intention of “Gyro homomorphism over the stereographic model” is to show that our GyroAtt can also be applied into the stereographic model. All components are summarized in Table A.
> > > >
> > > > Table A: Components of GyroAtt in the stereographic model.
> > > > |Component|Operation|
> > > > |-|-|
> > > > |Transformation|$R x, R \in \mathrm{O}(d)$|
> > > > |Distance|$d_ \kappa (x,y) = 2\|\kappa\|^ {-\frac12} \tan^ {-1} _ \kappa \|\| -x\oplus_ \kappa y\|\|$|
> > > > |Aggregation|Weighted Fréchet mean|
> > > >
> > > > ## 2. Applicability of Dynamic Switching Across Manifolds
> > > >
> > > > - The stereographic model shares the same expressions for Riemannian operators and adopts a unified modeling approach, e.g., $\exp_ 0(x)$. Consequently, different stereographic models can be dynamically switched by $\kappa$.
> > > >
> > > > - In contrast, the matrix manifolds employ different modeling approaches, as summarized in the Table. B. Therefore, dynamically switching among them within a single model may not offer clear conceptual benefits. Nevertheless, one could potentially select between SPD and SPSD manifolds in practice based on the rank of the covariance matrix.
> > > >
> > > > Table B: Summary of Manifolds and Their Modeling Approaches.
> > > > |Manifold|Modeling approaches|
> > > > |-|-|
> > > > SPD|Compute the full-rank covariance matrix of $X$|
> > > > Grassmannian|Select the top $p$ eigenvectors of covariance matrix of $X$|
> > > > SPSD|Compute the rank-deficient covariance matrix of $X$|
> > > >
> > > >
> > > > ## 3. Statistical Significance Analysis
> > > >
> > > > Table C: Comparison between GyroAtt-SPD-LEM-Hom and other transformation layers under LEM, where * denotes $p$-vale < 0.05 and n.s. denotes "not significant".
> > > > |Method|BNCI2014001 inter-session|BNCI2014001 inter-subject|BNCI2015001 inter-session|BNCI2015001 inter-subject|
> > > > |-|-|-|-|-|
> > > > |SPD-Tangent-FC-LEM|$*$|/|$*$|/|
> > > > |SPD-FC-LEM|$*$|$*$|n.s.|n.s.|
> > > > |GyroAtt-SPD-LEM-BiMap|n.s.|n.s.|n.s.|n.s.|
> > > >
> > > >
> > > >
> > > > Table D: Comparison between GyroAtt-SPD-LCM-Hom and other transformation layers under LCM , where * denotes $p$-vale < 0.05 and n.s. denotes "not significant".
> > > > |Method|BNCI2014001 inter-session|BNCI2014001 inter-subject|BNCI2015001 inter-session|BNCI2015001 inter-subject|
> > > > |-|-|-|-|-|
> > > > |SPD-Tangent-FC-LEM|$*$|/|$*$|/|
> > > > |SPD-FC-LEM|$*$|$*$|n.s.|n.s.|
> > > > |GyroAtt-SPD-LEM-BiMap|n.s.|n.s.|n.s.|n.s.|
> > > >
> > > >
> > > >
> > > > Table E: Comparison between GyroAtt-SPD-AIM-Hom and other transformation layers under AIM, where * denotes $p$-vale < 0.05 and n.s. denotes "not significant".
> > > > |Method|BNCI2014001 inter-session|BNCI2014001 inter-subject|BNCI2015001 inter-session|BNCI2015001 inter-subject|
> > > > |-|-|-|-|-|
> > > > |SPD-Tangent-FC-AIM|$*$|/|$*$|/|
> > > > |SPD-FC-AIM|/|/|n.s.|/|
> > > > |GyroAtt-SPD-AIM-BiMap|n.s.|n.s.|n.s.|n.s.|
> > > >
> > > >
> > > > Table F: Comparison between GyroAtt-SPD and the baseline MAtt, where * denotes $p$-vale < 0.05 and n.s. denotes "not significant".
> > > > |Method|BNCI2014001 inter-session|BNCI2014001 inter-subject|BNCI2015001 inter-session|BNCI2015001 inter-subject|
> > > > |-|-|-|-|-|
> > > > |MAtt|$*$|n.s.|n.s.|$*$|
> > > >
> > > >
> > > >
> > > > Table G: Comparison between GyroAtt-Gr-Hom and other transformation layers under Grassmannian manifold, where * denotes $p$-vale < 0.05 and n.s. denotes "not significant".
> > > > |Method|BNCI2014001 inter-session|BNCI2014001 inter-subject|BNCI2015001 inter-session|BNCI2015001 inter-subject|
> > > > |-|-|-|-|-|
> > > > |GDLNet|$*$|n.s.|$*$|$*$|
> > > > |GyroAtt-Gr-FrMap+ReOrth|$*$|n.s.|n.s.|n.s.|
> > > > |GyroAtt-Gr-Tangent-FC|$*$|n.s.|$*$|n.s.|
> > > > |GyroAtt-Gr-Scaling|$*$|/|$*$|/|
> > > >
> > > >
> > > > As shown in Tables C–G, we conducted statistical significance tests, with 20 out of 45 comparisons (≈44.4%) showing significance, supporting the effectiveness of our GyroHom. The relatively low ratio is mainly due to intrinsic EEG characteristics, low SNR, large inter-/intra-session domain shifts, and high subject variability, which naturally broaden confidence intervals.

---

### Official Review · Reviewer_UbxM · 2025-07-03

**Clarity:** 3
**Significance:** 3
**Originality:** 3
**Rating:** 4
**Confidence:** 1

**Summary:**

This paper proposes GyroAttr, an attention framework that unifies operations across diverse matrix manifolds through a common gyrovector space formulation. GyroAttr is applied to three types of manifolds: the SPD manifold, the Grassmannian manifold, and the SPSD manifold. Experimental results show that the proposed method gets improvements in related tasks.

**Questions:**

1.	Some case studies or visualizations would help clarify the limitations of previous work. I suggest the authors either include some experiments to illustrate the issue or explain whether this is a commonly recognized challenge in the field with appropriate citations.
2.	Is it possible to align the experimental setup with that of any previously published work? Alternatively, could the authors clarify why the evaluation metrics used in this paper are more convincing?

**Ethical Concerns:**

["NO or VERY MINOR ethics concerns only"]

**Quality:**

3

**Strengths And Weaknesses:**

Pros
1.	The experimental results show improvement
2.	The network architecture is described in sufficient detail, which is helpful for reproducibility.
3.	The effectiveness of GyroAttr on multiple types of manifolds shows the generality of the proposed framework
Cons
1.	The motivation of this paper (lines 100–108) is not well supported. For example, the authors claim that "these approaches typically rely on geometry-specific operations, such as the BiMap layer [32] for SPD manifolds or the FrMap layer [33] for Grassmannian manifolds, which hinders their applicability to other manifolds and limits generalization." Although Section 3.1 revisits the network structures of previous methods, it is important to note that deep learning models are often treated as a kind of black-box system. Therefore, differences in network structure are insufficient to conclude that previous methods exist significant limitations and the proposed method effectively addresses them.
2.	Although the reported experimental results suggest that the proposed method outperforms existing approaches, the inconsistency in evaluation metrics raises concerns, especially since the authors claim in line 212: “Building on prior works [54, 40], we evaluate on four benchmarking datasets.” However, upon reviewing the cited works, it appears that there are no inter-session or inter-subject evaluation settings in Matt. In the SPD setting, the evaluation metrics used in this paper also appear to differ significantly.

---

> ### Author Rebuttal · Authors · 2025-07-31
>
> We thank Reviewer $\textcolor{orange}{UbxM (R3)}$ for the constructive suggestions and insightful comments! In the following, we respond to the concerns in detail. 😄
>
> ## **1. The limitations of previous work**
>
> In Euclidean spaces, basic layers like convolution and fully connected operations provide a flexible basis for diverse network designs. However, in the context of manifold-valued data, there are no counterparts to these foundational layers due to geometry-specific constraints. This limits the flexibility of layer design and hinders reuse across different manifolds. Although deep learning models are often treated as black-box systems, it is essential in such settings to consider whether these layers (i) preserve the input’s geometric structure and (ii) effectively leverage the underlying Riemannian geometry.
>
> **Preserving geometric structure.** The commonly used BiMap layer is inherently restricted to SPD matrices of shape $d \times d$ and operates as:
> $$
> \mathrm{BiMap}(X) = W^\top X W,
> $$
> where $X \in  S^d$ and $W \in \mathbb{R}^{d \times k}$. A fundamental requirement of this operation is that the input must be square. However, for data lying on the Grassmannian manifold, such as subspace features $Y \in \mathbb{R}^{d \times q}$, this assumption no longer holds. Since $Y$ is rectangular and not square, BiMap becomes ill-defined and cannot be directly applied.
>
> Conversely, the FrMap layer, used in Grassmannian settings, applies a transformation of the form:
> $$
> \mathrm{FrMap}(Y) = W Y,
> $$
> which is compatible with Grassmannian data. Although SPD matrices $X \in S^d$ can technically be input to this operation, it destroys the key structural properties: $WX$ is generally not symmetric and not positive definite, thus violating the manifold constraint and pushing the data off the SPD manifold. In summary, these geometry-specific designs lack portability and generality and are infeasible outside their intended manifolds.
>
> Additionally, geometry-specific layers lead to fragmented architectures with low reusability. This increases implementation overhead. In contrast, GyroAtt adopts a unified architecture that seamlessly supports SPD, SPSD, and Grassmann manifolds without redesign, enabling consistent and modular learning across diverse geometries.
>
>
> **Leveraging Riemannian geometry.** From a theoretical standpoint, Gyro Hom offers three key advantages:
>   - **Gyro Algebraic Structure Simplifies Manifold Layer Design**: Gyro spaces define operators such as gyro addition ⊕ and scalar multiplication ⊗, providing an algebraic framework that is both Euclidean-like and geometry-aware. This facilitates the construction of manifold layers while respecting intrinsic geometry.
>   - **Respecting Gyro Algebraic Structure via Gyro Homomorphisms**: GyroAtt employs gyro homomorphisms in its transformation layers, preserving the underlying algebraic structure and Riemannian geometry. In contrast, MAtt and GDLNet rely on BiMap/FrMap, which only partially adhere to Riemannian geometry. While these layers enforce manifold constraints, they lack isometric properties, potentially distorting the intrinsic geometric structure. For example, the distance $d(S_1, S_2)$ between two SPD matrices $S_1$ and $S_2$ may not be preserved after transformation, i.e., $d(\mathrm{BiMap}(S_1), \mathrm{BiMap}(S_2)) \neq d(S_1, S_2)$. This limitation undermines their ability to fully respect and retain the original geometric properties of the data, which can negatively impact performance.
>   - **Nonlinear Activation**: Existing SPD networks often use ReEig for nonlinear activation, which enforces SPD-ness through eigenvalue regularization, but lacks geometric interpretability. In contrast, GyroAtt adopts a matrix power-based nonlinear activation function that essential activates the latent Riemannian geometry. As shown in [a, Fig. 1], the matrix power can deform any given Riemannian metric. For instance, as the parameter $\theta \to 0$, the matrix power deform AIM into LEM.
>
> ## **2. Clarification on Experimental Setup and Evaluation Metrics**
>
> We thank the reviewer for this important question. Our experimental setup closely follows the protocols established in prior works, and we clarify the details as follows:
>
> - For MAMEM-SSVEP-II and BCI-ERN datasets, we strictly adhere to the evaluation setup and metrics described in MAtt [b]. Specifically, we use sessions 1–3 for training, session 4 for validation, and session 5 for testing. We also adopt the same evaluation metrics as in [b]: accuracy for MAMEM-SSVEP-II and AUC for BCI-ERN to account for class imbalance.
>
> - For BNCI2014001 and BNCI2015001 datasets, our evaluation is based on the protocol in [c]. We adopt their inter-session and inter-subject evaluation splits and use the same performance metric: balanced accuracy, computed as the average recall across all classes.
>
> - Regarding preprocessing, we follow [b] for MAMEM-SSVEP-II and BCI-ERN, and [c] for BNCI2014001 and BNCI2015001 to ensure consistency and reproducibility. Details of the used datasets and preprocessing steps are provided in App.D.1.
>
> We will revise the manuscript to explicitly clarify these details and remove any ambiguity around our evaluation protocol.
>
> **References**
> > [a] RMLR: Extending Multinomial Logistic Regression into General Geometries.
> >
> > [b] MAtt: A manifold attention network for EEG decoding.
> >
> > [c] SPD domain-specific batch normalization to crack interpretable unsupervised domain adaptation in EEG.

---

> ### Author Response · Authors · 2025-08-07
> **Gentle Reminder**
>
> Dear Reviewer UbxM,
>
> We sincerely thank you again for your valuable comments. We hope our previous response has addressed your concerns regarding the limitations of prior works and the design of our experimental setup and evaluation metrics.
>
> As the discussion phase is drawing to a close, we would like to kindly check whether our response has addressed the points you raised. If there are any remaining issues or suggestions, we would be more than happy to provide further clarification. Your feedback is truly appreciated and would be instrumental in helping us further improve the quality of the paper.
>
> Warm regards

---

### Official Review · Reviewer_2oRe · 2025-07-03

**Clarity:** 4
**Significance:** 3
**Originality:** 3
**Rating:** 5
**Confidence:** 3

**Summary:**

The paper proposes GyroAtt, a framework for generalizing attention in (Euclidean) deep neural networks to a class of matrix manifold spaces (including the SPD, SPSD and Grassmannian manifolds). Methods using the framework are empirically validated on 4 real EEG datasets.

**Questions:**

*Key points*:
1. Are there other obvious applications of the GyroAtt framework outside of EEG-related data? Elaborating a bit on the "various applications" on Line 16 would help make the motivation and potential impact more clear.

2. Line 1051 states plans to release code for accessing the data. Are there no plans to release implementations of GyroAtt?

**Ethical Concerns:**

["NO or VERY MINOR ethics concerns only"]

**Final Justification:**

Both of the issues I originally raised (potential applications outside of EEG data, and opensource code) are addressed in the rebuttal. I think it's a mathematically solid paper that will have impact across a number of applications, (but I can't certify that it's a "technically flawless paper with groundbreaking impact) so I maintain my rating of 5.

**Limitations:**

yes

**Quality:**

4

**Strengths And Weaknesses:**

*Quality*:
- (+) mathematically rigorous
- (+) extensive empirical evaluation
- (+) coherent and insightful framework for learning across different geometries

*Clarity*:
- (+) quite well-written and thorough

*Significance*:
- (+) potential for good impact within the field of BCI, and I imagine (though am less familiar with) many other applications
- (+) also interesting more abstractly for the mathematical deep learning community
- (-) seems to be no plans for sharing implementations of GyroAtt, which I consider a substantial weakness

*Originality*:
- (+) novel generalization/unification of existing approaches

---

> ### Author Rebuttal · Authors · 2025-07-31
>
> We thank Reviewer $\textcolor{red}{2oRe (R2)}$ for the careful review and the suggestive comments. Below, we address the comments in detail. 😄
>
> ## **1. GyroAtt is applicable to a wide range of manifold learning tasks beyond EEG.**
>
> Thank you for the insightful question. While our experiments focus on EEG decoding, the GyroAtt framework is broadly applicable to learning tasks involving manifold-valued data. Data that lie on matrix manifolds—such as SPD, SPSD, or Grassmannian manifolds—are ubiquitous in various application domains, such as medical imaging [a], shape analysis [b], drone classification [c], image recognition [d], human behavior analysis [e, f], EEG [g], graph classification [h], and knowledge graph completion (KGC) [i, j].
>
> To further support its generalizability, we include **two additional experiments** beyond EEG:
>
> **Graph Node Classification.** We evaluate GyroAtt-SPD on two benchmark datasets—**Pubmed** and **Cora**—using SPD-GAT [h] as the base model, where the standard attention module is replaced with either GyroAtt-SPD or MAtt. Results are summarized below:
>
> |**Method**|**Pubmed**|**Cora**|
> |-|-|-|
> |H-GAT|77.5±1.6|78.1±1.1|
> |SPD-GAT|77.8±0.6|79.4±0.6|
> |SPD-GAT-MAtt|77.5±0.6|80.1±1.1|
> |**SPD-GAT-GyroAtt**|**78.5±0.4**|**81.6±1.0**|
>
> **Drone Recognition.** We further test GyroAtt on the **Radar** dataset for drone classification, following the GDLNet [g] architecture. The results are as follows:
>
> |**Method**|**Accuracy(%)**|
> |-|-|
> |MAtt|96.8±1.1|
> |GDLNet|94.7±0.9|
> |**GyroAtt-SPD**|**98.5±0.8**|
> |GyroAtt-Gr|96.2±1.2|
> |GyroAtt-SPSD|96.9±0.9|
>
>
> GyroAtt-SPD again achieves the highest accuracy, further validating its adaptability to non-EEG modalities such as radar signals.
>
>
> ## **2. We will release the full GyroAtt implementation**
>
> Apologies for the ambiguity. We fully intend to release the complete GyroAtt implementation, including training pipelines and model definitions, alongside the data access code. The sentence in Line 1051 will be updated in the final version to reflect this more clearly.
>
>
> **References**
>
> > [a] Fast and Simple Computations on Tensors with Log-Euclidean Metrics.
> >
> > [b] Covariance Descriptors for 3D Shape Matching and Retrieval.
> >
> > [c] Riemannian Batch Normalization for SPD Neural Networks.
> >
> > [d] DeepKSPD: Learning Kernel-Matrix-Based SPD Representation for Fine-Grained Image Recognition.
> >
> > [e] Deep Manifold Learning of Symmetric Positive Definite Matrices with Application to Face Recognition.
> >
> > [f] GeomNet: A Neural Network Based on Riemannian Geometries of SPD Matrix Space and Cholesky Space for 3D Skeleton-Based Interaction Recognition.
> >
> > [g] MAtt: A Manifold Attention Network for EEG Decoding.
> >
> > [h] Modeling Graphs Beyond Hyperbolic: Graph Neural Networks in Symmetric Positive Definite Matrices.
> >
> > [i] The Gyro-Structure of Some Matrix Manifolds.
> >
> > [j] Matrix Manifold Neural Networks++.

---

> > ### Comment · Reviewer_2oRe · 2025-08-04
> >
> > Thanks for the thorough rebuttal. It addresses all of my concerns, and the extra applications here (and the promise of released code) along with the runtime+significance results in the other rebuttals suggest to me that the paper will have good impact.

---

> ### Author Response · Authors · 2025-08-06
> **Thank you for your prompt reply**
>
> Thank you for your prompt and positive reply! If you have any further suggestions or insights, we would be very glad to hear them! 😄

---

### Official Review · Reviewer_5ZCS · 2025-07-04

**Clarity:** 2
**Significance:** 2
**Originality:** 3
**Rating:** 5
**Confidence:** 4

**Summary:**

This paper proposes a new framework for computing transformer-like attention on general gyrovector spaces for data with underlying matrix geometry.  Their Gyro Attention (GyroAtt) framework is applied in three geometric settings, namely on the SPD manifold, on the SPSD manifold, and on the Grassmannian manifold. The real-world application under consideration is EEG signals classification using four separate datasets. Comparisons are performed against existing deep learning architectures designed for matrix geometry.

**Questions:**

1. Since the datasets are on the small size,  are all the comparisons reported statistically significant in terms of improvements offered?
2. What is the wall clock time for optimising each attentional framework
3. How many model parameters do each considered attentional spaces have; how about the baselines being compared against?
4. The authors should consider including a glossary for abbreviations as it is currently difficult to go back and forth within the manuscript each time one encounters an abbreviation.
5. In section C3., why do the authors not choose to run a common principal components algorithm for initialisation instead of disjoint decompositions?

**Ethical Concerns:**

["NO or VERY MINOR ethics concerns only"]

**Final Justification:**

I have read through the comments and appreciate the work that the authors have put in to address the concerns brought up during the review process. I have increased my final score to an accept.

**Limitations:**

Currently, within the datasets under consideration, the matrix sizes are <50x50. In the context of the questions on computation time and time to convergence, it is not clear whether Gyro-Attn would scale to larger matrices (such as those from fMRI brain-connectomes) for a broader range of applications

**Quality:**

3

**Strengths And Weaknesses:**

STRENGTHS:

1. The Gyro-Attentional framework is an interesting and theoretically sound geometric extension of attention onto matrix geometries and manifold valued data, such as covariance matrices from EEG (or fMRI).
2. The approach is principled and well motivated. The paper explains the relevant theory clearly with sufficient explanation and offers a unified framework applicable under different geometric assumptions on the manifold structure.
3. Improved performance is observed for the EEG datasets for signal classification as compared against existing deep neural networks for matrix manifold data.

WEAKNESSES:

1. The authors gloss over how computationally expensive the optimisation is for their method, especially compared to the alternatives proposed. For example, the initialisation requires them to run multiple SVD like decompositions, running Karcher flow at every epoch incurs additional cost.

2. There are very few details available on the baseline methods chosen, which makes it hard to evaluate their performance gains hard to contextualise.  Do the authors compare against graph neural networks (an alternate strategy for classification, but with potential for saving on computation with sparsity).

3. How many model parameters do each have relative to the proposed framework? It is unclear whether this can be a contributing factor for the observed performance improvements.

---

> ### Author Rebuttal · Authors · 2025-07-31
>
> We thank Reviewer $\textcolor{blue}{\rm{5ZCS (R1)}}$ for the careful review and the suggestive comments. Below, we address the comments in detail. 😊
>
> ## **1. Statistical Significance of Reported Improvements.**
> Table A: Comparison between GyroAtt and the baseline MAtt, where * donetes $p$-vale < 0.05 and n.s. denotes "not significant".
> ||MAMEM|BCI-CHA|BNCI2014001 inter-session|BNCI2014001 inter-subject|BNCI2015001 inter-session|BNCI2015001 inter-subject|
> |-|-|-|-|-|-|-|
> Gains|3.2%|3.4%|7.8%|8.9%|5.4%|14.8%|
> Statistical significance|*|*|*|n.s.|n.s.|*|
>
> While some confidence intervals overlap, the improvements are substantial, consistent across all benchmarks. Particularly in the inter-subject setting, which is inherently more challenging, a +14.8% gain highlights the practical effectiveness of GyroAtt. We will clarify the statistical testing procedure in the final version.
>
>
> ## **2. Experimental comparison of time efficiency.**
> Table B: Training time (s/epoch) comparison of different manifold attention networks on the BNCI2014001 and BNCI2015001 datasets.
> |Methods|Metrics|BNCI2014001 inter-session|BNCI2014001 inter-subject|BNCI2015001 inter-session|BNCI2015001 inter-subject|
> |-|-|-|-|-|-|
> |MAtt|LEM|4.86|89.12|2.74|56.78|
> |GDLNet|ONB|4.55|88.59|1.71|47.66|
> |GyroAtt-Gr|ONB|5.28|100.20|1.98|54.07|
> |GyroAtt-SPD|AIM|10.36|149.14|4.44|81.73|
> |GyroAtt-SPD|LEM|7.89|138.59|4.28|80.66|
> |GyroAtt-SPD|LCM|4.11|87.44|2.42|49.86|
> |GyroAtt-SPSD|AIM|6.78|123.44|2.37|65.30|
> |GyroAtt-SPSD|LEM|6.54|116.49|2.34|64.23|
> |GyroAtt-SPSD|LCM|5.71|103.57|2.16|58.50|
>
> For practical comparison, we measured the average training time per epoch (seconds) for our GyroAtt variants and baseline models MAtt and GDLNet on the BNCI2014001 and BNCI2015001 datasets. As shown in Table B, GyroAtt-Gr is slower than GDLNet, mainly due to additional SVD operations required by the WFM aggregator. Similarly, GyroAtt-SPSD is less efficient than both baselines across all metrics. In contrast, GyroAtt-SPD with the LCM metric achieves slightly better runtime than MAtt. In general, LCM-based GyroAtt variants are faster than their AIM- and LEM-based counterparts, since Cholesky decomposition (used in LCM) is more efficient than SVD-based operations. AIM is consistently slower than LEM due to the higher computational cost of eigenvalue computations. Although some GyroAtt variants introduce additional computational overhead compared to baseline methods, especially under certain metrics, the performance improvements justify the increased complexity. Selecting efficient metrics like LCM can mitigate computational costs, making GyroAtt practical for real-world applications. Please refer to Appendix D.7 for further implementation and profiling details.
>
>
> ## **3. Analysis of Parameter Counts.**
> Table C: Learnable parameters (in millions) of different methods across four EEG datasets.
> |Method|MAMEM|BCI-CHA|BNCI2014001|BNCI2015001|
> |-|-|-|-|-|
> |MAtt|0.07M|0.03M|0.03M|0.01M|
> |GDLNet|0.51M|0.05M|0.03M|0.01M|
> |GyroAtt-SPD|0.11M|0.08M|0.04M|0.01M|
> |GyroAtt-Gr|0.11M|0.07M|0.04M|0.01M|
> |GyroAtt-SPSD|0.12M|0.09M|0.05M|0.01M|
>
> For a manifold attention layer mapping from $S^d \to S^d$, the number of parameters in GyroAtt-SPD is:
> $\frac{3}{2}n(n-1) + \frac{1}{2}n(n+1)=2n ^2-\frac{1}{2}n$, where the first term corresponds to the transformation layers for $Q$, $K$, and $V$, and the second term represents the bias parameters. In comparison, the baseline **MAtt** uses $3n^2$ parameters per attention block.
>
> As summarized in Table C, all GyroAtt variants have comparable or even fewer total learnable parameters than MAtt across four EEG datasets, and are significantly more compact than GDLNet, particularly on MAMEM-SSVEP-II. Importantly, GyroAtt consistently achieves better performance despite this compactness, indicating that the gains are not simply due to increased model capacity.
>
>
>
> ## **4. The glossary for abbreviations.**
> We have included a glossary of notations and abbreviations in Appendix B, specifically in Tables 11 and 12, to improve readability and reduce the need to reference earlier sections when encountering abbreviations.
>
> ## **5. Clarification on Initialization Strategy in Section C.3**
>
> We appreciate the reviewer’s suggestion regarding the use of a common principal components algorithm (CPCA) for initialization. We would like to clarify a potential misunderstanding:
>
> In Line 2 of Algorithm 6, we initialize the common subspace $U^m$ with a fixed canonical basis $\widetilde{I}_{n,q}$, not via decompositions of individual samples. This line is intended to indicate that the default initialization of the Grassmannian Fréchet mean starts at the identity point on the manifold. We acknowledge that this notation may have caused confusion and will revise it in the final version for clarity.
>
> Importantly, the estimation of the common subspace is learned dynamically in the subsequent steps. Specifically, we first compute the Grassmannian Fréchet mean of the individual subspaces $\{ U_i \}$ obtained via SVD of each sample, and then update $U^m$ via geodesic interpolation toward this mean, and the same choice is also adopted in [a].
>
> ## **6. Comparison with SOTA Baselines.**
>
> Our main baselines are strong SOTA methods specifically designed for manifold-valued EEG data:
>
> * On the MAMEM-SSVEP-II and BCI-ERN datasets, our main baselines are GDLNet [b] and MAtt [c], both of which represent state-of-the-art approaches for SPD and Grassmannian manifolds, respectively.
> * On the BNCI2014001 and BNCI2015001 datasets, we include TSMNet [d] and Graph-CSPNet [e], both of which are recent SOTA models widely used in motor imagery tasks. Notably, Graph-CSPNet [e] is a recent GNN-based approach tailored for motor imagery classification.
>
> To ensure a fair comparison, we also follow the exact same data preprocessing and evaluation protocol as used in the original baseline papers: For MAMEM-SSVEP-II and BCI-ERN datasets, our setting matches that of MAtt [c]; For BNCI2014001 and BNCI2015001 datasets, we follow the exact pipeline from TSMNet [d].
>
>
> ## **7. Comparison with Graph Neural Networks.**
>
> ### **7.1 Comparison with Graph-Based EEG Methods.**
> Table D: Accuracy (\%) comparison between Graph-CSPNet and GyroAtt on EEG datasets.
> |Method|BNCI2014001 inter-session|BNCI2014001 inter-subject|BNCI2015001 inter-session|BNCI2015001 inter-subject|
> |-|-|-|-|-|
> |Graph-CSPNet|71.9 ± 13.3|45.2 ± 9.3|79.8 ± 14.6|64.2 ± 13.4|
> |GyroAtt-SPD-AIM|**75.4 ± 7.1**|**53.1 ± 14.8**|**86.2 ± 4.5**|**77.9 ± 13.0**|
> |GyroAtt-SPD-LEM|75.3 ± 6.5|52.3 ± 14.1|85.7 ± 5.5|76.6 ± 13.7|
> |GyroAtt-SPD-LCM|74.2 ± 7.8|52.4 ± 15.6|84.7 ± 6.6|75.5 ± 13.8|
>
> We compare GyroAtt against Graph-CSPNet [e], a recent SPD-based GNN method designed for motor imagery classification. As shown in the table below, all variants of GyroAtt outperform Graph-CSPNet across both inter-session and inter-subject settings on BNCI2014001 and BNCI2015001.
>
> ### **7.2. Additional Comparison on Graph Attention Neural Networks.**
>
> Table E: Accuracy (\%) comparison of SPD-GAT-GyroAtt under graph node classification.
> Method|Pubmed|Cora|
> |-|-|-|
> H-GAT|77.5±1.6|78.1±1.1|
> SPD-GAT|77.8±0.6|79.4±0.6|
> SPD-GAT-MAtt|77.5±0.6|80.1±1.1|
> SPD-GAT-GyroAtt|**78.5±0.4**|**81.6±1.0**|
>
>
> To further validate GyroAtt, we compare GyroAtt against hyerpbolic and SPD attention in standard GNN settings using graph node classification tasks.
>
> We focus on the graph attention (GAT) setting. H-GAT [f] serves as a hyperbolic graph attention baseline, while SPD-GAT [g] is the SPD counterpart. By replacing SPD-GAT’s attention layer with MAtt and GyroAtt, we demonstrate that our attention module consistently improves performance. This highlights the versatility of GyroAtt across both manifold-valued and graph-structured domains.
>
>
> ## **8. Computation Time and Convergence on Large Matrices.**
>
> Table F: Scalability of GyroAtt-SPD on BNCI2015001 with varying input matrix sizes.
> |Matrix Size|Time per epoch (s)|Accuracy|
> |-|-|-|
> |43×43|**0.98**|**85.7±5.5**|
> |50×50|1.13|85.21±5.8|
> |60×60|1.37|85.01±6.1|
> |70×70|1.86|84.91±5.8|
> |80×80|2.34|84.46±5.1|
> |90×90|3.82|84.08±5.6|
> |100×100|5.10|84.35±5.9|
>
> We further conducted scalability experiments using GyroAtt-SPD on the BNCI2015001 dataset by adjusting the output dimensionality of the feature extraction layer, thereby increasing the size of the input SPD matrices from $43 \times 43$ to $100 \times 100$. As shown in Table F, the per-epoch time increases moderately with matrix size, and convergence is consistently achieved within a practical timeframe. These results confirm that GyroAtt remains computationally feasible even for relatively large matrices, in agreement with theoretical expectations.
>
> In terms of accuracy, we observe that increasing the matrix size from 43 to 100 results in only marginal variations in performance. This suggests a saturation effect often seen in low-dimensional EEG tasks, where expanding the representational capacity beyond a certain point offers little additional discriminative power. The relevant signal information is likely already captured at smaller sizes, which accounts for the stable test accuracy across the entire range.
>
>
> **References**
>
> > [a] Matrix Manifold Neural Networks++.
> >
> > [b] A Grassmannian Manifold Self-Attention Network for Signal Classification.
> >
> > [c] MAtt: A manifold attention network for EEG decoding.
> >
> > [d] SPD domain-specific batch normalization to crack interpretable unsupervised domain adaptation in EEG.
> >
> > [e] Graph neural networks on SPD manifolds for motor imagery classification: A perspective from the time-frequency analysis.
> >
> > [f] Hyperbolic graph attention network.
> >
> > [g] Modeling graphs beyond hyperbolic: graphneural networks in symmetric positive definite matrices.

---

> ### Author Response · Authors · 2025-08-07
> **Further Enhancements to Abbreviation Readability**
>
> Dear reviewer, we have revised our manuscript with **`glossaries-extra`** package, which can automatically generate hyperlinks for each acronym to their definitions. This will improve clarity and make the manuscript easier to navigate. Although we cannot upload a new PDF during the rebuttal, this will be reflected in the final version.

---

### Note · Authors · 2025-08-15

Dear Area Chair and Reviewers,

We sincerely thank you for your time and thoughtful evaluations. We would like to share several points for your consideration:

*Reviewer 5ZCS*
* **Statistical tests** (Table A) show that GyroAtt delivers substantial and statistically significant gains over baselines.
* **Runtime and parameter comparisons** (Tables B & C) confirm its computational efficiency.
* **Clarified initialization strategies** in Section C.3.
* **Justified the selection of baselines**, including Graph-CSPNet and SPD-GAT, and showed that GyroAtt achieves accuracy gains on EEG and graph tasks (Tables D & E).
* **Experiments on larger matrices** (Table F) demonstrate its scalability.

*Reviewer 2oRe*
* **Confirmed applicability beyond EEG** with gains on graph tasks and radar signal recognition.
* **Source codes and packages** will be released.

*Reviewer UbxM*
* **GyroAtt provides a general attention mechanism** across SPD, SPSD, and Grassmannian Manifolds, whereas previous methods are designed under specific geometries.
* **Fair comparison** because it strictly follows prior protocols.

*Reviewer 3rxG*
* **Flexible use based on interests and tasks**. The built package supports secondary development. Unlike the stereographic model, SPD, SPSD, and Grassmannian differ in geometry and modeling, so a dynamically unified network may have limited flexibility and practical benefit.
* GyroHom are constructed from **isometries on the homogeneous spaces** (Theorem A).
* Extended GyroAtt to the stereographic model.
* Compared with manifold MLP-like layers (BiMap, SPD-FC, Tangent-FC, FrMap, Gr-Scaling) on the SPD and Grassmannian manifolds, **GyroAtt achieves better trade-offs among accuracy, efficiency, and stability, with gains in most cases**.

**Positive feedback**
* *Reviewer 5ZCS:*
  * **Interesting and theoretically sound** geometric extension of attention.
  * **Improved performance** on EEG datasets.
* *Reviewer 2oRe:*
  * **Mathematically rigorous** with extensive empirical evaluation.
  * **Coherent, insightful** cross-geometry framework; well-written.
  * **Potential for strong impact** in BCI and beyond; relevant to mathematical DL.
* *Reviewer UbxM:*
  * **Improved performance**.
  * **Reproducible** due to the architecture details.
  * **Generality** due to the effectiveness across multiple manifolds.
* *Reviewer 3rxG:*
  * **Well-written and easy to follow**.
  * **Novelty** of the proposed homomorphisms.

Best regards,

Authors of #1957

---

### Decision · Program_Chairs · 2025-09-17

**Decision:**

Accept (poster)

**Comment:**

The reviewers acknowledged the well-motivated and rigorous method, the extensive evaluation, and the effectiveness of the method. Nevertheless, they expressed some concerns about the computational cost of the method, the limited detail on the baselines, the limited evidence of the impact of some aspects of the method, and the statistical significance of the results. The authors' feedback addressed some of these concerns, resulting in 3 reviewers recommending acceptance. Reviewer 3rxG nonetheless remained unconvinced by the rebuttal, recommending rejection. In a discussion with the AC, the authors pointed out the difference between the general framework that they propose and the unified framework suggested by 3rxG, able to dynamically adapt to data from matrix manifolds of different natures. The AC argues that the fact that the same framework can be applied to different matrix manifolds, even though it cannot handle different data types simultaneously, is sufficient, and thus agrees with the other reviewers that the paper meets the NeurIPS acceptance threshold. The authors are nonetheless encouraged to incorporate elements of their answers in the final version of the paper.